# CFTR function, pathology and pharmacology at single-molecule resolution

Jesper Levring[1], Daniel S. Terry[2], Zeliha Kilic[2], Gabriel Fitzgerald[3], Scott C. Blanchard[2✉] & Jue Chen[1,4✉]

The cystic fibrosis transmembrane conductance regulator (CFTR) is an anion channel that regulates salt and fluid homeostasis across epithelial membranes[1]. Alterations in CFTR cause cystic fibrosis, a fatal disease without a cure[2,3]. Electrophysiological properties of CFTR have been analysed for decades[4–6]. The structure of CFTR, determined in two globally distinct conformations, underscores its evolutionary relationship with other ATP-binding cassette transporters. However, direct correlations between the essential functions of CFTR and extant structures are lacking at present. Here we combine ensemble functional measurements, single-molecule fluorescence resonance energy transfer, electrophysiology and kinetic simulations to show that the two nucleotide-binding domains (NBDs) of human CFTR dimerize before channel opening. CFTR exhibits an allosteric gating mechanism in which conformational changes within the NBD-dimerized channel, governed by ATP hydrolysis, regulate chloride conductance. The potentiators ivacaftor and GLPG1837 enhance channel activity by increasing pore opening while NBDs are dimerized. Disease-causing substitutions proximal (G551D) or distal (L927P) to the ATPase site both reduce the efficiency of NBD dimerization. These findings collectively enable the framing of a gating mechanism that informs on the search for more efficacious clinical therapies.

CFTR belongs to the ATP-binding cassette transporter family of proteins, but uniquely functions as an ion channel[4]. It consists of two transmembrane domains that form an ion permeation pathway, two cytosolic NBDs that bind and hydrolyse ATP, and a cytosolic regulatory (R) domain that includes several phosphorylation sites. Decades of electrophysiological, biochemical and structural studies (reviewed in refs. [5,6]) established that CFTR activity requires phosphorylation of the R domain by protein kinase A (PKA)[7]. Once phosphorylated, ATP binding drives pore opening. CFTR contains two functionally distinct ATP-binding sites[8]. The 'consensus' site is catalytically competent, whereas the 'degenerate' site is not[9]. ATP hydrolysis at the consensus site leads to pore closure[10]. Pore opening in the absence of ATP and non-hydrolytic pore closure can occur, albeit very rarely[11,12].

Cryogenic electron microscopy (cryo-EM) studies of CFTR have thus far revealed two globally distinct conformations. In the absence of phosphorylation and ATP, CFTR forms a pore-closed conformation in which the NBDs are separated by approximately 20 Å, and the R domain sterically precludes NBD dimerization[13] (Fig. 1a). The phosphorylated and ATP-bound CFTR, structurally characterized using the hydrolysis-deficient E1371Q variant[14], exhibits a pre-hydrolytic conformation, in which the NBDs form a closed dimer with two ATP molecules bound at their interface (Fig. 1a).

Despite these advances, major gaps in our understanding of CFTR function and regulation remain. For example, although extant structures of CFTR indicate that large-scale conformational changes are required for channel opening, they fall short of addressing the mechanistic relationship between NBD dimerization and the gating mechanism. How ion permeation is coupled to ATP hydrolysis and NBD isomerization remains contested. One model proposes that in every gating cycle one round of ATP hydrolysis is coupled with one pore-opening event and one NBD-dimerization and NBD-separation event[13,15–17]. Alternative models posit that the NBDs remain dimerized through several gating cycles with only partial disengagement of the dimer interface at the consensus site[10,18,19]. The CFTR pore has been suggested to be either strictly[10,18] or probabilistically[19] coupled to nucleotide state in the consensus site. Attempts to differentiate these models have thus far been inconclusive. Moreover, the steps rate-limiting to CFTR activity in unaffected individuals and patients with cystic fibrosis, and thus most likely to be sensitive to pharmacological modulation, remain unclear.

To address these open questions, we undertook an integrative approach combining ensemble measurements of ATPase activity, single-molecule fluorescence resonance energy transfer (smFRET) imaging, electrophysiology and kinetic simulations to examine the structure–function relationship in human CFTR. The information obtained reveals an allosteric gating mechanism in which ATP-dependent NBD dimerization is insufficient to enable pore opening. Although phosphorylated CFTR predominantly occupies

[1]Laboratory of Membrane Biology and Biophysics, The Rockefeller University, New York, NY, USA. [2]Department of Structural Biology, St. Jude Children's Research Hospital, Memphis, TN, USA. [3]Department of Physiology and Biophysics, Weill Cornell Medicine, New York, NY, USA. [4]Howard Hughes Medical Institute, The Rockefeller University, New York, NY, USA. ✉e-mail: Scott.Blanchard@stjude.org; juechen@rockefeller.edu

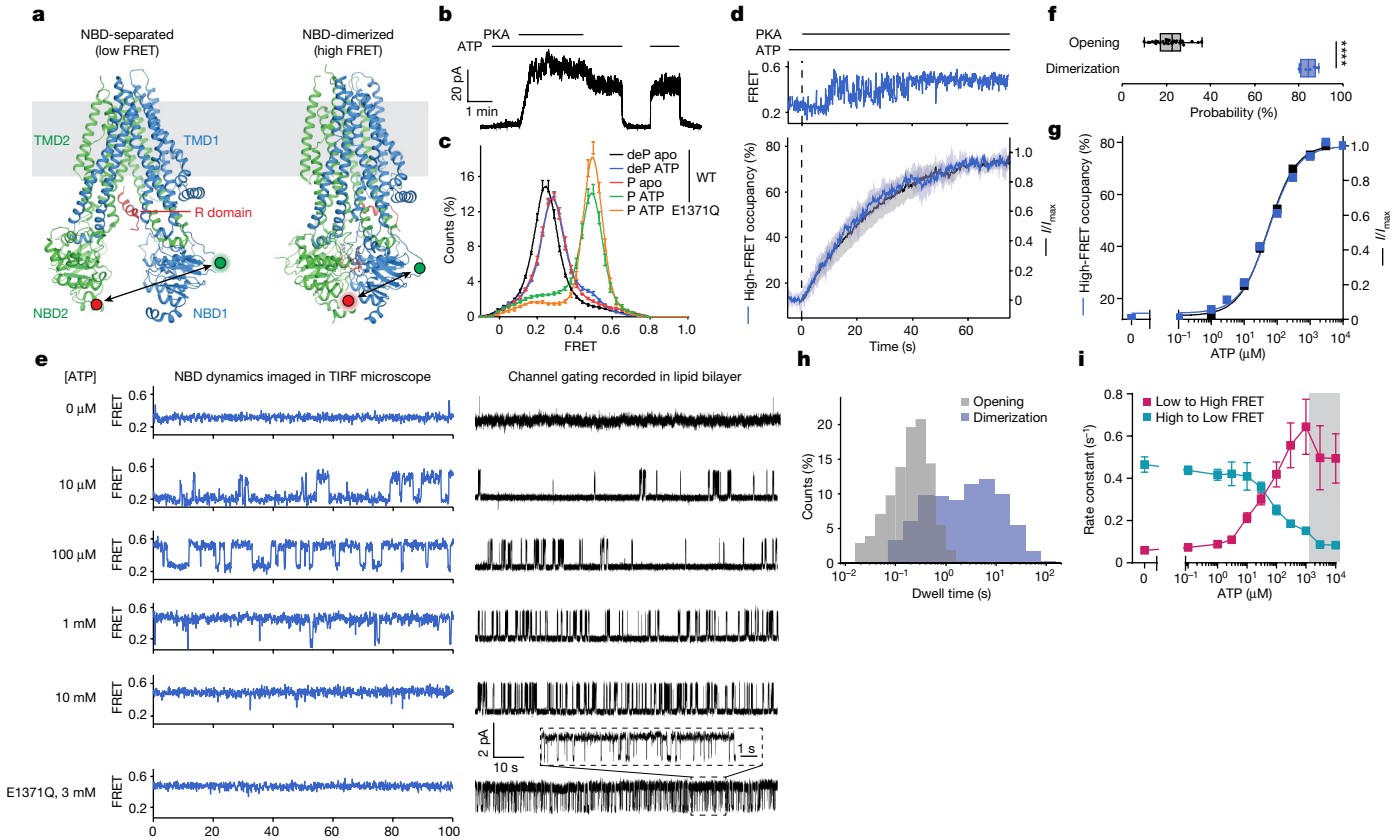

**Fig. 1 | Dependence of CFTR pore opening and NBD dimerization on phosphorylation and ATP. a**, CFTR structures in dephosphorylated, ATP-free (left, Protein Data Bank 5UAK) and phosphorylated, ATP-bound (right, Protein Data Bank 6MSM) states. Green and red circles indicate fluorophore positions. **b**, Inside-out excised patch showing dependence of wild-type (WT) CFTR-mediated currents on phosphorylation and ATP. Concentrations of 300 nM PKA and 3 mM ATP were used. **c**, FRET histograms for dephosphorylated (deP) and phosphorylated (P) wild-type CFTR$_{FRET}$ in the presence and absence of ATP, and phosphorylated CFTR$_{FRET}$(E1371Q) with ATP. Data represent means and standard errors for $n$ independent experiments. $n$ = 6 for wild-type dephosphorylated and phosphorylated apo, $n$ = 5 for wild-type dephosphorylated with ATP, $n$ = 7 for wild-type phosphorylated with ATP, and $n$ = 3 for phosphorylated E1371Q with ATP. **d**, Activation of pore opening and increase in occupancy of the high-FRET state after application of 300 nM PKA (at the dashed line), in the presence of 3 mM ATP. Upper panel, representative smFRET trace of CFTR$_{FRET}$ during phosphorylation. Lower panel, population-wide time-dependent changes in current and high-FRET occupancy after PKA application. Data represent means and standard deviations (shaded area) for three patches and three FRET experiments.

**e**, Sample 100-s excerpts of traces from smFRET with phosphorylated CFTR$_{FRET}$ (left) and single-channel electrophysiology in lipid bilayers with phosphorylated wild-type CFTR (right) at the indicated ATP concentrations. In electrophysiology traces, upward deflections correspond to opening. The bottom traces are with the E1371Q variant in 3 mM ATP. **f**, Probabilities of opening and dimerization of phosphorylated CFTR in 3 mM ATP. Whiskers represent minima and maxima and boxes represent 25th, 50th and 75th percentiles for 39 bilayers and 8 FRET experiments. Statistical significance was tested by two-tailed Student's $t$-test (****$P$ = 2 × 10$^{-18}$). **g**, ATP dose responses of CFTR-mediated current and high-FRET-state occupancy. Responses were fitted using the Hill equation with an EC$_{50}$ of 53 ± 4 µM for opening and an EC$_{50}$ of 55 ± 8 µM for high-FRET occupancy. Hill coefficients were fixed to 1. **h**, Dwell-time distributions of opening and dimerization events for phosphorylated CFTR in 3 mM ATP. **i**, ATP dose responses for rates of transitioning between low- and high-FRET states for phosphorylated CFTR$_{FRET}$. Data represent means and standard errors for three experiments. The shaded area indicates the regime in which transitions are obscured by time averaging, resulting in erroneous rate estimates.

an NBD-dimerized conformation at physiological ATP concentration, downstream conformational changes in CFTR governed by ATP turnover are required for chloride conductance. Disease-associated alterations and the pharmacological potentiators ivacaftor and GLPG1837 influence the efficiency of the coupling between NBD dimerization and ion permeation. These findings identify an allosteric link between the catalytically competent ATP-binding site and the channel pore that functions as a critical rate-limiting conduit for physiological and pharmacological regulation in CFTR.

## CFTR variant for smFRET retains native activity

To enable smFRET imaging of the protein's conformational state, we sought to develop a variant of human CFTR that could be labelled with maleimide-activated donor and acceptor fluorophores. After substituting 16 of the 18 native cysteines (Extended Data Fig. 1a), we further introduced cysteines into NBD1 (T388C) and NBD2 (S1435C). This variant, CFTR$_{FRET}$, was labelled with maleimide-activated forms of self-healing donor (LD555) and acceptor (LD655) fluorophores[20] to create an NBD-dimerization sensor (Fig. 1a). Labelling of the two introduced cysteines was >90% specific (Extended Data Fig. 2a).

We next tested whether CFTR$_{FRET}$ retains the functional properties of the wild-type CFTR. Macroscopic currents were measured in excised inside-out membrane patches using unlabelled wild-type CFTR and CFTR$_{FRET}$, both fused to a carboxy-terminal GFP tag (Extended Data Fig. 1b–g). These data show that CFTR$_{FRET}$ conducted phosphorylation- and ATP-dependent currents and retained sensitivity to the potentiator

GLPG1837 in a manner indistinguishable from that of wild-type CFTR (Extended Data Fig. 1b,c). The time courses for current activation on ATP application and current relaxation on ATP withdrawal were also indistinguishable between wild-type CFTR and CFTR$_{FRET}$ (Extended Data Fig. 1d–g).

We further evaluated the effects of conjugating fluorophores to CFTR using purified protein. Digitonin-solubilized and fluorophore-labelled CFTR$_{FRET}$ (Extended Data Fig. 2a,b) hydrolysed ATP at a rate nearly identical to that of wild-type CFTR (Extended Data Fig. 1h). On reconstitution into synthetic planar lipid bilayers, the fluorophore-labelled CFTR$_{FRET}$ and wild-type CFTR (without fluorophore labels) exhibited similar current–voltage relationship, open probability, and response to GLPG1837 (Extended Data Fig. 1i–m). Single-channel conductance of fluorophore-labelled CFTR$_{FRET}$ was slightly higher (Extended Data Fig. 1i,j), possibly due to the C343S substitution, a residue bordering the pore. On the basis of these observations, we conclude that the conformational and gating dynamics of CFTR$_{FRET}$ closely recapitulate those of wild-type CFTR.

## NBD dimerization is insufficient for pore opening

To examine the relationship between ATP binding and NBD dimerization directly, we carried out smFRET imaging on digitonin-solubilized, C-terminally His-tagged CFTR$_{FRET}$ molecules that were surface-tethered within passivated microfluidic chambers via a streptavidin–biotin–tris-(NTA-Ni$^{2+}$) bridge (Extended Data Fig. 2c). Imaging was carried out using a wide-field total internal reflection fluorescence (TIRF) microscope equipped with scientific complementary metal–oxide sensor (sCMOS) detection and stopped-flow capabilities[21] at 10 or 100 ms time resolution. Monomeric CFTR$_{FRET}$ molecules were tethered with high specificity as demonstrated by near-quantitative release from the surface with imidazole (Extended Data Fig. 2d–g).

Based on extant structures, fluorophore-labelled CFTR$_{FRET}$ is expected to exhibit low FRET efficiency in NBD-separated conformations and higher FRET efficiency in NBD-dimerized conformations (Fig. 1a). Indeed, in the absence of ATP and phosphorylation, CFTR exhibited a homogeneous low-FRET-efficiency distribution centred at 0.25 ± 0.01 (mean ± s.d. across six repeats) and exhibited few, if any, FRET fluctuations (Fig. 1c and Extended Data Fig. 3a). Consistent with current increase on phosphorylation and ATP addition (Fig. 1b), smFRET measurements also showed that adding ATP to phosphorylated CFTR caused a shift to higher FRET efficiency (0.49 ± 0.02), in which only brief excursions to lower-FRET states were evidenced (Fig. 1c and Extended Data Fig. 3d). Substitution of the catalytic base in the consensus site (E1371Q), which prevents ATP hydrolysis, further stabilized CFTR in higher-FRET-efficiency conformations (Fig. 1c). On the basis of these observations, we ascribed the ≈0.25 and ≈0.49 FRET states to NBD-separated and NBD-dimerized CFTR conformations evidenced by cryo-EM, respectively.

In contrast to the case for phosphorylated CFTR, addition of ATP to the dephosphorylated channel caused only a small shift in FRET efficiency, from ≈0.25 to 0.28 ± 0.01 (Fig. 1c and Extended Data Fig. 3b). The FRET distribution of phosphorylated, ATP-free CFTR was also centred at 0.28 ± 0.02 (Fig. 1c and Extended Data Fig. 3c). To explore the molecular basis of this shift, we determined the cryo-EM structure of the dephosphorylated wild-type CFTR in the presence of 3 mM ATP to 4.3 Å resolution (Extended Data Fig. 4 and Extended Data Table 1). Consistent with the smFRET data, the overall CFTR architecture was largely indistinguishable from that of the ATP-free CFTR structure. However, at both NBD1 and NBD2 binding sites, density corresponding to the ATP molecule was clearly evidenced (Extended Data Fig. 4d). These data indicate that ATP binding to the dephosphorylated CFTR does not induce any global conformational change. The small shift in FRET efficiency is probably due to local changes that affect either the position and/or dynamics of the sites of labelling.

Consistent with the gradual increase in open probability observed for single channels[6], pre-steady-state measurements of PKA-mediated CFTR phosphorylation in the presence of saturating ATP (3 mM) revealed that individual CFTR$_{FRET}$ molecules did not always instantaneously transition to a stably NBD-dimerized state (Fig. 1d and Extended Data Fig. 3m). Instead, stable NBD dimerization was achieved through processes that involved rapid sampling of NBD-separated and NBD-dimerized states. Parallel electrophysiological recordings revealed matching progression of current activation (Fig. 1d). NBD dimerization was fully reversible by phosphatase treatment (Extended Data Fig. 3n,p,r). By contrast, the E1371Q substitution slowed NBD separation (Extended Data Fig. 3o,q–r), indicating that ATP turnover facilitates NBD separation. These observations suggest that the gradual transition to steady-state channel activation probably reflects stochastic ATP binding to the individual NBDs and/or transient reinsertion of partially phosphorylated R domain, which resolve to stable NBD dimerization only when the R domain becomes fully phosphorylated and both NBDs are simultaneously ATP bound.

The ATP dose responses for NBD dimerization and current activation for fully phosphorylated CFTR strongly correlated, both yielding half-maximum effective concentration (EC$_{50}$) values of approximately 50 μM (Fig. 1e,g and Extended Data Fig. 5a,b,e). This finding is indicative of both processes being limited by the same underlying molecular event. NBD dimerization and channel-open probabilities differed substantially: at saturating ATP concentration, approximately 85% of CFTR$_{FRET}$ molecules were in the NBD-dimerized conformation but the channel-open probability was only 22% (Fig. 1f). We thus conclude that both conductive and non-conductive NBD-dimerized states must exist.

Consistent with this notion, the observed FRET dynamics differed from the evidenced gating dynamics (Fig. 1e,h). The rate of CFTR pore opening exhibits a saturable dependence on ATP concentration whereas the channel closing rate remains constant[16,22]. By contrast, NBD-dimerization and NBD-separation rates both changed monotonically with ATP concentration (Fig. 1i). At saturating ATP concentration, the dwell time of the NBD-dimerized state was approximately 20 times longer than that of the channel-open state (Fig. 1h), suggesting that FRET-silent processes occur within the NBD-dimerized conformation that trigger channel opening and closure and that only subtle rearrangements at the dimer interface are required for nucleotide exchange. This conclusion was supported by analogous imaging studies carried out at both 10 and 100 ms time resolutions (Extended Data Fig. 3e). We conclude that CFTR remains stably dimerized through multiple gating cycles or that transitions to partially separated NBD states are either FRET silent or occur on timescales markedly exceeding the temporal resolution of our measurements (100 s$^{-1}$). Both models nonetheless specify that NBD dimerization is not strictly coupled to channel opening.

At a cellular ATP to ADP ratio (≈10:1), fully phosphorylated CFTR$_{FRET}$ predominantly occupied dimerized conformations (Extended Data Fig. 3l), in line with CFTR predominantly binding ATP in the physiological setting. High ADP concentrations were, however, able to competitively inhibit both NBD dimerization and channel opening[23,24] (Extended Data Fig. 3f–l).

To validate the physiological relevance of these findings, we carried out targeted smFRET imaging studies with phosphorylated CFTR$_{FRET}$ reconstituted into proteoliposomes (Extended Data Fig. 6a). In the absence of ATP, membrane-embedded CFTR$_{FRET}$ molecules stably occupied the NBD-separated (0.28) FRET state (Extended Data Fig. 6b). On addition of 3 mM ATP, CFTR$_{FRET}$ molecules transitioned to the NBD-dimerized (0.49) FRET state (Extended Data Fig. 6c). The fraction of ATP-responsive molecules was reduced, probably due to degradations in channel activity or mixed orientations in the bilayer. However, the molecules that responded predominantly occupied NBD-dimerized conformations at steady state, with only rare, transient excursions to states with low FRET efficiency (Extended Data Fig. 6d). Also consistent

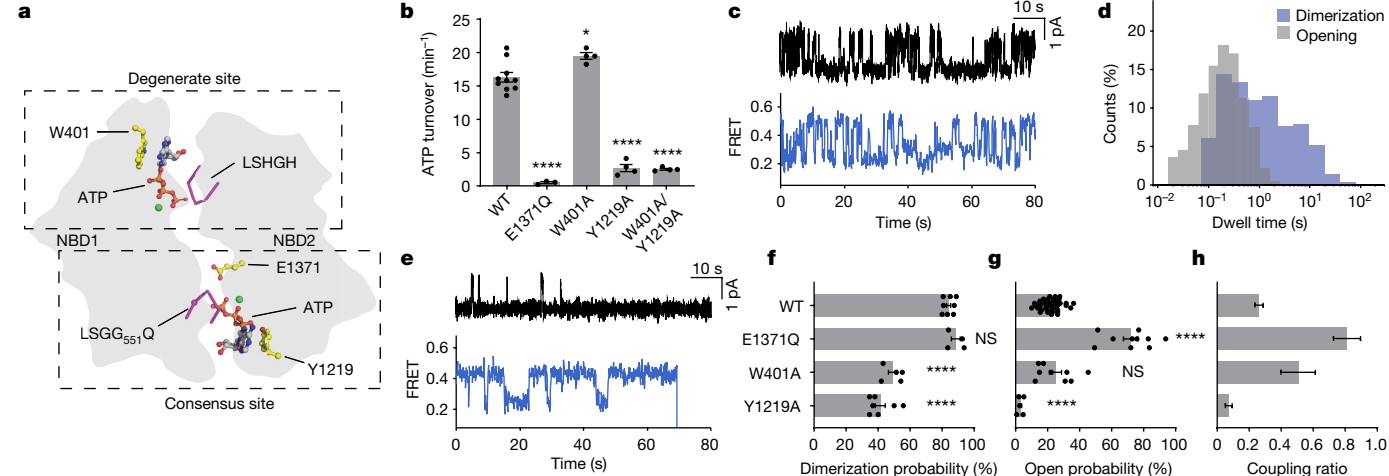

**Fig. 2 | Asymmetric contributions of degenerate and consensus ATP-binding sites. a**, Schematic of degenerate and consensus sites as viewed from the plasma membrane. **b**, Steady-state ATP hydrolysis rates for the wild-type CFTR and variants. Data represent means and standard errors for 10 (wild-type), 3 (E1371Q) or 4 (W401A, Y1219 and W401A/Y1219A) measurements. *$P = 0.014$; ****$P = 1.2 \times 10^{-11}$ (E1371Q), $2.7 \times 10^{-11}$ (Y1219A) and $2.1 \times 10^{-11}$ (W401A/Y1219A). **c**, Sample traces from single-channel electrophysiology (top) and smFRET (bottom) of the CFTR(W401A) variant. The substitution was made in wild-type CFTR and CFTR$_{FRET}$ backgrounds for electrophysiology and smFRET, respectively. In electrophysiology traces, upward deflections correspond to opening. **d**, Dwell-time distributions of opening and dimerization events for CFTR(W401A). **e**, As in **c**, but with the CFTR(Y1219A) variant. **f**, Dimerization probabilities of wild-type CFTR$_{FRET}$ and variants. Data represent means and standard errors for 8 (wild-type), 4 (E1371Q), 5 (W401A) and 7 (Y1219A) measurements. NS, not significant; ****$P = 8.0 \times 10^{-9}$ (W401A) and $4.4 \times 10^{-11}$ (Y1219A). **g**, Open probabilities of CFTR variants. Data represent means and standard errors for 39 (wild-type), 10 (E1371Q), 9 (W401A) and 5 (Y1291A) bilayers. ****$P = 10^{-15}$ (E1371Q) and $4.9 \times 10^{-5}$ (Y1219A). **h**, Coupling ratios of CFTR variants, defined as open probability divided by dimerization probability. Data represent means and standard errors. Phosphorylated CFTR variants at 3 mM ATP were used in all panels. For relevant panels, statistical significance relative to the wild-type was tested by one-way analysis of variance.

with expectation, CFTR$_{FRET}$ molecules relaxed to the NBD-separated state on ATP withdrawal (Extended Data Fig. 6e,f). These observations demonstrate that physical properties of the digitonin-solubilized CFTR$_{FRET}$ recapitulate those present in the lipid bilayer. To ensure the most robust signals and statistics, we carried out the remainder of our smFRET experiments using digitonin-solubilized CFTR$_{FRET}$.

## The ATP-binding sites contribute asymmetrically

In CFTR, the consensus ATP-binding site hydrolyses approximately 0.3 to 1 ATP molecules per second, whereas the degenerate site retains ATP for minutes[18,25]. We reasoned that ATP binding in the degenerate site alone is sufficient for NBD dimerization, whereas ATP binding in the consensus site is required for channel opening. To test this hypothesis, we sought to deconvolute the individual contributions of the two ATP-binding sites by substituting aromatic ATP-stacking residues, W401 and Y1219 (ref. [26]), with alanine to reduce the affinity for ATP at the degenerate and consensus sites, respectively (Fig. 2a). Whereas the degenerate site variant W401A hydrolysed ATP at a rate comparable to that of the wild-type CFTR, the ATPase activity of the consensus site variant Y1219A only marginally exceeded the background, established by analogous measurements of the E1371Q variant (Fig. 2b). The activity of the double variant (Y1219A/W401A) was indistinguishable from that of the Y1219A variant (Fig. 2b). These data show that the Y1219A substitution nearly abolished functionally relevant ATP-binding events at the consensus site.

The conformational dynamics of the W401A and Y1219A variants were markedly different, both from each other, and from those wild-type CFTR$_{FRET}$ (compare Fig. 1e with Fig. 2c,e). Relative to wild-type CFTR, the W401A variant, which is capable of binding and hydrolysing ATP at the consensus site, underwent comparatively rapid transitions between NBD-separated and NBD-dimerized states that more closely resembled the dynamics of pore opening measured in electrophysiological recordings (Fig. 2c). This was predominantly attributed to a specific reduction in the dwell time of the NBD-dimerized state (compare Fig. 1h with

Fig. 2d). By contrast, the Y1219A variant, which binds ATP principally at the degenerate site, slowly transitioned between NBD-dimerized and NBD-separated states (Fig. 2e). Whereas NBD dimerization and channel-open probabilities became more comparable in the W401A variant (Fig. 2c,d), single-channel measurements of the Y1219A variant exhibited only sporadic opening events (Fig. 2e). These findings indicate that NBD dimerization is largely uncoupled from channel gating when ATP binding and hydrolysis is abrogated at the consensus site.

At 3 mM ATP, the dimerization probabilities of the W401A and Y1219A variants were comparable, at about 50% of the wild-type level (Fig. 2f). The channel-open probabilities of the two variants were, however, very different (Fig. 2g). Whereas the W401A variant functioned like wild-type CFTR in this regard, the open probability of the Y1219A variant was nearly zero. These data indicate that ATP binding at either degenerate or consensus sites is sufficient for NBD dimerization. They further support that transitions to NBD-dimerized states do not necessarily precipitate ATP hydrolysis or channel opening and that channel opening largely depends on ATP binding to the consensus site.

These conclusions were further substantiated through assessment of the 'coupling ratio' between open probability and the probability of NBD dimerization (Fig. 2h), which showed that the coupling efficiency was far more sensitive to occupancy of the consensus site by ATP. The coupling ratio of the W401A variant was sixfold greater than that of the Y1219A variant. The extent of coupling between NBD dimerization and channel opening was the greatest for the E1371Q variant, which traps the pre-hydrolytic NBD-dimerized state with both sites occupied by ATP (Fig. 1e).

## NBD dimerization precedes channel opening

To examine the temporal relationship between ATP-dependent NBD dimerization and channel opening, we carried out parallel experiments in which the pre-steady state of smFRET and electrophysiological CFTR reaction coordinates were monitored in response to rapid ATP addition (Fig. 3a). Here we separately tracked the time courses

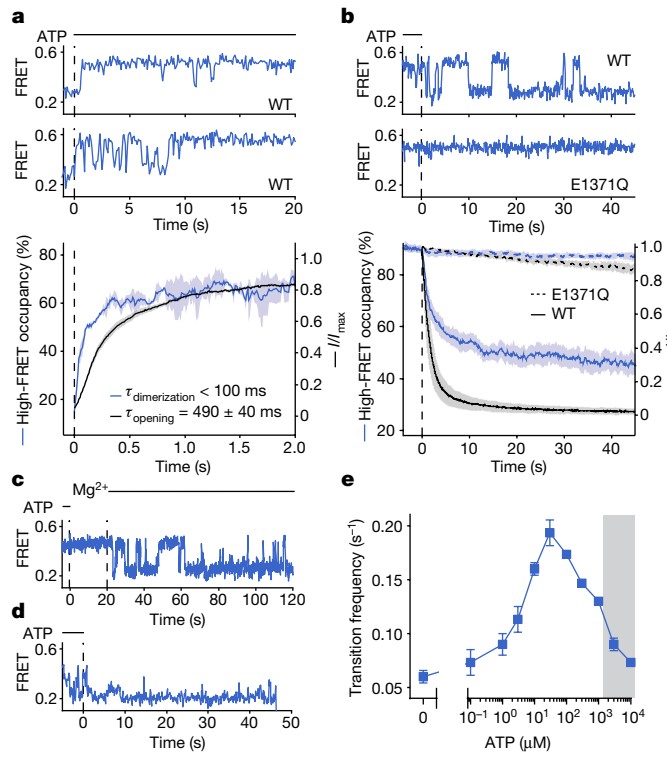

**Fig. 3 | Temporal resolution of NBD conformation from pore state in the pre-steady state. a**, Upper panels, representative smFRET traces of ATP delivery (at the dashed line) to phosphorylated and nucleotide-free wild-type CFTR$_{FRET}$. Lower panel, time-dependent changes in high-FRET occupancy of CFTR$_{FRET}$ and wild-type CFTR current after ATP delivery. Data represent means (solid line) and standard errors (shaded area) of 3 FRET experiments and 42 patches. Individual time courses were fitted as mono-exponential relaxations (see Extended Data Fig. 7a,b). Means and standard errors of exponential time constants are reported. **b**, Upper panels, representative smFRET traces of ATP withdrawal (at the vertical dashed line) from phosphorylated wild-type CFTR$_{FRET}$ and the CFTR$_{FRET}$(E1371Q) variant. Lower panel, time-dependent changes in high-FRET occupancy of CFTR$_{FRET}$ and CFTR current after ATP withdrawal from the wild-type (solid lines) and the E1371Q (dashed lines) variant. Data represent means (line) and standard errors (shaded area) of 5 FRET experiments and 41 (wild-type) or 6 (E1371Q) patches. **c**, Representative single-molecule trace of ATP withdrawal from wild-type CFTR$_{FRET}$. Initially Mg$^{2+}$ is absent, followed by reintroduction of 2 mM Mg$^{2+}$. **d**, Representative single-molecule trace of ATP withdrawal from the CFTR$_{FRET}$(W401A) variant. **e**, ATP dose response for the frequency of transition between low- and high-FRET states for phosphorylated wild-type CFTR$_{FRET}$. Data represent means and standard errors for three experiments. The shaded area indicates the regime in which transitions are obscured by time averaging, resulting in erroneous rate estimates. ATP was used at 3 mM in all panels.

of NBD dimerization and macroscopic current increase on application of saturating ATP (3 mM) to CFTR$_{FRET}$, which had been previously phosphorylated by PKA treatment, followed by complete ATP removal from the system.

In these experiments, individual CFTR$_{FRET}$ molecules transitioned either directly to a stably NBD-dimerized state, or through a highly dynamic interval with rapid NBD-isomerization events (Fig. 3a), resembling the W401A variant at steady state (Fig. 2c). Given that no change in R domain phosphorylation occurs in this experiment, we conclude that the observed heterogeneity in NBD-dimerization kinetics reflects stochastic and sequential ATP binding to the individual NBDs, which ultimately equilibrate to both NBDs being occupied by ATP.

Comparison of the activation time courses of both reaction coordinates further revealed that channel opening on ATP binding was delayed relative to NBD dimerization (Fig. 3a). The rate of channel opening ($\tau_{opening}$ = 490 ± 40 ms; Fig. 3a and Extended Data Fig. 7a) was approximately threefold slower than the solvent exchange rate of the perfusion system ($\tau_{exchange} \approx 150$ ms; Extended Data Fig. 7c,d). By contrast, the fitted rates for NBD dimerization from FRET measurements ($\tau_{dimerization} \approx 100$ ms) were on the same scale as the solvent exchange rate in the fluorescence microscope ($\tau_{exchange}$ = 115 ms; Extended Data Fig. 7b,e). Thus, the observed delay in current activation could not be ascribed to differences in rates of mixing of the two experimental methods. We therefore conclude that the observed delay reflects conformational changes within the NBD-dimerized state that precede channel opening and that the mean first-passage time of this process is approximately 400–500 ms.

## Dimerization persists through cycles of hydrolysis

To understand the molecular events surrounding the process of pore closure, we monitored conformational changes and macroscopic current decays of fully phosphorylated CFTR on sudden ATP withdrawal (Fig. 3b and Extended Data Fig. 7f–l). Consistent with the findings of previous studies[15,27], our observations showed that ATP removal leads to rapid current decay that is dependent on ATP hydrolysis (Extended Data Fig. 7f,g). Parallel FRET experiments showed that the time course of NBD separation is biphasic, with time constants of 1.6 s and 20 s, respectively (Fig. 3b). These rates correlate with the double-exponential time constants reported for CFTR current decay and ligand exchange[18,28]. This apparent correlation suggests a common underlying molecular mechanism determining both transitions.

Inhibiting ATP hydrolysis with the E1371Q substitution markedly slowed NBD separation, and biochemical approaches to reduce ATP hydrolysis or conformational events immediately following hydrolysis—including magnesium withdrawal, as well as beryllium fluoride or aluminium fluoride addition—also resulted in much slower NBD separation (Fig. 3b,c and Extended Data Fig. 7i,j).

On ATP withdrawal, individual CFTR$_{FRET}$ molecules first exhibited dynamic NBD isomerization, followed by stable NBD separation (Fig. 3b). This dynamic period resembled the steady-state behaviour of the Y1219A variant (Fig. 2e). Disruption of ATP binding at the degenerate site by the W401A substitution eliminated this dynamic period, such that transitions occurred directly from the NBD-dimerized state to the NBD-separated state on ATP withdrawal (Fig. 3d and Extended Data Fig. 7h,k). These observations suggest that the dynamic period represents a post-hydrolytic state, in which the consensus site becomes vacated, and the degenerate site retains ATP. As ATP rebinding is not possible in this experiment, subsequent ATP dissociation from the degenerate site then precipitates stable NBD separation (Extended Data Fig. 7l). Dissociation of ATP from both sites probably leads to the reversible rundown of CFTR currents that occurs after prolonged exposure to nucleotide-free solutions[29].

ATP rebinding at physiological ATP concentrations (approximately 1–10 mM) is expected to occur rapidly to the post-hydrolytic CFTR molecule, thereby initiating new catalytic cycles before NBD separation. Consistent with this notion, the transition frequency between the low- and high-FRET states exhibited a bell-shaped dependence on ATP concentration (Fig. 3e and Extended Data Fig. 5f). These findings suggest that the probability of ATP rebinding exceeds that of complete NBD separation when ATP concentration is greater than 100 μM. This concept, consistent with ligand exchange experiments[18], suggests that repetitive cycles of ATP turnover can occur in an ostensibly NBD-dimerized conformation with only subtle changes at the consensus site required for nucleotide exchange. In cellular settings, repetitive gating cycles are therefore expected to persist until the finite rate of NBD separation at cellular ATP concentrations allows the dephosphorylated R domain to reinsert, terminating CFTR gating.

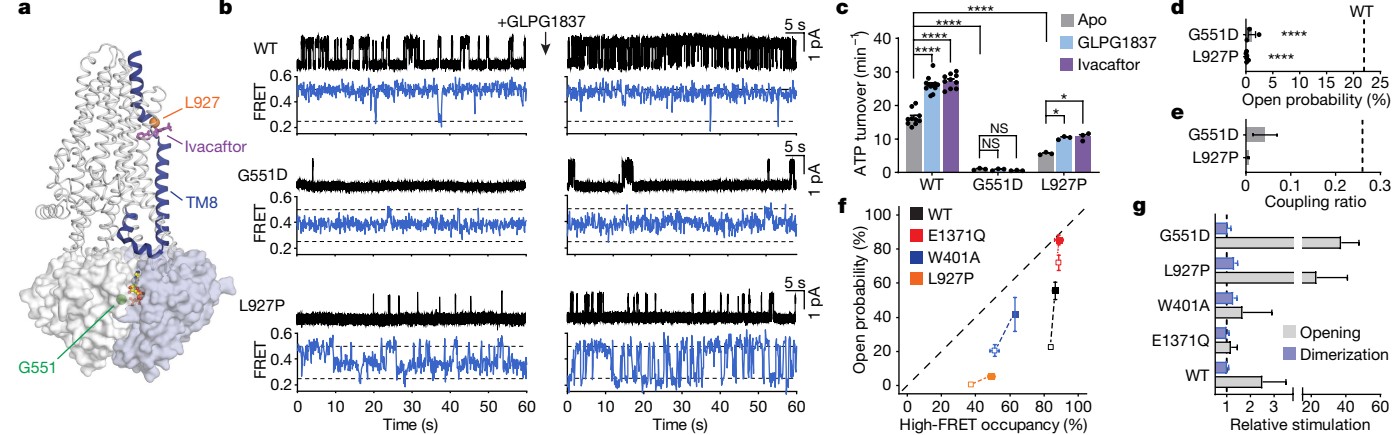

**Fig. 4 | Cystic fibrosis-associated variants and pharmacological potentiation. a**, Cartoon representation of CFTR, indicating the positions of residues G551 and L927, and the ivacaftor-binding site. The structural link between the potentiator-binding site and the NBD interface through TM8 is coloured blue. The consensus-site ATP is shown as sticks. **b**, Sample traces from single-channel electrophysiology (top) and smFRET (bottom) of wild-type CFTR and the CFTR(G551D) and CFTR(L927P) variants in the absence or presence of 10 μM GLPG1837. Substitutions were made in wild-type CFTR and CFTR$_{FRET}$ backgrounds for electrophysiology and smFRET experiments, respectively. Horizontal dashed lines indicate mean FRET efficiencies of low- and high-FRET states. **c**, Steady-state ATP hydrolysis rates for the wild-type CFTR and the CFTR(G551D) and CFTR(L927P) variants. Measurements were carried out in the absence or presence of 10 μM GLPG1837 or 1 μM ivacaftor. Data represent means and standard errors for 10 (wild-type apo and with ivacaftor), 12 (wild-type with GLPG1837) and 3 (G551D and L927P all conditions) measurements. *$P$ = 0.032 (L927P apo versus GLPG1837) and 0.017 (L927P apo versus ivacaftor); ****$P$ = 3 × 10$^{-14}$ (wild-type apo versus GLPG1837), 4 × 10$^{-15}$ (wild-type apo versus ivacaftor), 2 × 10$^{-14}$ (wild-type versus G551D) and 8.8 × 10$^{-10}$ (wild-type versus L927P). **d**, Open probabilities of G551D and L927P variants. Data represent means and standard errors for 3 (G551D) or 6 (L927P)

bilayers. The dashed line indicates mean open probability of wild-type CFTR. ****$P$ = 3.1 × 10$^{-7}$ (G551D) and 4.3 × 10$^{-11}$ (L927P). **e**, Coupling ratios of G551D and L927P variants. Data represent means and standard errors. The dashed line indicates the coupling ratio for wild-type CFTR. **f**, Correlation of probabilities of dimerization and opening for wild-type CFTR, CFTR(E1371Q), CFTR(W401A) and CFTR(L927P) in the absence (open squares) or presence (filled squares) of 10 μM GLPG1837. Data represent means and standard errors for 39 (wild-type apo), 9 (wild-type GLPG1837 and W401A apo), 10 (E1371Q apo), 6 (E1371Q GLPG1837 and L927P apo), 5 (W401A GLPG1837) and 3 (L927P GLPG1837) open-probability measurements and for 8 (wild-type apo), 4 (wild-type GLPG1837, E1371Q apo and E1371Q GLP1837), 5 (W401A apo and L927P apo) and 3 (W401A GLPG1837 and L927P GLPG1837) FRET measurements. The dashed line indicates equality between probabilities of opening and dimerization. **g**, Relative stimulation of opening and dimerization probabilities with 10 μM GLPG1837. Data represent means and standard deviations. The dashed line indicates no stimulation. Relative stimulation of G551D opening was determined by macroscopic current measurements in excised inside-out patches. Phosphorylated CFTR variants at 3 mM ATP were used in all panels. For relevant panels, statistical significance was tested by one-way analysis of variance.

## Disease mutations disrupt allosteric coupling

A wide range of alterations in CFTR are directly linked to cystic fibrosis[30] (Extended Data Fig. 8a). The mechanisms by which such alterations affect an individual's health have been broadly categorized into those that interfere with CFTR expression, folding or localization or the function of the channels on the cell surface (https://www.cftr2.org/mutations_history). Here we examine two clinically evidenced variants that affect channel gating at the cell surface, G551D and L927P, to understand the molecular basis of their defects.

The G551 residue forms part of the consensus ATP-binding site that coordinates the phosphate moieties for hydrolysis[31] (Figs. 2a and 4a). Substituting G551 with an aspartate nearly abolished channel opening[32] and ATP hydrolysis (Fig. 4b–d). On addition of saturating ATP (3 mM) to the phosphorylated CFTR$_{FRET}$(G551D) variant, we observed an upward shift in FRET efficiency from the NBD-separated state (≈0.25 FRET efficiency) to 0.37 ± 0.01. This intermediate FRET efficiency value was clearly distinct from that of the NBD-dimerized conformation (≈0.49 FRET) observed for wild-type CFTR$_{FRET}$, indicative of a conformation involving an intermediate approach of the NBDs (Fig. 4b and Extended Data Fig. 8b,d). From this intermediate conformation, excursions to the 0.49 FRET efficiency states were evidenced, albeit rarely (Fig. 4b). The high-FRET, NBD-dimerized CFTR(G551D) conformation is likely to be different from that of CFTR(E1371Q) previously observed by cryo-EM, evident by a lower coupling ratio (Fig. 4e) and a shorter lifetime (Extended Data Fig. 8e,f). In agreement with these data, the findings of a recent cryo-EM study showed that the G551D variant

adopts conformations in between those of the fully NBD-separated and NBD-dimerized conformations[33].

The L927 residue in CFTR resides within a transmembrane hinge region that mediates local conformational changes during gating[34] (Fig. 4a). Thus, it would be reasonable to propose that L927P causes cystic fibrosis by altering the flexibility of the transmembrane hinge in NBD-dimerized CFTR conformations. Compared to the wild-type CFTR, the L927P substitution resulted in a 65% reduction of the ATPase activity and a >99% reduction in channel-open probability (Fig. 4c,d). Its open dwell time was reduced by approximately 15-fold (Extended Data Fig. 8g,h). Notably, the L927P substitution, although 50 Å away from either ATP-binding site, was also detrimental to the NBD-dimerization process. In the absence of ATP, the fully phosphorylated L927P variant behaved similarly to wild-type CFTR$_{FRET}$ with the NBDs constitutively separated (Extended Data Fig. 8c,d). However, on ATP introduction (3 mM), the L927P variant adopted a conformation exhibiting an intermediate extent of NBD closure (FRET efficiency = 0.31 ± 0.01) from which relatively frequent, although transient, NBD-dimerization events occurred (Fig. 4b and Extended Data Fig. 8c–f).

For both G551D and L927P variants, FRET transitions exhibited ATP dependence indicative of wild-type ATP binding affinities (Extended Data Fig. 8i–k). Their functional defects are caused by deficits in ATP effecting formation of a tight NBD dimer and in the coupling of the allosteric processes within NBD-dimerized CFTR that give rise to channel opening (Fig. 4e). On the basis of the positions of the G551D and L927P substitutions, we posit that an allosteric link transmits conformational information from the consensus ATP-binding site at the NBD dimer

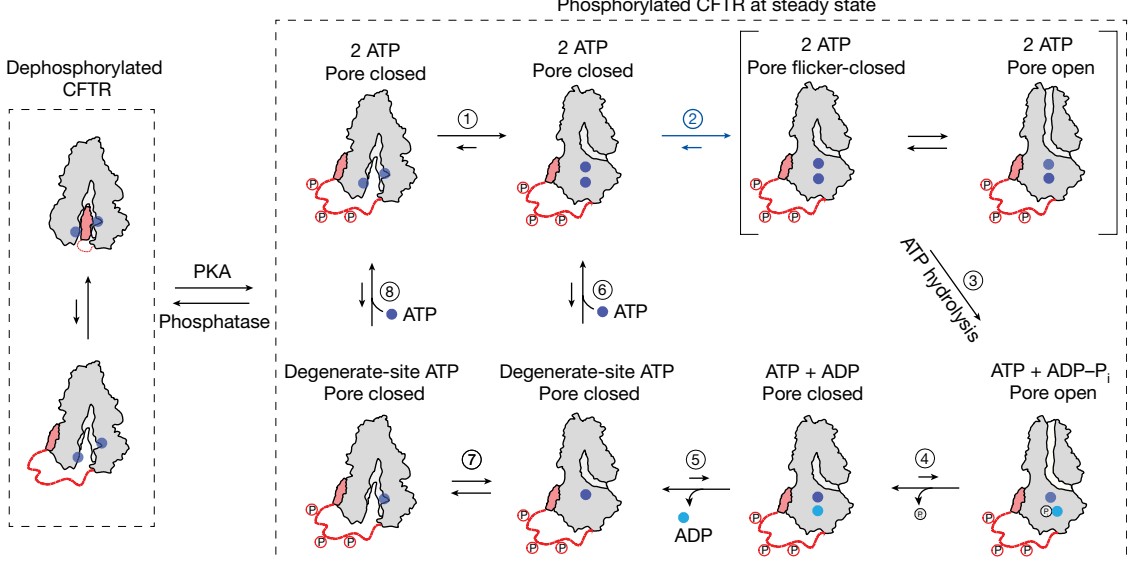

Phosphorylated CFTR at steady state

**Fig. 5 | CFTR gating model.** Model of the wild-type CFTR gating cycle at physiological ATP concentration. Dephosphorylated CFTR adopts an NBD-separated, auto-inhibited conformation. At steady state, the NBDs of fully phosphorylated CFTR dimerize rapidly with ATP bound at both sites (step 1). Dimerization is followed by conformational change to enable pore opening (step 2) and ATP hydrolysis at the consensus site (step 3). During the pre-hydrolytic open burst, CFTR rapidly samples a flicker-closed state. Post-hydrolytic CFTR remains open, but eventually relaxes to a non-conductive dimerized state (step 4). ADP dissociation (step 5) leads to a dynamically isomerizing intermediate (step 7). ATP rebinding may occur with subtle rearrangement at the dimer interface (step 6) or with complete NBD separation (step 8) to initiate a new gating cycle.

interface through transmembrane helix 8 (TM8) to the gating region on the opposing side of the membrane (Fig. 4a). Distant G551D and L927P substitutions both impact this pathway of allosteric communication, indicating that local disruptions at either end of this link can affect both NBD dimerization and channel opening.

## Potentiators promote opening of dimerized channels

Ivacaftor, a drug approved by the US Food and Drug Administration, and the investigational compound GLPG1837 (ref. [35,36]) both bind to CFTR within the TM8 hinge region to promote channel opening[37] (Fig. 4a). Although the effects of these compounds on gating kinetics have been extensively characterized[38–41], how they alter the conformational landscape of CFTR remains elusive.

Consistent with the findings of previous reports[38–41], our observations showed that both potentiators induced marked increases of channel-open probabilities (Fig. 4b,f and Extended Data Fig. 9a). By comparison their effects on NBD dimerization were much smaller for all CFTR variants tested (Fig. 4b,f and Extended Data Fig. 9a). For example, GLPG1837 increased the open probability of the G551D variant by more than 30-fold, whereas the change in NBD dimerization was marginal (Fig. 4b,g). This observation, together with the recent cryo-EM study of the CFTR(G551D) variant in the presence of ivacaftor[33], demonstrates that neither ivacaftor nor GLPG1837 promotes NBD dimerization. Similarly, for the L927P variant, the relative stimulation of open probability greatly exceeded the relative stimulation of dimerization probability (Fig. 4f,g).

The potency with which GLPG1837 promoted NBD dimerization, measured at the $EC_{50}$ for ATP, was approximately 60 nM (Extended Data Fig. 9b,c), similar to the estimated affinity from electrophysiology[37]. The apparent potency of ATP to mediate dimerization also increased in a dose-dependent manner with GLPG1837 (Extended Data Fig. 9d). Furthermore, the rate of NBD separation on ATP withdrawal was slowed by GLPG1837 or ivacaftor (Extended Data Fig. 9e), analogous to their impacts on the rate of current relaxation after ATP withdrawal[38].

Potentiators both shorten the closed dwell time and extend the open dwell time of the pore[39]. Here we show that the steady-state ATP

hydrolysis rates of wild-type CFTR and CFTR(L927P), for which hydrolysis rates were measurable, were increased by 60–100% by ivacaftor or GLPG1837 (Fig. 4c). Hence, both potentiators exert, by opposing effects on the open and closed dwell times, a net increased flux through the gating cycle by targeting the allosteric pathway linking the NBDs to the channel gate.

These data lead to the conclusion that the main effect of ivacaftor or GLPG1837 is not to support transition from NBD-separated to NBD-dimerized conformations. Rather, these potentiators principally operate by promoting pore opening when NBDs are already dimerized. In other words, potentiators affect the coupling efficiency between NBD dimerization and channel opening, possibly by stabilizing the transmembrane domains in the pore-open configuration[41]. This effect also manifests in variants unable to form a canonical NBD dimer, such as G551D and a variant devoid of the entire NBD2 (ref. [38]).

## Discussion

It has long been debated whether NBD dimerization in CFTR is strictly coupled to pore opening[5,6]. By directly comparing the kinetics of NBD isomerization and channel gating, we show that NBD dimerization and ion permeation are not strictly coupled but instead probabilistically linked through allosteric control mechanisms. At physiological ATP concentrations, fully phosphorylated CFTR remains NBD-dimerized for many cycles of ATP turnover and pore opening. The structure of the NBD-dimerized CFTR[14] suggests that only small changes at the consensus site, such as disrupting the hydrogen bond between R555 and T1246 (ref. [15]), would be sufficient for nucleotide exchange. Notably, the allosteric relationship evidenced between NBD dimerization and pore opening held true across diverse conditions and CFTR variants and was sensitive to both nucleotide state in the consensus site and potentiator binding within the membrane more than 50 Å away.

These findings reveal an allosteric pathway linking the consensus ATPase site, through TM8 and the potentiator-binding site, to the gate of the pore on the opposing membrane surface. Structurally, we speculate that this pathway of long-distance information transfer minimally consists of TM8 and TM9 and the transverse alpha helix

between them (Fig. 4a). The transmission of structural information along this allosteric pathway physically linking NBD dimerization to pore opening is rate-limiting to CFTR function. Substitutions causing cystic fibrosis (for example, G551D and L927P) attenuate the strength of this allosteric pathway whereas the potentiators ivacaftor and GLPG1837 enhance it. The observation that both G551D and L927P variants are also defective in NBD dimerization suggests that modulators that quantitatively rescue this defect should work additively with ivacaftor and GLPG1837. The investigational compound 5-nitro-2-(3-phenylpropylamino) benzoate was proposed to stimulate pore opening by such a mechanism[42].

The data presented herein, in conjunction with the vast body of literature in the field, permit us to propose a model that describes the main events accompanying the wild-type CFTR gating cycle at physiological ATP concentrations (Fig. 5). Dephosphorylated CFTR adopts an NBD-separated, auto-inhibited conformation as observed by cryo-EM[13]. Following phosphorylation of the R domain, the NBDs can dimerize rapidly with ATP bound at both sites (step 1 in Fig. 5). Rate-limiting conformational changes within CFTR that allosterically communicate information from the consensus ATP-binding site across the lipid bilayer can subsequently open the pore (step 2 in Fig. 5) and enable ATP hydrolysis (step 3 in Fig. 5). Before ATP hydrolysis, pore opening is transient, and flicker-closed states are rapidly sampled[43]. The post-hydrolytic channel, with ADP and inorganic phosphate bound at the consensus site, remains open but eventually relaxes to a non-conductive dimerized state (step 4 in Fig. 5). Dissociation of ADP (step 5 in Fig. 5) results in a dynamic intermediate to which ATP can rebind (steps 6–8 in Fig. 5) thereby initiating another gating cycle. Rare events are not depicted in this model as their fractional contributions are expected to be low at physiological ATP concentration. These events include release of ATP from the degenerate site[25,26], and channel opening with ATP at only one site[32,44,45] or in the complete absence of nucleotide[11,12,32,46].

The topology of this scheme was validated through steady-state kinetic simulations of NBD dynamics and ion conduction for fully phosphorylated CFTR at saturating ATP concentrations (Extended Data Fig. 10 and Supplementary Methods). Kinetic constants within this topology were estimated on the basis of the model's capacity to recapitulate experimental observables including pre-steady-state rates of NBD dimerization, channel current and conformational relaxation as well as ATP hydrolysis rates from ensemble measurements (Extended Data Fig. 10a). Stochastic simulations for wild-type CFTR and CFTR(E1371Q) gating carried out with this topology and rate information (Extended Data Fig. 10b,c) closely recapitulated the key dynamical features of both wild-type and E1371Q variant gating (Extended Data Fig. 10d–h and Supplementary Video 1). However, establishment of this model revealed notable quantitative discrepancies that are worthy of consideration. First, our simulation predicts a modestly greater steady-state ATP hydrolysis rate than is experimentally estimated. We speculate that this may reflect either inadequacies of the simplified model or the presence of an inactive fraction in the bulk measurement that lowers the apparent turnover rate. Second, the simulated model does not recapitulate the multimodality of NBD-dimerized and NBD-separated dwell-time distributions. Such findings, viewed in light of our analyses of altered degenerate and consensus binding sites, suggest that these distinct modes probably reflect periods of NBD dynamics and gating in CFTR in which only one of the two ATP-binding sites is occupied. Such considerations imply requirements for additional complexities to the presented model topology that will need to be explored by developing techniques that simultaneously detect conformational state and functional output at single-molecule resolution. The present investigations nonetheless reveal a physical framework for understanding CFTR function, pathology and pharmacology and exploring the possibility that more potent activators of the rate-limiting allosteric events regulating CFTR gating mechanism, potentially specific to an individual's allelic variation, can be identified and leveraged for therapeutic purposes.

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

# Methods

## Protein expression, purification and labelling

CFTR was expressed as previously described[47]. Human CFTR with a C-terminal PreScission Protease-cleavable GFP tag was cloned into the BacMam vector. For single-molecule FRET, the following substitutions were introduced: C76L, C128S, C225S, C276S, C343S, T388C, C491S, C592M, C647S, C832S, C866S, C1344S, C1355S, C1395S, C1400S, C1410S, S1435C and C1458S. A deca-His tag was inserted C-terminally to CFTR and before the PreScission Protease cleavage site to allow for surface immobilization.

Recombinant baculovirus was generated using Sf9 cells (Gibco, catalogue number 11496015, lot number 1670337) cultured in sf-900 SFM medium (Gibco), supplemented with 5% (v/v) heat-inactivated fetal bovine serum and 1% (v/v) antibiotic–antimycotic (Gibco) as described previously[48]. HEK293S GnTI⁻ (ATCC CRL-3022, lot number 62430067) suspension cells were cultured in FreeStyle 293 medium (Gibco) supplemented with 2% (v/v) heat-inactivated fetal bovine serum and 1% (v/v) antibiotic–antimycotic (Gibco), shaking at 37 °C with 8% $CO_2$ and 80% humidity. Sf9 and HEK293S GnTI⁻ cells were authenticated by Gibco and ATCC, respectively and confirmed negative for mycoplasma contamination. At a density of $2.5 \times 10^6$ cells ml⁻¹, cells were infected with 10% (v/v) P3 baculovirus. After 12 h, the culture was supplemented with 10 mM sodium butyrate, and the temperature was reduced to 30 °C. After a further 48 h, the cells were collected and flash-frozen in liquid nitrogen.

For protein purification, cells were solubilized for 75 min at 4 °C in extraction buffer containing 1.25% (w/v) lauryl maltose neopentyl glycol (LMNG), 0.25% (w/v) cholesteryl hemisuccinate (CHS), 200 mM NaCl, 20 mM HEPES (pH 7.2 with NaOH), 2 mM $MgCl_2$, 10 µM dithiothreitol (DTT), 20% (v/v) glycerol, 1 mM ATP, 1 µg ml⁻¹ pepstatin A, 1 µg ml⁻¹ leupeptin, 1 µg ml⁻¹ aprotinin, 100 µg ml⁻¹ soy trypsin inhibitor, 1 mM benzamidine, 1 mM phenylmethylsulfonyl fluoride (PMSF) and 3 µg ml⁻¹ DNase I. Lysate was clarified by centrifugation at 75,000g for $2 \times 20$ min at 4 °C, and mixed with NHS-activated Sepharose 4 Fast Flow resin (GE Healthcare) conjugated with GFP nanobody, which had been pre-equilibrated in 20 column volumes of extraction buffer. After 1 h, the resin was packed into a chromatography column, washed with 20 column volumes of wash buffer containing 0.06% (w/v) digitonin, 200 mM NaCl, 20 mM HEPES (pH 6.8 with NaOH), 1 mM ATP and 2 mM $MgCl_2$, and then incubated for 2 h at 4 °C with 0.35 mg ml⁻¹ PreScission Protease to cleave off the GFP tag. The eluate was collected by dripping through Glutathione Sepharose 4B resin (Cytiva) to remove PreScission Protease, and CFTR was concentrated to 2 µM.

To label with fluorophores, CFTR was mixed with 9.5 µM maleimide-conjugated LD555 and 10.5 µM maleimide-conjugated LD655 (Lumidyne Technologies) for 10 min at 4 °C. Subsequent steps were carried out protected from light. The labelling reaction was quenched by addition of 2 mM DTT, and the labelled product was purified by gel filtration chromatography at 4 °C using a Superose 6 10/300 GL column (GE Healthcare), equilibrated with 0.06% (w/v) digitonin, 200 mM NaCl, 20 mM HEPES (pH 7.2 with NaOH), 1 mM ATP and 2 mM $MgCl_2$. Peak fractions were concentrated to 2 µM and mixed with 5 µM biotin–tris-NTA-Ni²⁺ for 30 min at 4 °C. The CFTR–Ni-NTA complex was purified by another round of gel filtration, concentrated to 2 µM, aliquoted, snap-frozen in liquid nitrogen, and stored at −80 °C.

For ATP hydrolysis measurements, the purification protocol was adjusted: extraction buffer contained 1.25% (w/v) LMNG, 0.25% (w/v) CHS, 200 mM KCl, 20 mM HEPES (pH 8.0 with KOH), 2 mM $MgCl_2$, 2 mM DTT, 20% (v/v) glycerol, 1 µg ml⁻¹ pepstatin A, 1 µg ml⁻¹ leupeptin, 1 µg ml⁻¹ aprotinin, 100 µg ml⁻¹ soy trypsin inhibitor, 1 mM benzamidine, 1 mM PMSF and 3 µg ml⁻¹ DNase I. Wash and gel filtration buffers contained 0.06% (w/v) digitonin, 20 mM HEPES (pH 8.0 with KOH), 200 mM KCl, 2 mM $MgCl_2$ and 2 mM DTT. The eluate from the GFP nanobody resin was concentrated, phosphorylated with PKA (NEB) for 1 h at 25 °C,

purified by gel filtration chromatography, and immediately used for hydrolysis measurements.

For proteoliposome reconstitution, the purification was also adjusted: extraction buffer contained 1.25% (w/v) LMNG, 0.25% (w/v) CHS, 200 mM NaCl, 20 mM HEPES (pH 7.2 with NaOH), 2 mM $MgCl_2$, 2 mM DTT, 20% (v/v) glycerol, 1 µg ml⁻¹ pepstatin A, 1 µg ml⁻¹ leupeptin, 1 µg ml⁻¹ aprotinin, 100 µg ml⁻¹ soy trypsin inhibitor, 1 mM benzamidine, 1 mM PMSF and 3 µg ml⁻¹ DNase I. Wash and gel filtration buffers contained 0.006% (w/v) glyco-diosgenin (GDN), 200 mM NaCl, 20 mM HEPES (pH 7.2 with NaOH) and 2 mM $MgCl_2$. The eluate from the GFP nanobody resin was concentrated, phosphorylated with PKA (NEB) for 1 h at 25 °C, purified by gel filtration chromatography, and immediately reconstituted.

## ATP hydrolysis measurements

Steady-state ATP hydrolysis activity was measured using an NADH-coupled assay[49]. Reaction buffer contained 50 mM HEPES (pH 8.0 with KOH), 150 mM KCl, 2 mM $MgCl_2$, 2 mM DTT, 0.06% (w/v) digitonin, 60 µg ml⁻¹ pyruvate kinase (Roche), 32 µg ml⁻¹ lactate dehydrogenase (Roche), 9 mM phosphoenolpyruvate and 150 µM NADH, and was prepared immediately before starting the assay. A 200 nM concentration of phosphorylated CFTR was diluted into reaction buffer. Aliquots of 30 µl in volume were distributed into a Corning 384-well Black/Clear Flat Bottom Polystyrene NBS Microplate. Samples were kept at 4 °C and light-protected until the reactions were initiated by addition of 3 mM ATP. The rate of fluorescence depletion was monitored at $\lambda_{ex} = 340$ nm and $\lambda_{em} = 445$ nm at 28 °C with an Infinite M1000 microplate reader (Tecan), and converted to ATP turnover with an NADH standard curve.

## Patch-clamp recording

Chinese hamster ovary cells (ATCC CCL-61, lot number 70014310) were maintained in DMEM-F12 (ATCC) supplemented with 10% (v/v) heat-inactivated fetal bovine serum and 1% (v/v) GlutaMAX (Gibco) at 37 °C. Chinese hamster ovary cells were authenticated by ATCC. The cells were plated in 35-mm cell culture dishes (Falcon) 24 h before transfection. Cells were transfected with C-terminally GFP-fused CFTR cloned into the BacMam expression vector, using Lipofectamine 3000 according to the manufacturer's protocol (Invitrogen). At 12 h following transfection, medium was replaced with DMEM-F12 supplemented with 2% (v/v) heat-inactivated fetal bovine serum and 1% (v/v) GlutaMAX, and the cells were then incubated for 24 h at 30 °C before recording.

Bath solution contained 145 mM NaCl, 2 mM $MgCl_2$, 5 mM KCl, 1 mM $CaCl_2$, 5 mM glucose, 5 mM HEPES and 20 mM sucrose (pH 7.4 with NaOH). Pipette solution contained 140 mM NMDG, 5 mM $CaCl_2$, 2 mM $MgCl_2$ and 10 mM HEPES (pH 7.4 with HCl). Perfusion solution contained 150 mM NMDG, 2 mM $MgCl_2$, 1 mM $CaCl_2$, 10 mM EGTA and 8 mM Tris (pH 7.4 with HCl). Magnesium was omitted where indicated. CFTR was activated by exposure to PKA (Sigma-Aldrich) and 3 mM ATP.

The rate of buffer exchange by the perfusion system was estimated by exchanging perfusion solution with 150 mM NMDG, 2 mM $MgSO_4$, 1 mM calcium gluconate, 10 mM EGTA and 8 mM Tris (pH 7.4 with $H_2SO_4$).

Pipettes were pulled from borosilicate glass (outer diameter 1.5 mm, inner diameter 0.86 mm, Sutter) to 1.5–2.5 MΩ resistance and fire polished. Recordings were carried out using the inside-out patch configuration with local perfusion at the patch. Membrane potential was clamped at −30 mV. Currents were recorded at 25 °C using an Axopatch 200B amplifier, a Digidata 1550 digitizer and the pClamp software suite (Molecular Devices). Recordings were low-pass-filtered at 1 kHz and digitized at 20 kHz. All displayed recordings were further low-pass filtered at 100 Hz. Data were analysed with Clampfit, GraphPad Prism and OriginPro.

## Proteoliposome reconstitution

A lipid mixture containing 1,2-dioleoyl-*sn*-glycero-3-phosphoetanolamine, 1-palmitoyl-2-oleyl-*sn*-glycero-3-phosphocholine and 1-palmitoyl-2-oleoyl-*sn*-glycero-3-phospho-ʟ-serine at a 2:1:1 (w/w/w) ratio was

resuspended by sonication in buffer containing 200 mM NaCl, 20 mM HEPES (pH 7.2 with NaOH) and 2 mM $MgCl_2$. Lipids were mixed with GDN to a final detergent concentration of 2% (w/v), and lipid concentration of 20 mg ml$^{-1}$ for 1 h at 25 °C covered by argon gas. Purified CFTR was mixed with the lipid mixture at a protein-to-lipid ratio of 1:100 or 1:250 (w/w) and incubated at 4 °C for 2 h covered by argon gas. Methylated beta-cyclodextrin was added to the reaction at a 1.2× molar ratio to GDN. After an additional 4 h, an equivalent amount of methylated beta-cyclodextrin was added. This procedure was repeated for a total of four additions. Proteoliposomes were collected by centrifugation at 150,000g for 45 min at 4 °C, resuspended in buffer containing 200 mM NaCl, 20 mM HEPES (pH 7.2 with NaOH) and 2 mM $MgCl_2$, aliquoted, snap-frozen in liquid nitrogen and stored at −80 °C.

## Planar lipid bilayer recording
Synthetic planar lipid bilayers were made by painting a 1,2-dioleoyl-*sn*-glycero-3-phosphoetanolamine, 1-palmitoyl-2-oleyl-*sn*-glycero-3-phosphocholine and 1-palmitoyl-2-oleoyl-*sn*-glycero-L-serine 2:1:1 (w/w/w) lipid mixture solubilized in decane across an approximately 100-μm-diameter hole on a plastic transparency. CFTR-containing proteoliposomes were phosphorylated with PKA (NEB) for 1 h at 25 °C, and then fused with the synthetic bilayers. Currents were recorded at 25 °C in symmetric buffer containing 150 mM NaCl, 2 mM $MgCl_2$ and 20 mM HEPES (pH 7.2 with NaOH), supplemented with ATP as indicated. Unless otherwise indicated voltage was clamped at 150 mV with an Axopatch 200B amplifier (Molecular Devices). Currents were low-pass filtered at 1 kHz, digitized at 20 kHz with a Digidata 1440A digitizer and recorded using the pCLAMP software suite (Molecular Devices). All displayed recordings were further low-pass filtered at 100 Hz. Data were analysed with Clampfit, GraphPad Prism and OriginPro.

## Single-molecule fluorescence imaging
Imaging was carried out as outlined in ref. [50]. PEG- and biotin–PEG-passivated microfluidic chambers were incubated for 5 min with 0.8 μM streptavidin (Invitrogen) in buffer containing 0.06% (w/v) digitonin, 150 mM NaCl, 2 mM $MgCl_2$ and 20 mM HEPES (pH 7.2 with NaOH). CFTR was either dephosphorylated by Lambda protein phosphatase (λ, NEB) or phosphorylated by PKA (Sigma-Aldrich) before immobilization. Fluorophore-conjugated and biotin–tris-NTA-$Ni^{2+}$-bound CFTR at 200 pM concentration was immobilized within the microfluidic chambers for 1 min, and unbound CFTR was cleared from the channel by washing with buffer. Imaging was carried out in deoxygenated imaging buffer containing 0.06% (w/v) digitonin, 150 mM NaCl, 2 mM $MgCl_2$, 20 mM HEPES (pH 7.2 with NaOH), 2 mM protocatechuic acid and 50 nM protocatechuate-3,4-dioxygenase to minimize photobleaching[51]. $MgCl_2$ was omitted where indicated. Microfluidic chambers were reused several times in the same day by dissociating the immobilized protein with 300 mM imidazole. Experiments were carried out at 25 °C.

For imaging of proteoliposome-reconstituted CFTR, vesicles containing fluorophore-labelled CFTR were extruded through 400-nm and then 100-nm polycarbonate filters (Whatman). The vesicles were then incubated with 1 μM biotin–tris-NTA-$Ni^{2+}$. Excess biotin–tris-NTA-$Ni^{2+}$ was removed by pelleting the vesicles by ultracentrifugation at 150,000g for 45 min, removing the supernatant and resuspending in buffer containing 150 mM NaCl, 2 mM $MgCl_2$ and 20 mM HEPES (pH 7.2 with NaOH). The procedure was repeated twice. Vesicles were immobilized within the microfluidic chambers for 5 min, and unbound vesicles were cleared from the channel by washing with buffer. Imaging was carried out in deoxygenated imaging buffer containing 150 mM NaCl, 2 mM $MgCl_2$, 20 mM HEPES (pH 7.2 with NaOH), 2 mM protocatechuic acid and 50 nM protocatechuate-3,4-dioxygenase.

Single-molecule imaging was carried out using a custom-built wide-field, prism-based total internal reflection fluorescence microscope. LD555 fluorophores were excited with an evanescent wave generated using a 532-nm laser (Opus, Laser Quantum). Emitted fluorescence from LD555 and LD655 was collected with a 1.27 NA 60× water-immersion objective (Nikon), spectrally separated using a T635lpxr dichroic (Chroma), and imaged onto two Fusion sCMOS cameras (Hamamatsu) with integration periods of 10 or 100 ms.

## Single-molecule FRET data analysis
Single-molecule fluorescence data were analysed using SPARTAN analysis software in MATLAB[21]. FRET trajectories were calculated from the emitted donor and acceptor fluorescence intensities ($I_D$ and $I_A$, respectively) as $E_{FRET} = I_A/(I_A + I_D)$. FRET trajectories were selected for further analysis on the basis of the following criteria: single-step donor photobleaching; a signal-to-noise ratio >8; fewer than 4 donor-blinking events; and FRET efficiency above baseline for at least 50 frames. Further, single-molecule traces exhibiting FRET values above 0.8 were excluded from analysis. This subpopulation was insensitive to phosphorylation and nucleotide, and probably reflected denatured molecules. For kinetic analysis, traces were also manually curated to remove obvious photophysical artefacts. FRET trajectories were idealized using the segmental *k*-means algorithm[52] with a model containing two non-zero-FRET states with FRET values of 0.25 ± 0.1 and 0.48 ± 0.1. Data were further analysed with GraphPad Prism and OriginPro.

## Electron microscopy data acquisition and processing
Dephosphorylated wild-type CFTR directly from gel filtration was concentrated to 5.5 mg ml$^{-1}$. Concentrations of 3 mM ATP and 3 mM fluorinated Fos-choline-8 were added to the sample immediately before application onto Quantifoil R1.2/1.3 400 mesh Au grids and then vitrification using a Vitrobot Mark IV (FEI).

Cryo-EM images were collected with a 300-keV Titan Krios transmission electron microscope equipped with a Gatan K2 Summit detector using SerialEM[53]. A total of 3,501 micrographs were collected in superresolution mode with a nominal defocus range of 0.8–2.5 μm. Micrographs had a physical pixel size of 1.03 Å (0.515 Å superresolution pixel size). Micrographs were recorded with 10-s exposure (0.2 s per frame) with a dose rate of 8 electrons per pixel per second.

Image stacks were gain-normalized, binned by 2, and corrected for beam-induced specimen motion with MotionCor2 (ref. [54]). Contrast transfer function estimation was carried out using GCTF[55]. Images with estimated resolutions below 4.5 Å were removed. Particles were initially picked with the Laplacian-of-Gaussian implementation in RELION[56]. Selected two-dimensional classes from this particle set were then used for template-based particle picking. The 710,322 picked particles were cleaned by several rounds of two- and three-dimensional classification. A total of 157,629 particles were included in the final refined map.

## Reporting summary
Further information on research design is available in the Nature Portfolio Reporting Summary linked to this article.

## Data availability
The cryo-EM map has been deposited in the Electron Microscopy Data Bank under the accession code EMD-29637. The corresponding atomic model has been deposited in the Protein Data Bank under accession code 8FZQ. The data that support the findings of this study are available from the authors upon reasonable request.

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

**Acknowledgements** We acknowledge the support from the Single-Molecule Imaging Center at St. Jude Children's Research Hospital and the Evelyn Gruss Lipper Cryo-EM Resource Center of The Rockefeller University. We thank L. Csanády and I. Chen for their comments on the manuscript. This work was financially supported by HHMI (to J.C.) and the National Institutes of Health (GM079238 to S.C.B.).

**Author contributions** J.L. prepared all CFTR samples, carried out electrophysiology experiments, ATP turnover assays and cryo-EM experiments, and analysed the data. J.L. carried out the single-molecule FRET experiments and analysed the data, with assistance from D.S.T. and G.F. Z.K. carried out kinetic modelling and simulations with assistance from all other authors. All authors contributed to writing the manuscript. S.C.B. and J.C. oversaw the project.

**Competing interests** S.C.B. has an equity interest in Lumidyne Technologies. The other authors declare no competing interests.

**Additional information**
**Correspondence and requests for materials** should be addressed to Scott C. Blanchard or Jue Chen.

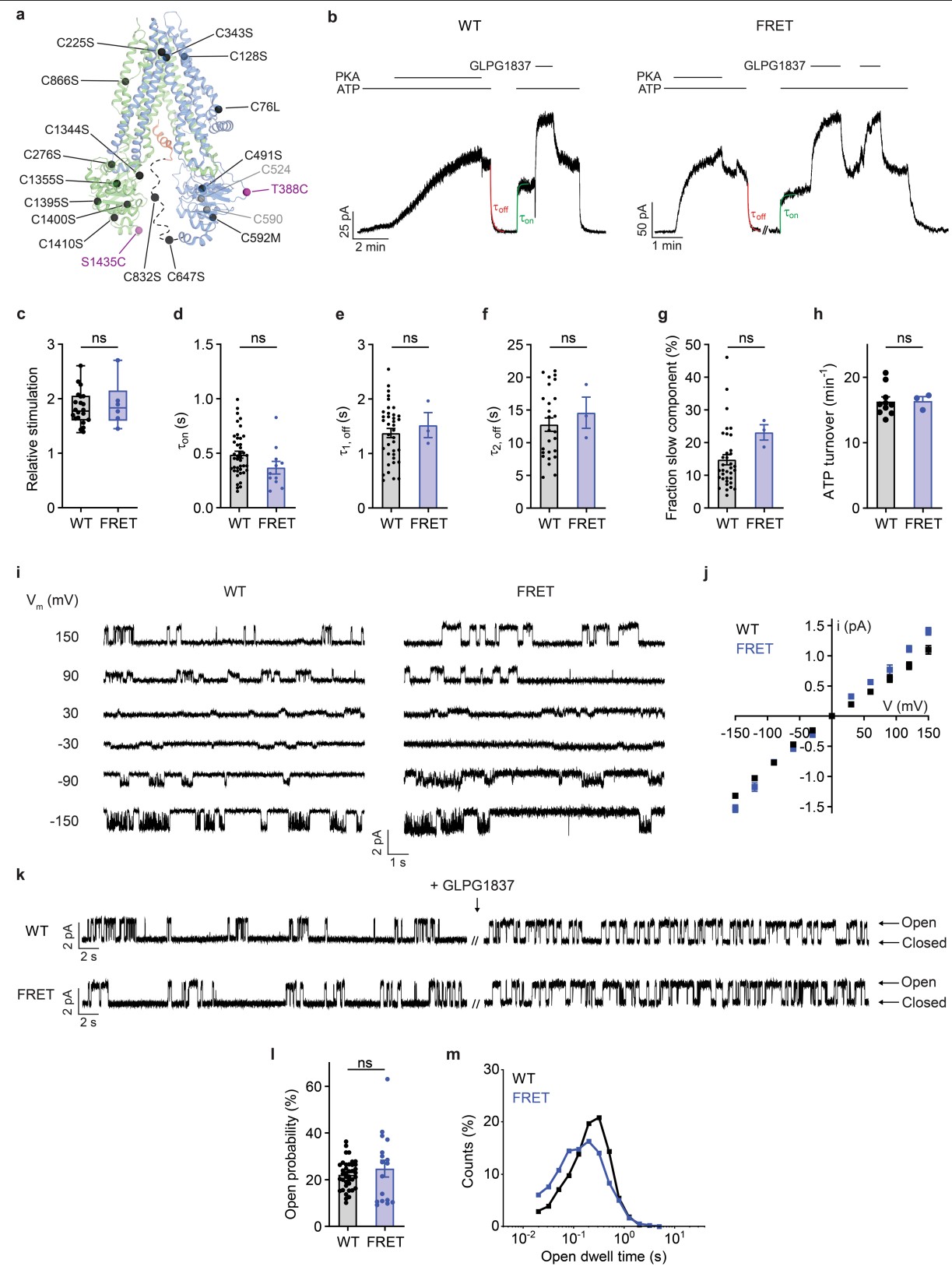

**Extended Data Fig. 1** | See next page for caption.

**Extended Data Fig. 1 | Functional characterization of CFTR$_{FRET}$. a**. Design of FRET variant CFTR. The positions of native substituted, native retained, and novel cysteines are indicated with black, grey, and magenta spheres, respectively. The dashed line represents the structurally unresolved part of the R-domain. The C1458S substitution in the disordered C-terminus is not annotated. **b**. Example recordings showing Protein Kinase A (PKA)-activated, ATP-dependent, and GLPG1837-stimulated current from C-terminally GFP-fused wild-type and FRET variant CFTR in inside-out excised patches. 3 mM ATP, 300 nM PKA, and 10 μM GLPG1837 were used. **c**. Relative GLPG1837-mediated stimulation of wild-type and FRET variant CFTR currents in excised inside-out patches. Whiskers represent minima and maxima and boxes represent 25$^{th}$, 50$^{th}$, and 75$^{th}$ percentiles for 21 (wild-type) or 6 (FRET variant) patches. **d**. Single exponential time-constants of current activation for wild-type and FRET variant CFTR after application of 3 mM ATP. Data represent means and standard errors for 42 (wild-type) or 11 (FRET variant) patches. **e-g**. Exponential time-constants of the fast (**e**) and slow (**f**) components of current relaxation after ATP withdrawal, and the relative weight of the slow component (**g**), for wild-type and FRET variant CFTR. Data represent means and standard errors for 38 (wild-type) or 3 (FRET variant) patches. **h**. Steady state ATPase activity of wild-type and fluorophore-labelled FRET variant CFTR determined from bulk experiments. Data points represent means and standard errors for 10 (wild-type) or 3 (FRET variant) measurements. **i**. Voltage families of individual wild-type (left) and fluorophore-labelled FRET variant (right) CFTR channels reconstituted in synthetic lipid bilayers. The membrane potential (V$_m$) is indicated using physiological convention. **j**. Current-voltage relationships of wild-type and fluorophore-labelled FRET variant CFTR. Data points represent means and standard errors for 3 (wild-type) or 7-18 (FRET variant) channels. **k**. Example recordings of wild-type and fluorophore-labelled FRET variant CFTR reconstituted in synthetic lipid bilayers. CFTR was phosphorylated with PKA prior to fusion with the bilayer. The recording was performed with 3 mM ATP before (left) and after (right) addition of 10 μM GLPG1837. **l**. Open probabilities of the phosphorylated wild-type and fluorophore-labelled FRET variant CFTR in 3 mM ATP. Data points represent means and standard errors for 39 (wild-type) or 17 (FRET variant) channels. **m**. Open dwell time distributions for the phosphorylated wild-type and fluorophore-labelled FRET variant CFTR in 3 mM ATP. Statistical significance was tested using two-tailed Student's t-test (ns: not significant).

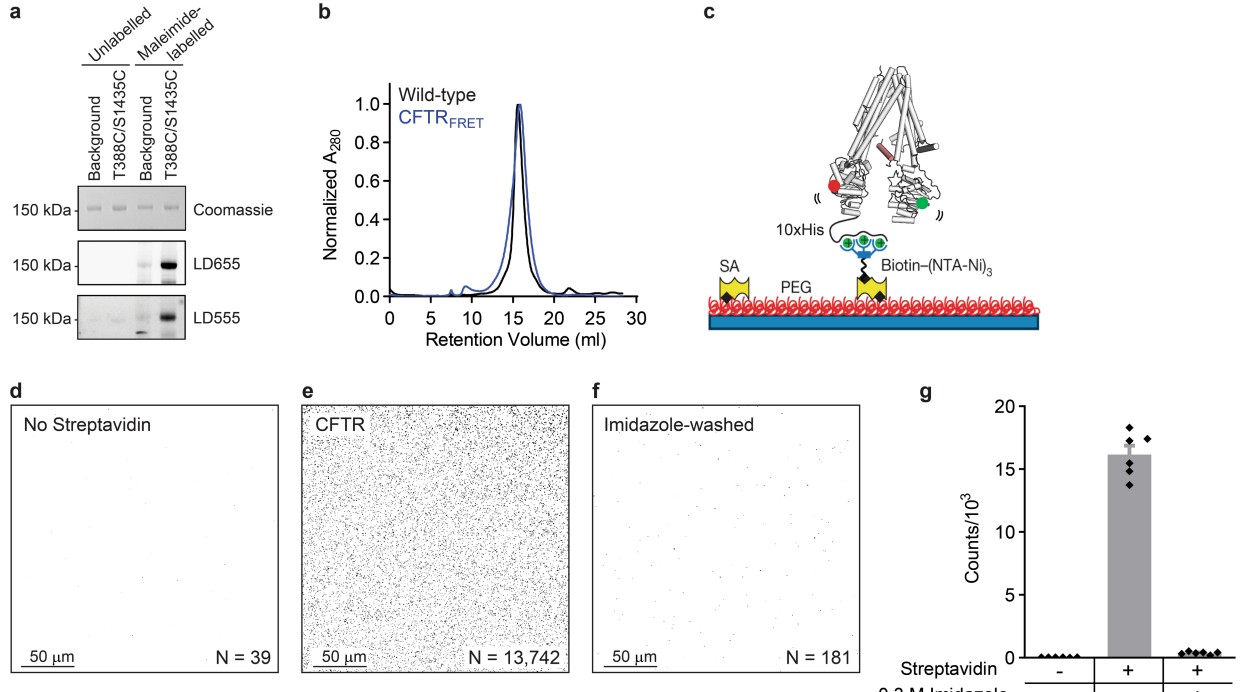

**Extended Data Fig. 2 | Site-specific labelling and surface-immobilization of CFTR. a**. Site-specificity of fluorophore-conjugation to novel cysteines. Cysteine-less CFTR, with or without T388C/S1435C substitutions, was incubated with maleimide-conjugated LD555 and LD655 fluorophores. The products were separated by SDS-PAGE. The gel was imaged for LD555 and LD655 fluorescence and then Coomassie-stained. Labeling was >90 % specific to the two introduced cysteines. The experiment was repeated twice independently with similar results. For gel source data, see Supplementary Fig. 1. **b**. Gel-filtration profiles of wild-type CFTR and fluorophore-labelled CFTR_FRET. **c**. Schematic drawing of the immobilization strategy. **d**–**f**. Wide-field fluorescence images of CFTR_FRET immobilized to a Streptavidin-free (**d**) or Streptavidin-coated (**e**) surface. (**f**) CFTR_FRET immobilized to a Streptavidin-coated surface was washed off with 0.3 M imidazole. N denotes the number of immobilized molecules detected. **g**. Quantification of the specificity of Streptavidin- and His-tag-dependent immobilization. Data points represent the means and standard errors of six immobilizations. The representation of the immobilization surface in **c** was recreated from ref. [57].

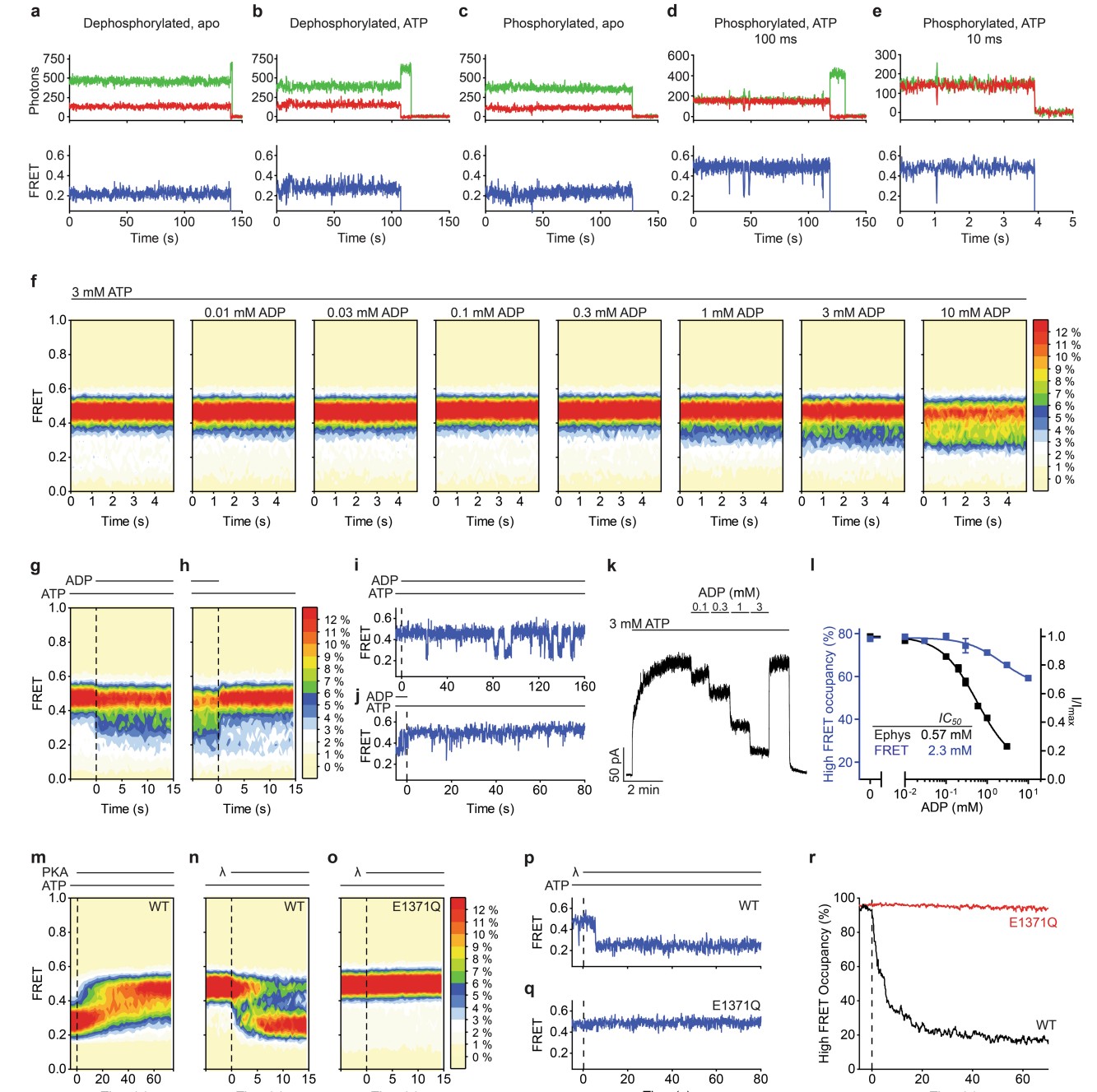

**Extended Data Fig. 3 | Phosphorylation- and nucleotide-dependence of CFTR dimerization. a–e.** Representative single-molecule traces showing donor (green) and acceptor (red) fluorescence intensities (top) and FRET (bottom) for dephosphorylated and nucleotide-free CFTR$_{FRET}$ (**a**), dephosphorylated CFTR$_{FRET}$ in the presence of 3 mM ATP (**b**), phosphorylated and nucleotide-free CFTR$_{FRET}$ (**c**), and phosphorylated CFTR$_{FRET}$ in the presence of 3 mM ATP (**d-e**). Panels **a–d** were collected with 100 ms integration time. Panel **e** was collected with 10 ms integration time. **f.** Contour plots of dose-dependent decrease in high FRET occupancy for phosphorylated CFTR$_{FRET}$ mediated by ADP in the presence of 3 mM ATP and increasing concentration of ADP. **g-h.** Contour plots showing time-dependent changes in FRET after rapid delivery (**g**) or withdrawal (**h**) of 10 mM ADP. CFTR$_{FRET}$ was phosphorylated prior to the experiments and 3 mM ATP was present throughout the experiments. **i-j.** Example single-molecule traces of changes in FRET dynamics of phosphorylated CFTR$_{FRET}$ upon rapid ADP delivery (**i**) or withdrawal (**j**). 3 mM ATP was present throughout the experiments. **k.** Representative titration of

ADP-mediated competitive inhibition of ATP-dependent CFTR current in an inside-out excised patch. CFTR was phosphorylated prior to the recording. **l.** ADP dose-responses of CFTR-mediated current and high FRET state occupancy for phosphorylated CFTR in the presence of 3 mM ATP. Responses were fitted using the Hill equation (solid lines) with an $IC_{50}$ of $0.57 \pm 0.05$ mM for opening and an $IC_{50}$ of $2.3 \pm 0.9$ mM for high FRET occupancy. Hill coefficients were fixed to 1. Data represent means and standard errors for 7-11 patches and 3 FRET experiments. **m.** Contour plot showing time-dependent increase in FRET after application of 300 nM PKA (at the dashed line) to CFTR$_{FRET}$ in the presence of 3 mM ATP. **n-o.** Contour plots showing time-dependent decreases in FRET after application of 1 μM λ phosphatase (indicated as λ) to phosphorylated wild-type (**n**) and E1371Q (**o**) CFTR$_{FRET}$, in the presence of 3 mM ATP. **p-q.** Example single-molecule traces of λ-dependent dephosphorylation of wild-type (**p**) and E1371Q (**q**) CFTR$_{FRET}$. **r.** Quantification of the rate of high FRET depopulation after λ phosphatase application for wild-type and E1371Q CFTR$_{FRET}$.

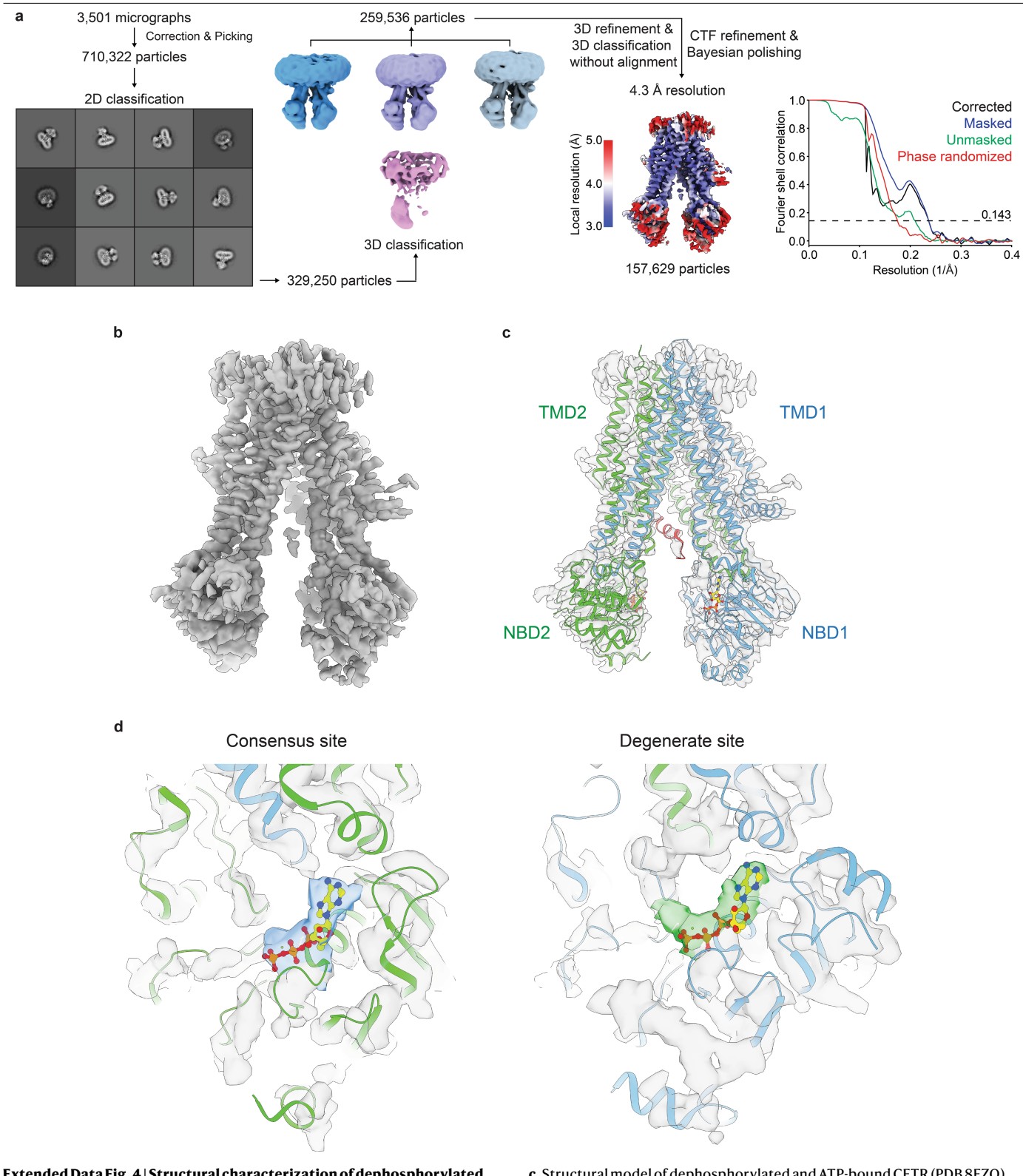

**Extended Data Fig. 4 | Structural characterization of dephosphorylated and ATP-bound CFTR. a.** Summary of cryo-EM workflow. **b.** cryo-EM map of dephosphorylated and ATP-bound wild-type CFTR at 4.3 Å resolution.

**c.** Structural model of dephosphorylated and ATP-bound CFTR (PDB 8FZQ) docked into the map. **d.** ATP densities in consensus (left) and degenerate (right) nucleotide binding sites.

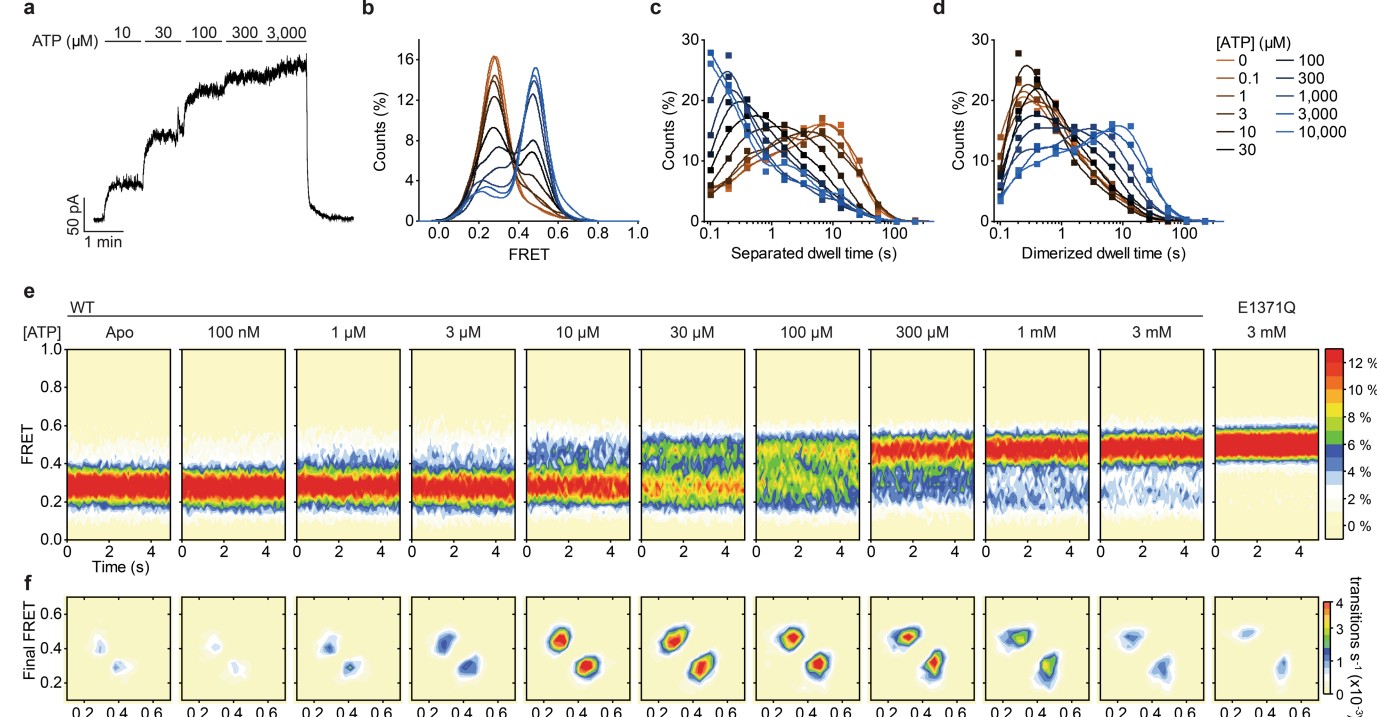

**Extended Data Fig. 5 | Nucleotide-sensitivity of CFTR dynamics.**
**a**. Representative ATP titration of CFTR-mediated current in an inside-out excised patch. C-terminally GFP-fused wild-type CFTR was phosphorylated prior to the recording. **b**. FRET distributions of phosphorylated CFTR$_{FRET}$ at the indicated ATP concentrations. **c-d**. Dwell time distributions of low FRET (**c**) and high FRET (**d**) states for phosphorylated CFTR$_{FRET}$ at the indicated ATP concentrations. **e**. Contour plots for phosphorylated wild-type and E1371Q variant CFTR$_{FRET}$ at the indicated ATP concentrations. **f**. Transition density plots for the same conditions as in **e**.

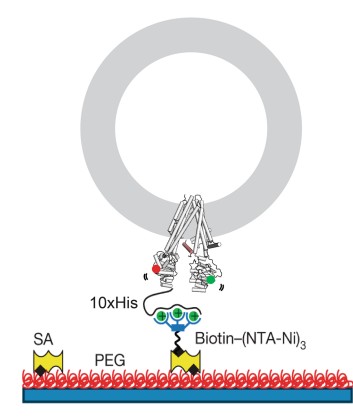

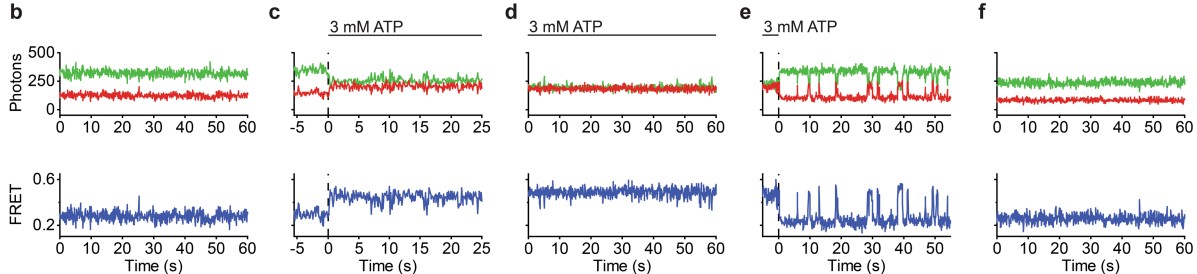

**Extended Data Fig. 6 | smFRET imaging of proteoliposome-reconstituted CFTR. a**. Schematic drawing of the immobilization strategy for proteoliposome-reconstituted CFTR$_{FRET}$. CFTR$_{FRET}$ molecules may be reconstituted with either orientation in the membrane. **b**–**f**. Example single-molecule traces showing donor (green) and acceptor (red) fluorescence intensities (top) and FRET (bottom) for phosphorylated and proteoliposome-reconstituted CFTR$_{FRET}$. Traces are before addition of ATP (**b**), upon addition of 3 mM ATP at the first dashed line (**c**), at steady state with ATP (**d**), upon ATP withdrawal at the second dashed line (**e**), and at steady state after ATP withdrawal (**f**). The representation of the immobilization surface in **a** was recreated from ref. [57].

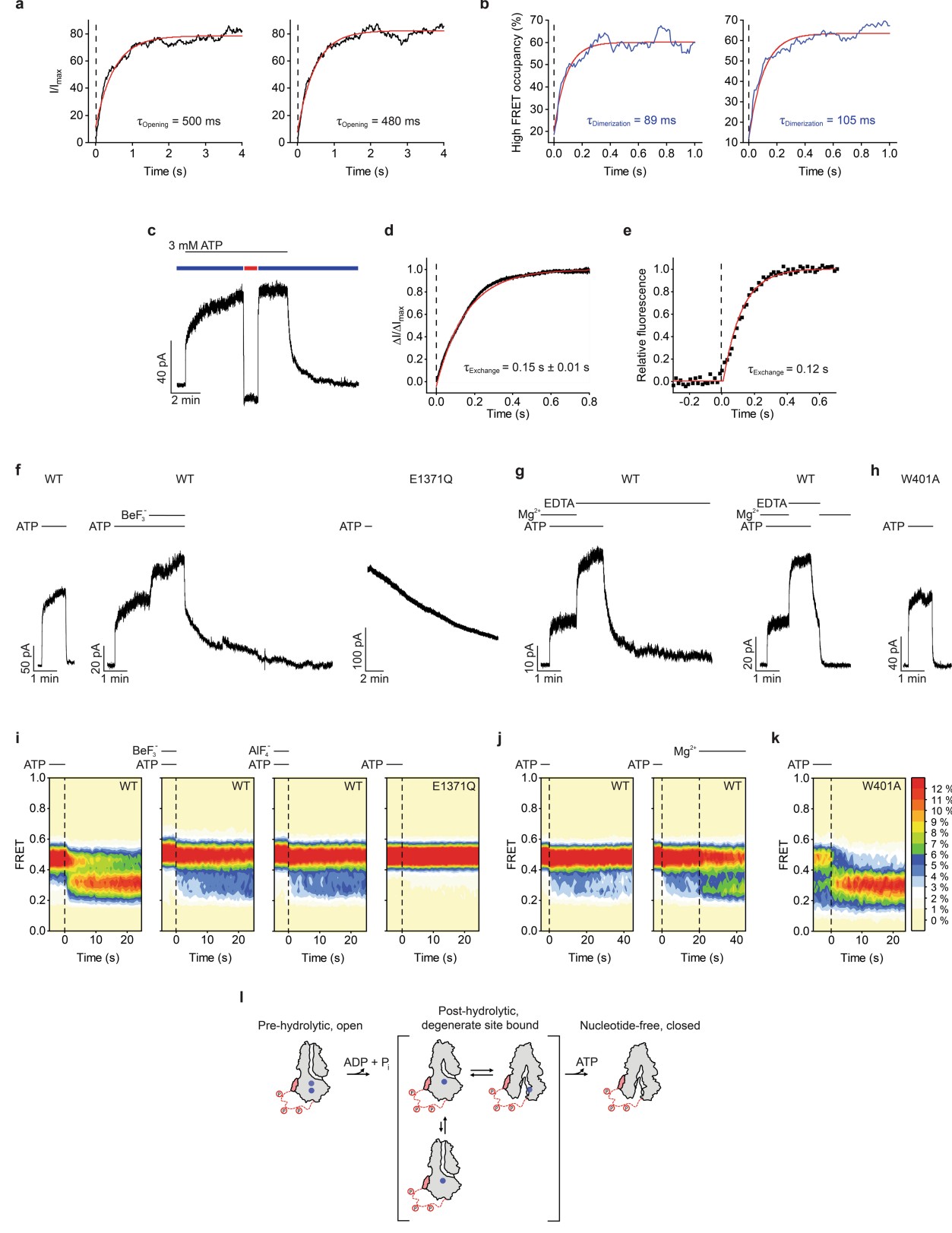

**Extended Data Fig. 7** | See next page for caption.

**Extended Data Fig. 7 | Correlating CFTR pore opening and closure with NBD conformation. a**. Example recordings of current responses to ATP application. 3 mM ATP was rapidly applied to an inside-out excised patch containing C-terminally GFP-fused wild-type CFTR (at the dashed lined). PKA-phosphorylation was performed prior to the recording. Time-courses were fitted with monoexponential functions (red lines). **b**. Example measurements of FRET responses to ATP application. 3 mM ATP was rapidly applied to PKA-phosphorylated CFTR$_{FRET}$ (at the dashed line). Time-courses were fitted with monoexponential functions (red lines). **c**. Representative inside-out excised patch showing the rate of solute exchange with local perfusion. CFTR was phosphorylated prior to the recording. CFTR current was then elicited by application of 3 mM ATP. The chloride Nernst potential was switched by exchanging from a chloride-containing perfusion solution (blue bar) to a sulfate-containing perfusion solution (red bar). **d**. Rate of current relaxation after solute exchange. Data represent means (solid black line) and standard errors (grey shaded area) for 7 patches. The time-course was fit with a monoexponential function (red line). **e**. Rate of relaxation in fluorescence intensity after injection (at the dashed line) of a DNA-conjugated Cy2 fluorophore into the imaging chamber. The experimental time-course (black points) was fitted with a monoexponential function (red line). **f**–**h**. Representative recordings of CFTR current relaxation upon ATP-withdrawal in inside-out excised patches:

ATP withdrawal from wild-type CFTR, BeF$_3^-$-trapped wild-type CFTR, and E1371Q CFTR (**f**); ATP withdrawal from wild-type CFTR in the absence of Mg$^{2+}$, and upon reapplication of Mg$^{2+}$ (**g**); ATP withdrawal from W401A CFTR (**h**). 2 mM Mg$^{2+}$ was present throughout the recordings in **f** and **h**. 3 mM ATP, 0.5 mM BeF$_3^-$, 2 mM Mg$^{2+}$, and 10 mM EDTA were perfused onto the patches where indicated. CFTR was activated by application of 300 nM PKA and 3 mM ATP prior to the displayed recordings. **i**–**k**. Contour plots of FRET responses after ATP-withdrawal from phosphorylated CFTR$_{FRET}$: ATP withdrawal from wild-type CFTR$_{FRET}$ in absence and presence of BeF$_3^-$ or AlF$_4^-$ and E1371Q CFTR$_{FRET}$ (**i**); ATP withdrawal from wild-type CFTR$_{FRET}$ in the absence of Mg$^{2+}$ and upon reapplication of Mg$^{2+}$ (**j**); ATP withdrawal from W401A CFTR$_{FRET}$ (**k**). 2 mM Mg$^{2+}$ was present throughout the experiments in **i** and **k**. 3 mM ATP, 0.5 mM BeF$_3^-$, 1 mM AlF$_4^-$, 2 mM Mg$^{2+}$, and 10 mM EDTA were present where indicated. **l**. Schematic of events underlying CFTR pore closure and NBD separation. CFTR bound to two ATP molecules is dimerized, open, and competent for hydrolysis. ATP hydrolysis at the consensus site is followed by release of ADP and inorganic phosphate (P$_i$). The degenerate site remains occupied by ATP and this intermediate dynamically transitions between dimerized and separated conformations. The dimerized state has low open probability without ATP in the consensus site. ATP release from the degenerate site leads to stable NBD separation and channel closure. ATP is shown as blue circles.

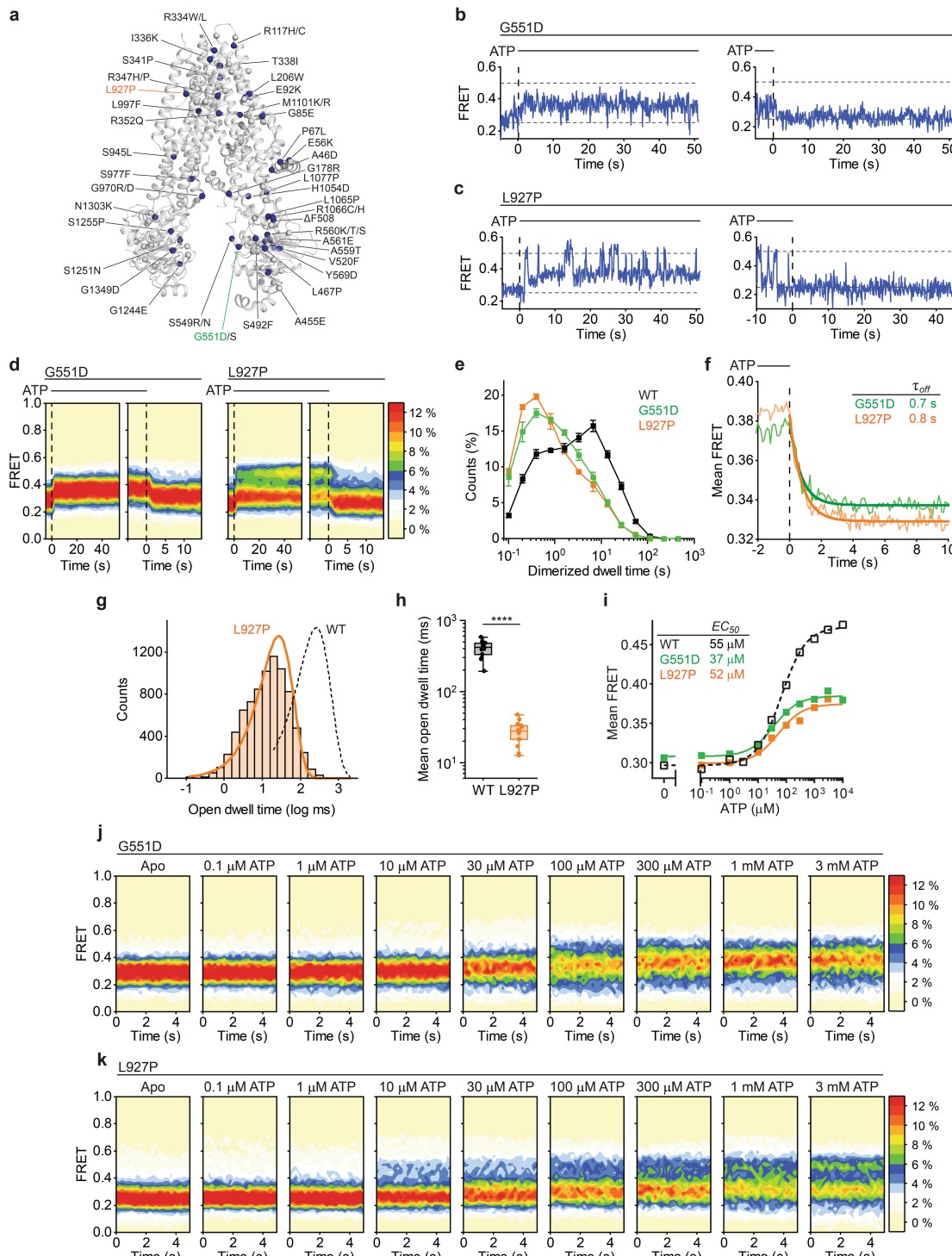

**Extended Data Fig. 8 | Characterization of cystic fibrosis-associated variants. a**. Positions of cystic fibrosis-causing missense mutations. Mutated sites are shown as spheres. **b–c**. Representative smFRET traces of ATP delivery to and withdrawal from phosphorylated G551D (**b**) and L927P (**c**) CFTR$_{FRET}$ (at the vertical dashed lines). Horizontal dashed lines indicate mean FRET efficiencies of the low and high FRET states. ATP concentration was 3 mM. **d**. Contour plots showing time-dependent changes in FRET after application and withdrawal of 3 mM ATP (at the dashed lines) to and from phosphorylated G551D and L927P variant CFTR$_{FRET}$. **e**. Dwell time distributions of the NBD-dimerized state for phosphorylated wild-type, L927P, and G551D CFTR$_{FRET}$ at 3 mM ATP. Data represent means and standard errors for 8 (wild-type) or 3 (G551D and L927P) experiments. **f**. FRET responses of G551D and L927P variants to ATP withdrawal (3 mM ATP to nucleotide-free at the dashed line).

Relaxation to low FRET was fit with monoexponential functions (solid lines). **g**. Open dwell time distribution of L927P variant CFTR. The distribution was fit with a monoexponential function (orange line). The dashed line is a monoexponential fit of the open dwell time distribution of wild-type CFTR. **h**. Mean open dwell times for wild-type and L927P variant CFTR. Whiskers represent minima and maxima and boxes represent 25th, 50th, and 75th percentiles for 13 (wild-type) or 13 (L927P) bilayer recordings. Statistical significance was tested using two-tailed Student's t-test (****$p = 2 \times 10^{-8}$). **i**. ATP dose-responses of mean FRET for G551D and L927P variants. Responses were fitted using the Hill equation with $EC_{50}$ values of 37 ± 7 μM for G551D and 52 ± 18 μM for L927P. **j-k**. Contour plots of phosphorylated G551D (**j**) and L927P (**k**) CFTR$_{FRET}$ at the indicated ATP concentrations.

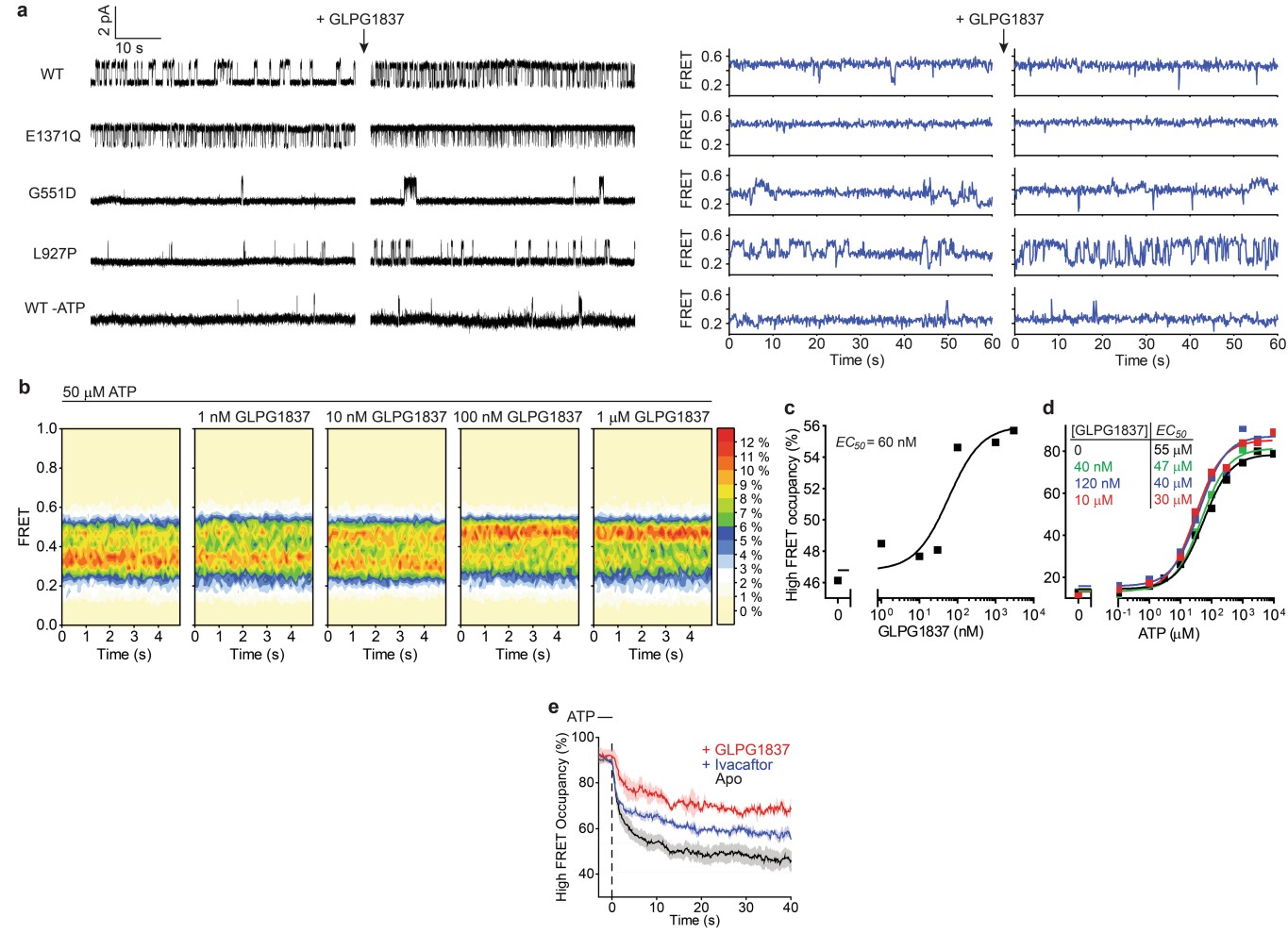

**Extended Data Fig. 9 | Potentiator-dependent changes in CFTR NBD dynamics. a**. Sample traces from single channel electrophysiology (black) and smFRET (blue) of wild-type, E1371Q, G551D, and L927P CFTR variants in the absence (left) or presence (right) of 10 μM GLPG1837. The bottom traces are wild-type CFTR in the absence of ATP. All other recordings were with 3 mM ATP. Measurements were performed with phosphorylated CFTR. In electrophysiology traces, upward deflections correspond to opening. **b**. Contour plots of phosphorylated wild-type CFTR$_{FRET}$ with 50 μM ATP and GLPG1837 at the indicated concentrations. **c**. GLPG1837 dose-response of high FRET occupancy for phosphorylated wild-type CFTR$_{FRET}$ at 50 μM ATP.

The response was fit using the Hill equation (solid line) with an $EC_{50}$ value of $0.06 \pm 0.04$ μM. **d**. ATP dose-responses of high FRET occupancy in the presence of GLPG1837 at the indicated concentrations. The responses were fit using the Hill equation (solid lines) with $EC_{50}$ values of $55 \pm 8$ μM (no GLPG1837), $47 \pm 8$ μM (40 nM GLPG1837), $40 \pm 9$ μM (120 nM GLPG1837), and $30 \pm 5$ μM (10 μM GLPG1837). **e**. Rates of depopulation from the high FRET state after ATP withdrawal from phosphorylated wild-type CFTR$_{FRET}$ in the absence of potentiator, with 10 μM GLPG1837, or with 100 nM Ivacaftor. Data represent means and standard errors (shaded area) for 5 (without potentiator) or 3 (with GLPG1837 or Ivacaftor) experiments.

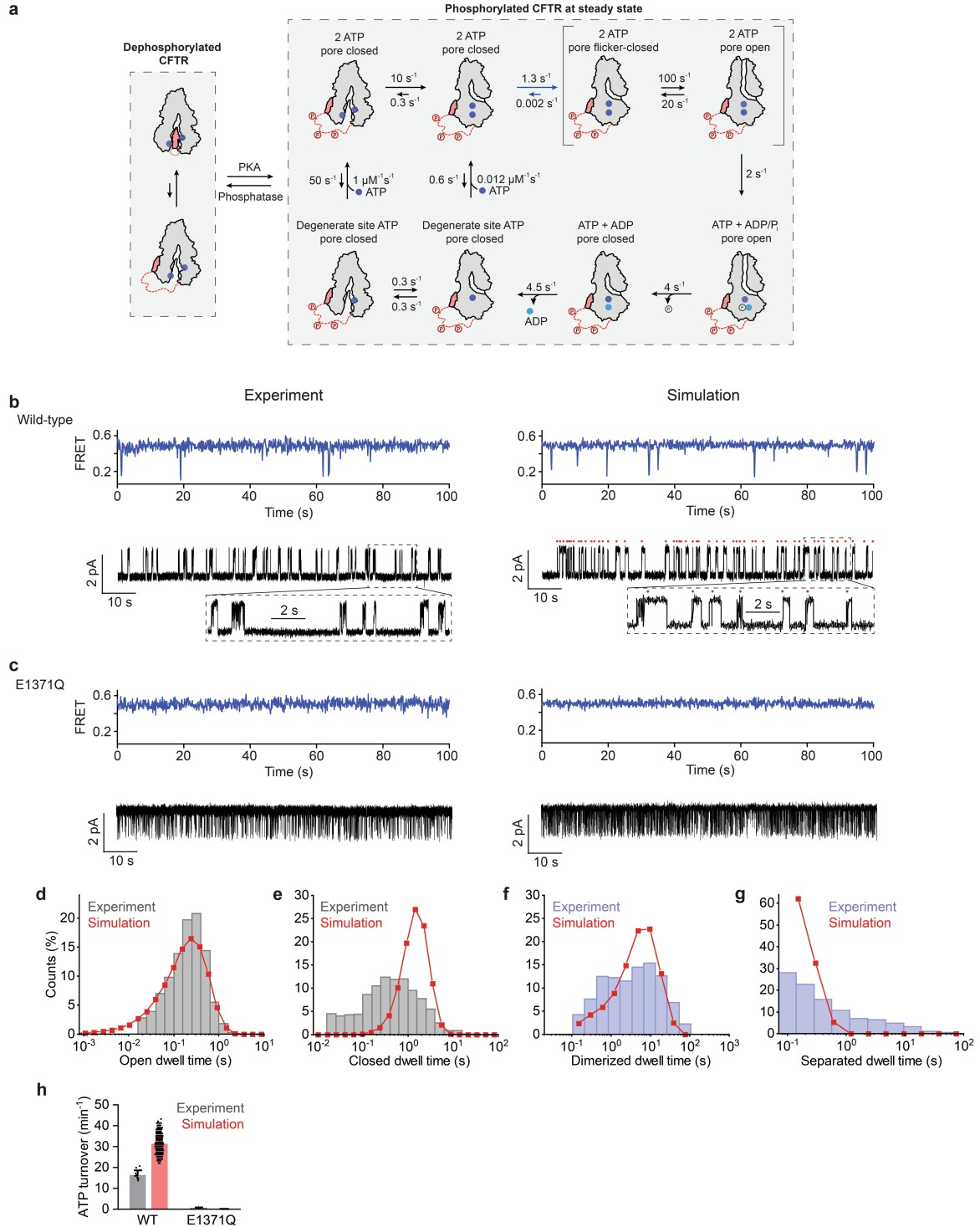

**Extended Data Fig. 10 | Stochastic simulation of CFTR pore and NBD dynamics. a**. Topology and rates of the simulated model. **b–c**. Experimental and simulated smFRET and single-channel electrophysiology traces of the wild-type (**b**) and E1371Q variant CFTR (**c**). The red dots above the simulated current traces indicate ATP hydrolysis events. **d–g**. Comparison of experimental and simulated dwell time distributions of pore open (**c**), pore closed (**d**), NBD dimerized (**e**), and NBD separated (**f**) states for wild-type CFTR.

Experimental distributions are shown as histograms, and simulated distributions as red lines. Flicker-closures were ignored in analysis and the open and closed dwell times therefore reflect open burst and interburst dwell times, respectively. **h**. Experimental and simulated steady state ATP hydrolysis rates. Data represent means and standard deviations for 10 (wild-type) or 3 (E1371Q) measurements and 1500 (wild-type or E1371Q) simulated molecules.

**Extended Data Table 1 | Cryo-EM data collection, refinement and validation statistics**

|  | Wild-type CFTR Dephosphorylated and ATP-bound (EMDB-29637) (PDB 8FZQ) |
|---|---|
| **Data collection and processing** | |
| Magnification | 29,000 |
| Voltage (kV) | 300 |
| Electron exposure (e–/Å$^2$) | 75.4 |
| Defocus range (μm) | 0.8 to 2.5 |
| Pixel size (Å) | 1.03 |
| Symmetry imposed | C1 |
| Initial particle images (no.) | 710,322 |
| Final  particle images (no.) | 157,629 |
| Map resolution (Å) | 4.3 |
| FSC threshold | 0.143 |
| Map resolution range (Å) | 2.4 to 12.4 |
|  | |
| **Refinement** | |
| Initial model used (PDB code) | 5UAK |
| Model resolution (Å) | 4.3 |
| FSC threshold | 0.5 |
| Map sharpening $B$ factor (Å$^2$) | -117.84 |
| Model composition | |
| Non-hydrogen atoms | 9244 |
| Protein residues | 1152 |
| Ligands | 4 |
| $B$ factors (Å$^2$) mean | |
| Protein | 106.59 |
| Ligand | 86.60 |
| R.m.s. deviations | |
| Bond lengths (Å) | 0.002 |
| Bond angles (°) | 0.515 |
| Validation | |
| MolProbity score | 1.46 |
| Clashscore | 8.33 |
| Poor rotamers (%) | 0.00 |
| Ramachandran plot | |
| Favored (%) | 97.98 |
| Allowed (%) | 2.02 |
| Disallowed (%) | 0.00 |

# Reporting Summary

## Statistics

For all statistical analyses, confirm that the following items are present in the figure legend, table legend, main text, or Methods section.

| n/a | Confirmed | |
|---|---|---|
| ☐ | ☒ | The exact sample size (*n*) for each experimental group/condition, given as a discrete number and unit of measurement |
| ☒ | ☐ | A statement on whether measurements were taken from distinct samples or whether the same sample was measured repeatedly |
| ☐ | ☒ | The statistical test(s) used AND whether they are one- or two-sided <br> *Only common tests should be described solely by name; describe more complex techniques in the Methods section.* |
| ☐ | ☒ | A description of all covariates tested |
| ☒ | ☐ | A description of any assumptions or corrections, such as tests of normality and adjustment for multiple comparisons |
| ☐ | ☒ | A full description of the statistical parameters including central tendency (e.g. means) or other basic estimates (e.g. regression coefficient) AND variation (e.g. standard deviation) or associated estimates of uncertainty (e.g. confidence intervals) |
| ☐ | ☒ | For null hypothesis testing, the test statistic (e.g. *F*, *t*, *r*) with confidence intervals, effect sizes, degrees of freedom and *P* value noted <br> *Give P values as exact values whenever suitable.* |
| ☒ | ☐ | For Bayesian analysis, information on the choice of priors and Markov chain Monte Carlo settings |
| ☒ | ☐ | For hierarchical and complex designs, identification of the appropriate level for tests and full reporting of outcomes |
| ☒ | ☐ | Estimates of effect sizes (e.g. Cohen's *d*, Pearson's *r*), indicating how they were calculated |

*Our web collection on statistics for biologists contains articles on many of the points above.*

## Software and code

Policy information about availability of computer code

| Data collection | Software used are commercial products: Labview 2017, Clampex 10.7, SerialEM 3.7. |
|---|---|
| Data analysis | All software used are commercial products or otherwise publicly available: Chimera 1.13.1, Clampfit 10.7, Coot 0.8.9.2, GraphPad Prism 8, ImageJ 2.0, Matlab R2019a, Motioncorr2, OriginPro 2022, Pymol2, Relion-3, and SPARTAN. |

For manuscripts utilizing custom algorithms or software that are central to the research but not yet described in published literature, software must be made available to editors and reviewers. We strongly encourage code deposition in a community repository (e.g. GitHub). See the Nature Portfolio guidelines for submitting code & software for further information.

## Data

Policy information about availability of data

All manuscripts must include a data availability statement. This statement should provide the following information, where applicable:
- Accession codes, unique identifiers, or web links for publicly available datasets
- A description of any restrictions on data availability
- For clinical datasets or third party data, please ensure that the statement adheres to our policy

Structural models in the RCSB PDB under accession codes 5UAK, 6MSM, and 6O2P were used in this study.
The CFTR2 database was used in this study.
The cryo-EM map has been deposited in the Electron Microscopy Data Bank (EMDB) under accession code: EMD-29637. The corresponding atomic model has been

deposited in the Protein Data Bank (PDB) under accession code 8FZQ.
The data that support the findings are available from the authors upon reasonable request.

## Human research participants

Policy information about studies involving human research participants and Sex and Gender in Research.

| | |
|---|---|
| Reporting on sex and gender | N/A |
| Population characteristics | N/A |
| Recruitment | N/A |
| Ethics oversight | N/A |

Note that full information on the approval of the study protocol must also be provided in the manuscript.

# Field-specific reporting

Please select the one below that is the best fit for your research. If you are not sure, read the appropriate sections before making your selection.

☒ Life sciences  ☐ Behavioural & social sciences  ☐ Ecological, evolutionary & environmental sciences

For a reference copy of the document with all sections, see nature.com/documents/nr-reporting-summary-flat.pdf

# Life sciences study design

All studies must disclose on these points even when the disclosure is negative.

| | |
|---|---|
| Sample size | Statistical methods were not used to determine sample sizes. Sample sizes were determined by the normal throughput of the instrumentation while ensuring that three or more independent replicates were performed for statistical analysis. For single molecule FRET experiments, each replicate reflects approximately 300 to 2000 analyzed trajectories. |
| Data exclusions | Single molecule FRET trajectories were selected for analysis based on the following pre-established criteria: (1) single-step donor photobleaching, (2) a signal-to-noise ratio > 8, (3) fewer than four donor blinking events, and (4) FRET efficiency above baseline for at least 50 frames. Details on data exclusion criteria are described in the manuscript.<br>Further, single molecule traces exhibiting FRET values above 0.8 were excluded from analysis. This sub-population was insensitive to phosphorylation and nucleotide addition, and likely reflected denatured molecules. This criterion was not pre-established.<br>Details on data exclusion can also be found in the manuscript. |
| Replication | Findings were reliably replicated across different preparations. The number of experimental replicates are specified in the legends of all figures. |
| Randomization | This study did not allocate experimental groups thus no randomization was required for the reported experiments. |
| Blinding | Blinding was not performed. No a priori knowledge could be assumed about the present observations and blinding is therefore not applicable. Data was analyzed systematically as described in the manuscript. |

# Reporting for specific materials, systems and methods

We require information from authors about some types of materials, experimental systems and methods used in many studies. Here, indicate whether each material, system or method listed is relevant to your study. If you are not sure if a list item applies to your research, read the appropriate section before selecting a response.

## Materials & experimental systems

| n/a | Involved in the study |
|---|---|
| ☐ | ☒ Antibodies |
| ☐ | ☒ Eukaryotic cell lines |
| ☒ | ☐ Palaeontology and archaeology |
| ☒ | ☐ Animals and other organisms |
| ☒ | ☐ Clinical data |
| ☒ | ☐ Dual use research of concern |

## Methods

| n/a | Involved in the study |
|---|---|
| ☒ | ☐ ChIP-seq |
| ☒ | ☐ Flow cytometry |
| ☒ | ☐ MRI-based neuroimaging |

# Antibodies

| | |
|---|---|
| Antibodies used | This study used an anti-GFP nanobody that was initially described in:<br>Kirchhofer, A., Helma, J., Schmidthals, K. et al. Modulation of protein properties in living cells using nanobodies. Nat Struct Mol Biol 17, 133–138 (2010). |
| Validation | The nanobody was previously used for the same application as described in:<br>Liu, F., Zhang, Z., Csanády, L., Gadsby, D. C., & Chen, J. Molecular structure of the human CFTR ion channel. Cell 169, 85–95 (2017).<br>Zhang, Z., Liu, F., & Chen, J. Molecular structure of the ATP-bound, phosphorylated human CFTR. Proc Natl Acad Sci USA 115, 12757–12762 (2018).<br>Liu, F., et al. Structural identification of a hotspot on CFTR for potentiation. Science, 364, 1184–1188 (2019). |

# Eukaryotic cell lines

Policy information about cell lines and Sex and Gender in Research

| | |
|---|---|
| Cell line source(s) | CHOK1 (CCL-61, lot number 70014310) and HEK293S GnTI- (CRL-3022, lot number 62430067) cells were acquired from the American Type Culture Collection (ATCC). Sf9 cells (catalog number 11496015, lot number 1670337) were acquired from GIBCO. |
| Authentication | CHOK1 and HEK293S GnTI- cells were authenticated by the American Type Culture Collection (ATCC). Details of authentication are outlined by the vendor.<br>CHOK1 cells: https://www.atcc.org/products/ccl-61#generalinformation. Specifically, CHOK1 cells were visually inspected for appropriate morphoplogy, validated to be the correct species by a COI assay, and tested for contamination of bacteria and mycoplasma.<br>HEK293S GnTI- cells: https://www.atcc.org/products/crl-3022. Specifically, HEK293S GnTI- cells were visually inspected for appropriate morphology, validated to be the correct species by a COI assay and STR analysis, and tested for contamination of bacteria, mycoplasma, and human pathogenic viruses.<br>Sf9 cells were authenticated by the vendor as outlined at: https://www.thermofisher.com/order/catalog/product/11496015. Sepcifically, Sf9 cells were tested for contamination of bacteria, yeast, mycoplasma and virus and were characterized by isozyme and karyotype analysis. |
| Mycoplasma contamination | Sf9 and HEK293S GnTI- cells tested negative for Mycoplasma contamination. CHOK1 cells were not tested for mycoplasma contamination. |
| Commonly misidentified lines<br>(See ICLAC register) | No commonly misidentified cell lines were used in this study. |

