## [Peer Review File · Nature]

Manuscript Title: CFTR function, pathology and pharmacology at single-molecule resolution

Reviewer Comments & Author Rebuttals

Reviewer Reports on the Initial Version:

Referees' comments:

Referee #1 (Remarks to the Author):

The Levring et al. paper seeks to determine how dimerization of CFTR NBDs is coupled to ATP binding and hydrolysis, and to pore opening, using a combination of smFRET and electrophysiology. CFTR cryo-EM studies revealed two conformations: one in absence of phosphorylation and ATP with the pore-closed and NBDs separated by and the R domain sterically blocking their dimerization and the other phosphorylated with ATP bound and NBDs dimerized with two ATP molecules bound at their interface. The goal here was to elucidate the relationship between NBD dimerization and gating, understand how ion permeation is coupled to ATP hydrolysis and NBD isomerization and gain insight into the effect of disease-causing mutations and pharmacological modulation for therapeutics. The manuscript presents a detailed molecular explanation for how conformational change and channel opening are coupled and provides insight into modes for potentiator action. The authors do a commendable job of integrating multiple types of biophysical and functional data. The writing is clear (in most places), the experiments are well designed and thoughtfully interpreted. The study contributes significantly to the field. Several issues of data presentation and interpretation should be addressed.

Major comments:

1) The conclusions hinge on quantitative comparisons between smFRET data and electrophysiology data. Some clarifications are needed.

The properties of CFTR are compared to the unlabeled CFTRFRET in E1b-g and the fluorophore-labelled version in Fig. E1i-k and E2. For Fig. 1 and 3 it is not clear whether this is the labeled or unlabeled version. The authors should clarify this in text, Methods and legends in a consistent manner. A clear nomenclature would help. If Figs. 1 and 3 use the unlabeled version, authors should justify why. Also, authors should note if the constructs being compared differ in possessing a C-terminal GFP.

In E1i-j the current–voltage relationships suggest that CFTRFRET has higher conductance (authors should indicate how many patches went into this comparison). In addition, missing is a comparison of open probability, which is critical for the interpretation.

2) Even when the reader knows that patch recordings and FRET come from different experiments, placing these side by side or one above the other could give the confusing impression that they were measured simultaneously. This confusion is aggravated by sentences like this in the text (line 206):

“The W401A variant, which is capable of binding and hydrolyzing ATP via the consensus site, underwent rapid transitions between NBD-separated and -dimerized states that correlated with channel gating (Figure 2c).” This sentence should be changed (e.g. to “more closely resembled,” not correlated). To further clarify this, authors should add to Fig 2 a panel showing dimerization probability and compare open and dimerized dwell times the way they did in WT channels in Fig 1. Authors should also define for panel f ratio between probabilities of opening and dimerization which is the numerator.

3) The conclusion of Fig. 3 is that NBD dimerization precedes channel opening. The average rise upon rapid ATP perfusion of the phosphorylated channel of high FRET occupancy (dimerization) in smFRET and of channel opening in patch is faster for dimerization. Quantification was by mono-exponential fits. These should be superimposed on the data. Tau of dimerization is shown as <100 ms. The precise value and SEM should be given.

Fig. E7 is supposed to demonstrate that differences in kinetic behavior seen in Fig 3 are not due to differences in solute exchange between the two sets of experiments, but this figure only displays e-phys data. What is the equivalent dataset for smFRET, and where is it displayed? The Tau of patch solution exchange in Fig. E7 is 150 ms, slower than the <100 ms value given for smFRET in Fig. 3a, so it is unclear how well the exchange rates actually match.

The authors state: “We conclude ... that the mean first passage time between NBD dimerization and channel opening in productive instances is approximately 500 ms.” However, their Tau of opening is 490 ms and they begin to see channels opening almost immediately. The delay they seek is the one between dimerization and opening and would be a smaller value.

Minor comments:

- 1) E1j is mistakenly referred to as E1k in the text (line 161)
- 2) All electrophysiology traces should have current amplitude scale bar (e.g. missing in Fig. 2c)
- 3) Figure 1. E1371Q should be shown without ATP as well.
- 4) No mass spec data to demonstrate that phosphorylation is consistent/present across all conditions
- 5) Figure 2. 2cd ATP concentration and phosphorylation status should be stated in legend for clarity. It is also not so easy to determine by eye the contribution of open probability and dimerization to the coupling ratio by looking at 2e and 2f. It would help to additionally show the dimerization probability for the five different variants.
- 6) Example traces in E1i look different at lower membrane potentials (-90 and -150)-- is there a better representative trace or what is the hypothesized reason for this difference?
- 7) Analogous examples of current from WT channels should be shown for comparison in E1b and E1k.
- 8) Line 140 NDB typo
- 9) A question regarding comparison of electrophysiology and FRET data-- are there any integral membrane phosphatases which could contribute to the observed differences between channel opening and NBD dimerization probability?
- 10) The authors should make clear that disease-causing CFTR variants have multiple modes/effects,

and should describe the direction in which a pharmacological agent should act (i.e. promote channel opening as opposed to promoting channel closure, etc). On p. 9, the effects of G551D and L927P on channel gating should be described first, before explaining effects on FRET-based readouts.

11) Presence of a native lipid membrane or detergent environment may affect the dynamics of the channel. To address this, the authors perform a set of FRET experiments in proteoliposomes, which should more accurately approximate a cell membrane. The FRET dynamics and dependence on ATP look similar to what they see in detergent, bolstering confidence in their experiments. One odd thing is that only a small fraction of channels are ATP sensitive. They attribute this to channel orientation within the bilayer. If the 10xHis tag is located on the NBD side of the protein, it is unclear why any immobilized molecules should have inward-facing NBDs which are inaccessible to ATP.

12) It is not clear if the kinetic model adds much to the discussion, but it also is presented with appropriate caveats. Does the model make a prediction in a way that could be used to test the alternative hypotheses presented in the introduction of the paper?

13) Figure 1i shows that a very large percentage of channel openings have a dwell time of at least 10 ms, which could be detected using smFRET. Though it may not be feasible for this study, it would be highly convincing if it were possible to design a FRET sensor of channel pore activity that approximates the electrophysiological readout? This would allow direct comparison within a single controlled system.

Referee #2 (Remarks to the Author):

The current manuscript by Levring et al. presents incredibly comprehensive studies of CFTR function as an ATP-gated ion channel; these studies include electrophysiological recordings of CFTR channel activity, real-time monitoring of the dimerization of CFTR's two nucleotide binding domains (NBDs) with smFRET imaging, biochemical measurements of ATP hydrolysis rate, and computer simulations based on a proposed gating model. Here are the main findings and their mechanistic implications:

1. The NBDs are readily dimerized by ATP binding to the two ATP-binding sites in fully phosphorylated CFTR, but the dimerized state (high FRET) is much more stable than the duration of either channel open state or the closed state at a saturating concentration of ATP. This uncoupling between NBD dimerization and channel gating suggests that multiple cycles of gate opening and closing can occur with NBDs remained dimerized (inferred from high FRET), contradicting previous idea of a strict coupling of NBD dimerization to gate opening and hydrolysis-triggered NBDs separation (low FRET) to gate closing (Csanady et al., 2010).

2. By manipulating ATP binding to each site (Y1219A for the consensus site and W401A for the degenerate site), they provided evidence for NBD dimerization by ATP occupancy at either site. While both NBD dimerization and gating were mostly preserved in W401A, Y1219A channels showed robust NBD dimerization but little activity, supporting previous reports that gating of CFTR is mainly through ATP binding to the consensus site (Vergani et al., 2003; Zhou et al., 2006).

3. By comparing the rate of NBD dimerization and the rate of channel activation upon sudden addition of ATP, they found that NBD dimerization takes place much faster than gate opening, contradicting previous proposition that NBD dimerization constitutes the rate-limiting step in CFTR

gating (Vergani et al., 2005). Although NBD dimerization precedes gate opening, NBD dimerization itself is not sufficient to open the gate because dimerization (high FRET) can be seen in conditions when there is little channel activity (e.g., upon ATP washout, or in the presence of inhibitory concentration of ADP).

4. Supporting the idea that ATP at the consensus site plays a critical role in opening the gate, the authors demonstrated that the disease-causing mutant G551D at the consensus site exhibits defective gating and NBD dimerization. Interestingly, the dimerized NBDs in G551D show FRET between high and low FRETs, suggesting the presence of an intermediate NBD dimer proposed previously (Tsai et al., 2010). Similar but not identical intermediate FRET was seen in another pathogenic mutation L927P (in TM8), which is 50 Å away from the consensus site. This latter observation suggests a long-distance effect of the L927P mutation on NBD dimerization, but the nature of this allosteric effect is unclear.

5. By testing the effects of CFTR potentiators ivacaftor and GLPG1837, they found a dramatic gating effect with minimal and moderate changes of NBD dimerization in G551D and L927P variants respectively. These are interesting observations, but it seems that not much additional insight can be gained (see below).

6. These results were put together to form the basis of a kinetic model, which was further examined through computer simulations. Although the proposed kinetic mechanism recapitulates many of the experimental data, some simulated results do deviate from actual data.

No doubt, these are important studies that reveal many unexpected phenomena, which lay a solid foundation for settling the debates over the coupling mechanism between NBD dimerization and gate opening, as well as for advancing our overall understanding of the structural-functional relationships of CFTR gating. This reviewer applauds the authors for such a remarkable attainment.

Major comments:

1. The two conformations of NBDs—separated versus dimerized—can be resolved by the shift of FRET from 0.25 to 0.49 (Fig. 1). In Fig. 1e, the FRET signal remains stable at 0.49 at 10 mM ATP while the electrophysiological recording shows repeated opening and closing of the WT-CFTR channel. The authors thus concluded that the NBDs remain dimerized throughout multiple rounds of ATP binding/hydrolysis (Fig. 5). Conceptually, the consensus site in the “dimerized NBDs” must open up a space wide enough to allow the exchange of hydrolytic products for the next ATP. Indeed, in ATP washout (and Mg addition) experiments (Fig. 3b, c), transition from high FRET to low FRET was observed immediately upon ATP removal (or Mg addition). As the waiting time is too short for ATP dissociation from the degenerate site, this result suggests that upon ATP hydrolysis at the consensus site, disengagement of the two NBDs does occur. The reason this high to low FRET transition is not seen in the continuous presence of millimolar ATP may be due to rapid rebinding and re-dimerization of the disengaged NBDs at millimolar ATP. This scenario supports the notion that the state “degenerate site ATP, pore closed” should assume a different dimeric state with a cracked consensus site to allow step 6 (Fig. 5) to take place. The canonical NBD dimer in the cryo-EM structure leaves no room for ATP to diffuse into (or out of) its binding pocket.

2. I wish the authors would have elaborated the rationale behind choosing positions 388 and 1435 for their FRET pair. T388 is at a position without a concrete secondary structure. Wouldn't this positioning result in some uncertainty in FRET measurements as the FRET efficiency is determined by not only the distance but also the relative orientation between the donor and the acceptor? It would be nice if another pair is chosen to replicate some of the key observations described above.

3. The W401A and Y1219A mutations were designed to weaken ATP binding. The authors observed that the NBDs of W401A and Y1219A mutants can still dimerize, and thus concluded that ATP binding at either site is sufficient for NBD dimerization (P7, line 215). Such conclusion is based on the assumption that the degenerate site in W401A (or the consensus site in Y1219A) must be vacant at 3 mM ATP. However, in electrophysiological study, the Y1219G mutation reduces the apparent affinity for ATP to a $K_{1/2}$ of ~ 5 mM (Zhou et al., 2006). If we assume that Y1219A affects the affinity similarly, at least a portion of the Y1219A channel population in Fig. 2 would still have ATP bound to the consensus site at 3 mM ATP. Given that the degenerate site has much higher affinity for ATP, it is unlikely that the degenerate site in W401A would be vacant at 3 mM ATP. Thus, the data in Fig. 2 only support that NBDs of W401A and Y1219A can dimerize. The occupancy of ATP at each site cannot be determined, and thus whether a single ATP is enough to trigger dimerization cannot be inferred conclusively. However, the conclusion may not be incorrect once data in Fig. 1j are taken into consideration. The ATP concentration dependence of high to low FRET transition rate suggests the existence of different states of NBD dimer with one or both sites occupied.

4. Some of the kinetic steps in Fig. 5 need justification/explanation:

a. Dissociation/binding of ATP in the degenerate site happens when the NBDs are separated (step 7, 8), but dissociation of hydrolytic products and binding of ATP to the consensus site happen when the NBDs are dimerized (step 4, 5, 6). Some intermediate conformations of NBDs need to exist.

b. Why step 3 is drawn differently in Fig. 5 and Fig. E11? Also, why did the authors propose only the open state hydrolyzes ATP, but not 2 ATP pore flicker-closed state or 2 ATP pore closed state?

c. Is the flicker-closed state the same as the cryo-EM structure of E1371Q-hCFTR? It's been known that many of the flickery events seen at negative membrane potentials are not likely real functional "state". Because of their high voltage dependence (see Fig. E1i.), many of the events are considered blocking of the pore by cytosolic large anions. To get the right kinetic parameter for the true flicker-closed state, one has to analyze data from recordings at positive membrane potentials.

d. There is a mistake both in the text and in the figure legend. Steps 6 – 8 represent transitions among closed states; therefore, it is not a new gating cycle, which by definition should include a closed event and an open event.

Minor comments:

1. I am not sure I truly understand the meaning of the term "hierarchical" gating mechanism.

2. P2, line 3: Do you mean electrophysiological properties?
3. P5, line 140: It should be "...to a stably NBD-dimerized state."
4. P6, line 157: Previous electrophysiological studies of R117H-CFTR and zebrafish CFTR, as well as the cryo-EM structure of E1371Q all indicate the presence of conductive and non-conductive states with dimerized NBDs. Please more explicitly state the unique insight from these data.
5. P6, line 161: ...with ATP concentration (Figure 1j), not 1k.
6. P6, line 172: The ADP experiments are interesting in that the low FRET value seems different from that without ATP. Is it reproducible? If it is, doesn't that suggest different degrees of disengagement of NBDs?
7. P7, line 191: The time constant for ligand exchange at the degenerate site is around 30 s (Tsai et al., 2010), which interestingly is very similar to the duration of high FRET upon ATP washout (Fig. 3b).
8. P7, line 226: It should be open probability, not opening probability. (In other parts of the text too.)
9. P8, line 237: The changes of FRET upon sudden addition of ATP show heterogeneity. This observation may be related to a recent paper by Yeh et al. (2021). The stated similarity to W401A and double exponential current rising upon addition of ATP (Fig. 1b and E1b) all remind this reviewer of the results published in this paper, which shows different closed channel conformations (perhaps different NBDs conformations) once ATP is removed.
10. P8, line 255: Of note, rapid removal of ATP caused a double exponential decay of WT-CFTR currents (Fig. 2B in Lin et al., 2014) with time constants ($\tau_1 < 1$ s and $\tau_2 = 29$ s) similar to these values reported for NBD dimerization relaxation. These shouldn't be just coincident.
11. P9, line 277: Not a correct reference and interpretation. The cited ligand exchange experiments lead to two different turn-over rates of ATP and the idea of a "partial dimer." On the other hand, by carrying out different ligand exchange experiments, Jih et al. (2012) propose a re-entry pathway featuring repetitive cycles of ATP turnover in a single opening burst. Both studies done by Hwang and colleagues invoke a partial dimer state where the consensus site is open up to allow replenish of ATP. They do not suggest that ATP turnover can occur in a single canonical NBD-dimerized conformation, where the ligand is occluded.
12. P10, line 312-314: I don't understand what the authors mean by the similar nature in "the similar nature of both mutations' impacts on NBD dimerization and channel opening". G551D and L927P actually exhibit different behaviors in their experiments. In Fig. 4b, the transitions between high and low FRET in G551D is different from those of L927P. Also, GLPG1837 drastically changes L927P's FRET pattern, but barely affects G551D. Moreover, electrophysiological properties of these two mutants and pharmacological effects of CFTR potentiators should be discussed in a more quantitative manner (see comment 18 below) before one can make more definitive conclusions.

13. P11, line 347: It should be noted that both VX-770 and GLPG1837 work on a CFTR variant with its NBD2 completely removed (Yeh et al., 2017; Yeh et al., 2015). These drugs also exert dramatical gating effects on G551D whose NBDs may not assume a canonical NBD dimer state (Lin et al., 2014; Yeh et al., 2017) (Fig. 4 in current study).

14. P11, line 375-377: Previous studies suggest that reagents such as NPPB and high-affinity ATP analogs, act synergistically with Ivacaftor and GLPG1837 on G551D channels by promoting NBD dimerization (Lin et al., 2016; Yeh et al., 2017). The authors may consider incorporating these studies in the Discussion.

15. Figure 1: I suggest the authors unify the spelling of "dimerization." It is dimerization in the text but dimerisation in Fig 1 and Fig 4.

16. Figure 1e: If I understand correctly, the electrophysiological recordings are from WT-CFTR, not from the CFTRFRET protein. Wouldn't it be a better comparison to use the electrophysiological recording of CFTRFRET protein here? Of note, the CFTRFRET construct altered the majority of the endogenous cysteine residues; this alteration could affect CFTR gating as seen in the literature. In Figure E1, the authors demonstrated their ability to use CFTRFRET proteins in electrophysiological experiments and concluded that its gating kinetics are similar to WT-CFTR (but see comment 19 below).

17. Figure 1j: I don't understand the data points at 0 ATP. Do the authors still observe FRET transitions even without ATP, meaning that the NBDs dimerize without ATP? In 1e, FRET always stays at low FRET.

18. Figure 4b: In response to GLPG1837, the current trace of G551D barely changes as if GLPG1837 has little effect. This potentiation effect is far from the reported ~30-fold stimulation shown in Fig. 4g. Please comment on the discrepancy and consider replacing the recording with a more representative one. Similar concerns are raised for the effect of GLPG1837 on L927P. The single-channel activity in 4b does not change 20-fold as indicated in Fig. 4g.

19. Figure 4f: It is shown that the P_o of E1371Q is ~0.7 and increases to ~0.8 with GLPG1837. It is puzzling how E1371Q has a P_o of only 0.7. Numerous studies showed that with hydrolysis eliminated, the closing rate of E1371Q (or E1371S) is >100-fold slower than WT, resulting in a P_o close to unity (Bompadre et al., 2005; Csanady et al., 2013; Vergani et al., 2003; Yu et al., 2016). Indeed, the E1371Q single channel in Figure 1f seems to be almost always open, though it is hard to determine whether those brief downward deflections are flicker closures or interburst closures with this time scale. It would be helpful to expand an E1371Q single channel trace to allow a better assessment of channel behavior in a quantitative manner.

20. Figure E1i: From the raw trace, the P_o of the FRET construct is apparently higher than the WT. The open burst time of FRET variant is visibly longer than WT. Is this a consistent finding? Please comment on the discrepancy.

21. Figure E8g: What is the second arrow below the ADP+Pi for?

22. Figure E11b, d, and e: In b, the simulated single-channel trace shows a much longer open time than the experiment, while the closed time seems not affected much (or slightly shorter). However, in d and e, the experimental and simulated dwell times match perfectly but the two closed times deviate significantly. Please comment on this discrepancy.

23. Figure E11h: The experimental ATP turnover rate is 15/min (or 0.25/s), corresponding to one ATP hydrolyzed every four seconds, a much longer time than a gating cycle time of wildtype CFTR (~1 s). If, as proposed, multiple ATP hydrolysis cycles occur, the theoretical ATP hydrolysis rate must be even higher than 1/s. As stated, there is a likelihood that this biochemical measurement is not very accurate because of imperfect protein preparation. Alternatively, the proposed gating model is overly simplified. Either way, I wonder whether the simulation really provides more insights into the gating mechanism of CFTR. Sometimes it may be better to be vaguely right than precisely wrong. I leave this to the authors to decide.

References:

- Bompadre, S. G., Cho, J. H., Wang, X., Zou, X., Sohma, Y., Li, M., & Hwang, T. C. (2005). CFTR gating II: Effects of nucleotide binding on the stability of open states. *J Gen Physiol*, 125(4), 377-394. <https://doi.org/10.1085/jgp.200409228>
- Csanady, L., Mihalyi, C., Szollosi, A., Torocsik, B., & Vergani, P. (2013). Conformational changes in the catalytically inactive nucleotide-binding site of CFTR. *J Gen Physiol*, 142(1), 61-73. <https://doi.org/10.1085/jgp.201210954>
- Csanady, L., Vergani, P., & Gadsby, D. C. (2010). Strict coupling between CFTR's catalytic cycle and gating of its Cl⁻ ion pore revealed by distributions of open channel burst durations. *Proc Natl Acad Sci U S A*, 107(3), 1241-1246. <https://doi.org/10.1073/pnas.0911061107>
- Jih, K. Y., Sohma, Y., Li, M., & Hwang, T. C. (2012). Identification of a novel post-hydrolytic state in CFTR gating. *J Gen Physiol*, 139(5), 359-370. <https://doi.org/10.1085/jgp.201210789>
- Lin, W.-Y., Jih, K.-Y., & Hwang, T.-C. (2014). A single amino acid substitution in CFTR converts ATP to an inhibitory ligand. *The Journal of General Physiology*, 144(4), 311-320. <https://doi.org/10.1085/jgp.201411247>
- Lin, W. Y., Sohma, Y., & Hwang, T. C. (2016). Synergistic Potentiation of Cystic Fibrosis Transmembrane Conductance Regulator Gating by Two Chemically Distinct Potentiators, Ivacaftor (VX-770) and 5-Nitro-2-(3-Phenylpropylamino) Benzoate. *Mol Pharmacol*, 90(3), 275-285. <https://doi.org/10.1124/mol.116.104570>
- Tsai, M. F., Li, M., & Hwang, T. C. (2010). Stable ATP binding mediated by a partial NBD dimer of the CFTR chloride channel. *J Gen Physiol*, 135(5), 399-414. <https://doi.org/10.1085/jgp.201010399>
- Vergani, P., Lockless, S. W., Nairn, A. C., & Gadsby, D. C. (2005). CFTR channel opening by ATP-driven tight dimerization of its nucleotide-binding domains. *Nature*, 433(7028), 876-880. <https://doi.org/10.1038/nature03313>
- Vergani, P., Nairn, A. C., & Gadsby, D. C. (2003). On the mechanism of MgATP-dependent gating of CFTR Cl⁻ channels. *J Gen Physiol*, 121(1), 17-36. http://www.ncbi.nlm.nih.gov/entrez/query.fcgi?cmd=Retrieve&db=PubMed&dopt=Citation&list_uids=12508051
- Yeh, H. I., Sohma, Y., Conrath, K., & Hwang, T. C. (2017). A common mechanism for CFTR

potentiators. *J Gen Physiol*, 149(12), 1105-1118. <https://doi.org/10.1085/jgp.201711886>

Yeh, H. I., Yeh, J. T., & Hwang, T. C. (2015). Modulation of CFTR gating by permeant ions. *J Gen Physiol*, 145(1), 47-60. <https://doi.org/10.1085/jgp.201411272>

Yeh, H. I., Yu, Y. C., Kuo, P. L., Tsai, C. K., Huang, H. T., & Hwang, T. C. (2021). Functional stability of CFTR depends on tight binding of ATP at its degenerate ATP-binding site. *J Physiol*, 599(20), 4625-4642. <https://doi.org/10.1113/JP281933>

Yu, Y. C., Sohma, Y., & Hwang, T. C. (2016). On the mechanism of gating defects caused by the R117H mutation in cystic fibrosis transmembrane conductance regulator. *J Physiol*, 594(12), 3227-3244. <https://doi.org/10.1113/jp271723>

Zhou, Z., Wang, X., Liu, H. Y., Zou, X., Li, M., & Hwang, T. C. (2006). The two ATP binding sites of cystic fibrosis transmembrane conductance regulator (CFTR) play distinct roles in gating kinetics and energetics. *J Gen Physiol*, 128(4), 413-422. <https://doi.org/10.1085/jgp.200609622>

Referee #3 (Remarks to the Author):

The manuscript by Levring et al describes characterization of molecular cycles involved in activation of the CFTR channels. Specifically, single molecule FRET studies done in parallel with single channel electrophysiology enable to characterize the relationship between ATP binding, hydrolysis and channel opening. Furthermore, how disease-causing mutations and therapeutic correctors modulates these events are studied and discussed.

The study is very nicely crafted, the experiments are very well performed, and the conclusions are convincing.

This work brings a clear view on a long-standing issue in the CF field and deserves publication.

Nevertheless, a number of points require clarification.

1/ FRET values

CryoEM structures of human CFTR have -so far- captured 2 conformations with the NBDs either separated or dimerized. Structural data on ABC structures suggest more extended conformational diversity with several different substates. The differences in FRET values observed in the present manuscript may be indicative of such diversity.

Therefore, the authors need to clearly assign -when possible- the link between the FRET value and the identified conformer .

The ATP-free CFTR shows a low FRET state at 0.25. Does this match the cryo-EM structures of the NBD separated state ? Similarly does the 0.49 FRET value match the NBD-dimerized state for the used probes?

The authors need to run careful simulations (for examples with FPS, see Kalinin et al) with the appropriate probe models to evaluate the calculated FRET value for each conformation.

ATP-bound dephosphorylated CFTR shows a 0.28 FRET value. How does that translate in terms of

distance change given the R0 for these probes ?

The corresponding cryo-EM structure showed in Figure E4 does not address these issues, in great part due to the limited resolution. In fact, the authors only vaguely describe “proximal changes in local structure”. What does that mean? In my opinion in its current state this structure does not bring relevant information and should be either improved or removed from the manuscript.

2/ The E1371Q mutant appear to be locked in the high FRET state during the time course of the experiments, at least in the trace shown (100s, Fig1f). However, the distribution on Fig1c seems to show a significant low FRET population (~5-10%), pretty much identical to that seen for the “wt” CFTR (which can hydrolyze ATP and thus should spend more time in the NBD separated). In that line, it is unclear to me why the authors see no NBD separation in this mutant after phosphatase treatment (Fig E3). One may expect to see some separation over the course of the measurement.

In the same Figure, can the author comment on the difference in ATP concentration dependence of channel gating vs NBD dimerization? At 3mM ADP 80% of the channels are still in dimers but gating is almost gone.

3/ Very little is shown regarding the membrane-embedded measurements. What does the sentence “While the fraction of the molecules responsive to ATP was significantly reduced” refer to ? Only single traces are shown. No statistics (“significantly” ?).

4/ ATP binding sites

The authors use the W401A and Y1219A mutants originally described by Hwang and colleagues (although mutated into Gly in that study). What do the authors mean by “destabilize ATP binding” ?

It should be noted that the mutations decrease affinity but may not prevent it. For example, W401A is still able to bind ATP (albeit with lower affinity) and 3mM is quite a large concentration.

Perturbing the ATP binding at the degenerate site may also alter the conformational equilibrium of NBD1 that we recently described (Scholl et al, 2021)

On P8 the authors state that for mutant Y1219A “ATP binds only at the degenerate site”. This is not proven.

On P7, the authors describe that “the Y1219A variant, which binds ATP mainly at the degenerate site, transitioned between NBD-dimerized and separated states slowly, and very rarely exhibited channel opening” but they only show single traces. They should show distributions.

5/ Figure E3K is unclear to me. Is it expected to see such a significant effect at 0.1 and 0.3 mM ADP while 3mM ATP is maintained (and while Fig E3I show predominantly NBD in dimer form)? I assume that the last part of the graph has 0mM ADP (correct?). Why is the rebound much faster than what is observed during the first 2 mins?

6/ The coupling ratio seems unclear to me. What if the two would anticorrelate ? The coupling ratio would still be 1 ?

7/ P7 the authors state “We infer from these observations that channel opening probability is enhanced by interactions of CFTR with the terminal phosphate moiety of the ATP molecule bound within the consensus site.”. This should be readily testable as the terminal phosphate at the consensus site is coordinated by K1250 and T1246. Point mutations should support this statement.

8/ Mutation G551D

The authors show an intermediate FRET value of 0.37. The authors suggest that the mutation introduce a “steric and electrostatic repulsion at the NBD interface” which is very vague.

An Asp at position 551 will likely prevent binding of ATP at the consensus site (at least in a NBD dimerize state) because the acidic side chain would directly and closely face the terminal phosphate. This suggests that the NBD1-NBD2 dimer is not formed at the consensus site and that the protein is adopting an intermediate/different conformation. This also suggest that the ATP “stimulation” is due to dimerization at the degenerate site.

Therefore the “dimerized dwell time” on Fig E9E may actually not correspond to dimerized state as “expected”, ie. where both sites are closed, as observed by Cryo EM.

In addition, the statement on P10 that “Importantly, both the G551D and L927P variants exhibited ATP-dependent FRET response indicative of wild-type ATP binding affinities” is hard to reconcile with the position of D551 in the mutant

Furthermore, the effect of potentiators on G551D are not clear, and not discussed.

Ivacaftor was approved to treat G551D patients, we would expect to see an effect. The authors show some individual traces with GLPG1837, but not the approved drug Ivacaftor (why?) from which there seem to be no significant effect. The authors should explore and or discuss this.

In contrast, stimulation of the L927P mutation does lead to fully dimerized state which does seem to correlate with increase channel opening (although Fig 4f shows quite modest improvement).

Author Rebuttals to Initial Comments:

Point-by-point response

We sincerely thank the Reviewers for their time and constructive comments, which have helped to significantly improve the clarity and scholarship of the manuscript.

Reviewer #1 (Remarks to the Author):

The Levring et al. paper seeks to determine how dimerization of CFTR NBDs is coupled to ATP binding and hydrolysis, and to pore opening, using a combination of smFRET and electrophysiology. CFTR cryo-EM studies revealed two conformations: one in absence of phosphorylation and ATP with the pore-closed and NBDs separated by and the R domain sterically blocking their dimerization and the other phosphorylated with ATP bound and NBDs dimerized with two ATP molecules bound at their interface. The goal here was to elucidate the relationship between NBD dimerization and gating, understand how ion permeation is coupled to ATP hydrolysis and NBD isomerization and gain insight into the effect of disease-causing mutations and pharmacological modulation for therapeutics. The manuscript presents a detailed molecular explanation for how conformational change and channel opening are coupled and provides insight into modes for potentiator action. The authors do a commendable job of integrating multiple types of biophysical and functional data. The writing is clear (in most places), the experiments are well designed and thoughtfully interpreted. The study contributes significantly to the field. Several issues of data presentation and interpretation should be addressed.

Major comments:

1) The conclusions hinge on quantitative comparisons between smFRET data and electrophysiology data. Some clarifications are needed. The properties of CFTR are compared to the unlabeled CFTR_{FRET} in E1b-g and the fluorophore-labelled version in Fig. E1i-k and E2. For Fig. 1 and 3 it is not clear whether this is the labeled or unlabeled version. The authors should clarify this in text, Methods and legends in a consistent manner. A clear nomenclature would help.

We revised the text, figure legends, and methods to clearly indicate which CFTR construct was used in every experiment.

If Figs. 1 and 3 use the unlabeled version, authors should justify why. Also, authors should note if the constructs being compared differ in possessing a C-terminal GFP.

Macroscopic current measurements using inside-out patches (e.g. **Extended Data Figure 1b-g**) were not labeled with fluorophores because efficient and specific labeling of CFTR cannot be achieved in the heterogeneous cellular membrane. C-terminally GFP-fused CFTR were used in these experiments as it has been shown previously that C-terminal fusion with GFP does not affect CFTR gating (https://bpsbioscience.com/media/wysiwyg/60506_3.pdf). In our revised manuscript, we now provide further evaluations of fluorophore-labelled CFTR_{FRET} that support our conclusion that it recapitulates wild-type CFTR behaviour (see **Extended Data Figure 1**). Site-specifically labeled CFTR_{FRET} solubilized in detergent (**Figure E2a-b**) hydrolyzed ATP with

41 a rate nearly identical to that of wild-type CFTR (**Figure E1h**). Fluorophore-labelled CFTR_{FRET}
and wild-type CFTR (without fluorophore labels) reconstituted into synthetic planar lipid bilayers
also exhibited similar current–voltage relationships, open probabilities, and responses to
GLPG1837 (**Figure E1i-m**).

In E1i-j the current–voltage relationships suggest that CFTR_{FRET} has higher conductance
(authors should indicate how many patches went into this comparison). In addition, missing is a
comparison of open probability, which is critical for the interpretation.

We have now discussed the higher conductance on page 4 “*Single-channel conductance of*
*fluorophore-labelled CFTR_{FRET} was slightly higher (Figure E1i-j), possibly due to the C343S*
*substitution, a residue bordering the pore.*” This difference does not affect interpretation of single-
molecule FRET measurements, as it does not relate to the dynamics of the nucleotide-binding
domains. We also noted that the data points in **Extended Data Figure 1j** reflect means and
standard errors of 3-18 channels reconstituted in synthetic planar lipid bilayers.

Per the Reviewer’s suggestion, we determined the open probability of the fluorophore-conjugated
CFTR_{FRET} variant channels and show that it is essentially identical to that of the wild-type,
unlabeled CFTR (**Extended Data Figure 1l**).

2) Even when the reader knows that patch recordings and FRET come from different experiments,
placing these side by side or one above the other could give the confusing impression that they
were measured simultaneously. This confusion is aggravated by sentences like this in the text (line
206): “The W401A variant, which is capable of binding and hydrolyzing ATP via the consensus
site, underwent rapid transitions between NBD-separated and -dimerized states that correlated with
channel gating (Figure 2c).” This sentence should be changed (e.g. to “more closely resembled,”
not correlated). To further clarify this, authors should add to Fig 2 a panel showing dimerization
probability and compare open and dimerized dwell times the way they did in WT channels in Fig
1. Authors should also define for panel f ratio between probabilities of opening and dimerization
which is the numerator.

We thank the Reviewer for pointing out this confusion and we have now revised the text to clearly
indicate that channel recordings and FRET were two different experiments. For example, on page
7: “*Relative to wild-type CFTR, the W401A variant, which is capable of binding and hydrolyzing*
*ATP at the consensus site, underwent comparatively rapid transitions between NBD-separated*
*and -dimerized states that **more closely resembled** the dynamics of pore opening **measured in***
*electrophysiological recordings*”. And: “*By contrast, the Y1219A variant, which binds ATP*
*principally at the degenerate site, slowly transitioned between NBD-dimerized and separated*
*states ... **single-channel measurements** of the Y1219A variant exhibited only sporadic opening*
*events*”. On page 8: “*Here, we **separately tracked** the time-courses of NBD dimerization and*
*macroscopic current increase upon application of saturating ATP (3 mM)*”.

As requested, we also added a panel quantifying dimerization probability of the variants (**Figure**
**2f**), and a panel comparing the open and NBD-dimerized dwell times for the W401A CFTR variant
(**Figure 2d**). The Y1219A variant has an open probably of nearly zero and gates slowly, preventing
82 us to quantify its open dwell time.

The definition of coupling ratio is now clarified in the **Figure 2** legend as: “open probability
divided by dimerization probability.”

3) The conclusion of Fig. 3 is that NBD dimerization precedes channel opening. The average rise
upon rapid ATP perfusion of the phosphorylated channel of high FRET occupancy (dimerization)
in smFRET and of channel opening in patch is faster for dimerization. Quantification was by
mono-exponential fits. These should be superimposed on the data. Tau of dimerization is shown
as <100 ms. The precise value and SEM should be given.

As requested, we now superimposed mono-exponential fits on the individual time-courses as
**Extended Data Figure 7a-b** in the revision. In **Figure 3a** we reported the means and standard
errors of $\tau_{opening}$ from 42 inside-out membrane patches. We did not specify the relaxation time
constant for dimerization as the measured rate may be limited by mixing time and therefore not
inform on the biologically relevant transition. The fitted values for $\tau_{Dimerization}$ (~100 ms) are on the
same scale as measured rates of mixing ($\tau_{Exchange} = 115$ ms, see below), indicating the measured
dimerization response may be limited by the rate of solvent mixing in the fluorescence microscope.
For these reasons, we believe it is appropriate to report the upper bound on the rate of dimerization
instead of a precise value.

Fig. E7 is supposed to demonstrate that differences in kinetic behavior seen in Fig 3 are not due to
differences in solute exchange between the two sets of experiments, but this figure only displays
e-phys data. What is the equivalent dataset for smFRET, and where is it displayed? The Tau of
patch solution exchange in Fig. E7 is 150 ms, slower than the <100 ms value given for smFRET
in Fig. 3a, so it is unclear how well the exchange rates actually match.

We performed additional experiments to measure the solute exchange within the microfluidics of
our single-molecule imaging microscope (**Extended Data Figure 7e**). The estimated exchange
rate is $\tau_{Exchange} = 115$ ms. We revised the text on pages 8-9 to include this new data: “The rate of
channel opening ($\tau_{opening} = 490 \pm 40$ ms) (**Figure 3a, E7a**), was approximately 3-fold slower than
the solvent exchange rate of the perfusion system ($\tau_{exchange} \sim 150$ ms) (**Figure E7c,d**). By contrast,
the fitted rates for NBD dimerization from FRET measurements ($\tau_{Dimerization} \sim 100$ ms) were on the
same scale as the solvent exchange rate in the fluorescence microscope ($\tau_{exchange} = 115$ ms) (**Figure**
**E7b,e**). Thus, the observed delay in current activation could not be ascribed to differences in rates
of mixing of the two experimental methods. We therefore conclude that the observed delay reflects
conformational changes within the NBD-dimerized state that precede channel opening...”

The authors state: “We conclude ... that the mean first passage time between NBD dimerization
and channel opening in productive instances is approximately 500 ms.” However, their Tau of
opening is 490 ms and they begin to see channels opening almost immediately. The delay they
seek is the one between dimerization and opening and would be a smaller value.

Given the imprecision of our estimate of the rate of the initial dimerization step ($\tau_{Dimerization} < 100$
121 ms), we cannot confidently deconvolve the individual sequential rates that lead to the pore open
state. However, the Reviewer is correct that the delay time must be corrected relative to $\tau_{Opening} =$
490 ms. We modify the text in the revision to include this consideration (page 9): “... the mean
first passage time of this process is approximately 400-500 ms.”

Minor comments:

1) E1j is mistakenly referred to as E1k in the text (line 161)

Corrected.

2) All electrophysiology traces should have current amplitude scale bar (e.g. missing in Fig. 2c)

Corrected.

3) Figure 1. E1371Q should be shown without ATP as well.

We agree that addition of panels reflecting ATP-free E1371Q CFTR would nicely demonstrate
retention of ATP-dependence for this variant. However, due to the slow rate of non-hydrolytic
ATP dissociation we have found it difficult to isolate a homogenous population of ATP-free
E1371Q CFTR molecules. Incubation with Apyrase for 1 hour at 20 °C was insufficient to produce
a uniformly low-FRET (NBD-separated) population (**Figure R1**).

**Figure R1 | Nucleotide-free E1371Q CFTR**

**a.** Contour plots of phosphorylated E1371Q CFTR. CFTR was incubated with 3 mM ATP (left), treated with Apyrase for 1 hour
at 20 °C in nucleotide-free buffer (middle), and 3 mM ATP was then reintroduced (right). **b-c.** Example single-molecule traces for
E1371Q CFTR in ATP-free buffer (**b**) and upon rapid delivery of 3 mM ATP (**c**).

Nonetheless, these experiments confirm that dimerization of E1371Q is ATP-dependent and
reversible, and are consistent with electrophysiological recordings. However, as we have been
unable to perform ‘ATP-free’ measurements of E1371Q CFTR at steady state, we believe that
addition of such panels to **Figure 1** may confuse the reader.

4) No mass spec data to demonstrate that phosphorylation is consistent/present across all
conditions.

In electrophysiological measurements protein kinase A (PKA) was applied until current amplitude
saturated (see e.g. **Figure 1b**). This state should be consistently phosphorylated.

In cases where a real-time read-out is not available such as for ATP hydrolysis or single-molecule
FRET measurements, CFTR was incubated with PKA and ATP for 30-60 minutes at room
temperature. We have experimentally determined that under these conditions the response to PKA-
phosphorylation saturated within 1-2 minutes (**Figure 1d**).

Similarly, CFTR was treated with Lambda protein phosphatase for 30 minutes at room temperature
to dephosphorylate the R-domain. Under these conditions the response to phosphatase also
saturated within 1-2 minutes (**Extended Data Figure 3r**).

While a minor population of molecules may be refractory to kinase or phosphatase treatment, we
would expect for this to be consistent across experiments. Mass spec quantification of the
phosphorylation status in each experiment would be a significant undertaking and beyond the
scope of the present study.

5) Figure 2. 2cd ATP concentration and phosphorylation status should be stated in legend for
clarity. It is also not so easy to determine by eye the contribution of open probability and
dimerization to the coupling ratio by looking at 2e and 2f. It would help to additionally show the
dimerization probability for the five different variants.

We have now specified ATP concentration to be 3 mM in the legend of **Figure 2**. We also add a
panel that shows the dimerization probability for the nucleotide binding site variants (**Figure 2f**).

6) Example traces in E1i look different at lower membrane potentials (-90 and -150)-- is there a
better representative trace or what is the hypothesized reason for this difference?

We have updated the single-channel traces of the current-voltage relationship in **Extended Data**
**Figure 1i** to better reflect open probabilities of the CFTR_{FRET} variant.

7) Analogous examples of current from WT channels should be shown for comparison in E1b
and E1k.

Example current traces of wild-type CFTR have been added to **Extended Data Figure 1b and 1k**.

8) Line 140 NDB typo

Corrected.

9) A question regarding comparison of electrophysiology and FRET data-- are there any integral
membrane phosphatases which could contribute to the observed differences between channel
opening and NBD dimerization probability?

Channel open probabilities were determined using purified CFTR reconstituted into synthetic lipid
bilayers. NBD dimerization probability was determined by FRET measurements performed with
purified protein. Neither system contains cellular phosphatases.

10) The authors should make clear that disease-causing CFTR variants have multiple
modes/effects, and should describe the direction in which a pharmacological agent should act (i.e.
promote channel opening as opposed to promoting channel closure, etc). On p. 9, the effects of
G551D and L927P on channel gating should be described first, before explaining effects on FRET-
based readouts.

As suggested, the channel gating and ATPase activities of the G551D and L927P mutants are now
described before the FRET data. The effects of the potentiators are now specified as to “*promote*
*channel opening*” (pages 10-11).

11) Presence of a native lipid membrane or detergent environment may affect the dynamics of the
channel. To address this, the authors perform a set of FRET experiments in proteoliposomes, which
should more accurately approximate a cell membrane. The FRET dynamics and dependence on
ATP look similar to what they see in detergent, bolstering confidence in their experiments. One
odd thing is that only a small fraction of channels are ATP sensitive. They attribute this to channel
orientation within the bilayer. If the 10xHis tag is located on the NBD side of the protein, it is
unclear why any immobilized molecules should have inward-facing NBDs which are inaccessible
to ATP.

Immobilization of proteoliposome-reconstituted CFTR through the C-terminal His-tag (**Figure**
**R2a**) was designed to select CFTR molecules with the ‘NBDs-out’ orientation, as the Reviewer
correctly specifies. This orientation is necessary for sensitivity to externally applied membrane
impermeable reagents.

In practice, proteoliposome experiments with CFTR proved challenging for a number of reasons.
First, although CFTR was reconstituted at a low protein-to-lipid ratio, aiming to produce vesicles
with zero to one CFTR molecules incorporated, the sample may also contain vesicles containing
more than one CFTR molecule. In these cases vesicles may be immobilized via a non-fluorophore
conjugated CFTR molecule with the ‘NBDs-out’ orientation, while FRET is being measured by a
passenger molecule with the ‘NBDs-in’ orientation. The proteoliposome reconstitution procedure
may also negatively impact CFTR folding and function so as to produce inactive protein within
the bilayer.

We also found that nearly half of our proteoliposomes tethered within microfluidic chambers were
unable to be released by imidazole treatment (**Figure R2b**). By contrast, near-quantitative release
from the surface with imidazole was demonstrated with the detergent-solubilized sample
(**Extended Data Figure 2g**). We have previously observed that anionic lipids (e.g. POPS) may
associate with Ni²⁺-coated surfaces and the POPS content of our vesicles may contribute to this
non-specific association issue. The extent to which such issues relate to losses in CFTR activity in
this context is not presently known.

Approximately half of the immobilized and phosphorylated CFTR channels displayed sensitivity
to externally applied ATP (**Figure R2c**). Consistent with a permanent loss of CFTR activity
associated with the proteoliposome reconstitution or immobilization process, the responsive and
non-responsive molecules did not dynamically interconvert within the time-scale of imaging.

These complications were a component of decision to focus our quantitative analysis on digitonin-
solubilized CFTR.

**Figure R2 | Proteoliposome-reconstituted CFTR**

**a.** Schematic drawing of the immobilization strategy for proteoliposome-reconstituted CFTR. **b.** Quantification of the specificity
 of His-tag-dependent immobilization. **c.** Contour plots showing the response of proteoliposome-reconstituted CFTR molecules to
 rapid application of 3 mM ATP. Proteoliposomes were kinase-treated before the experiments.

12) It is not clear if the kinetic model adds much to the discussion, but it also is presented with
 appropriate caveats. Does the model make a prediction in a way that could be used to test the
 alternative hypotheses presented in the introduction of the paper?

We acknowledge that the model is simplified and therefore cannot recapitulate all experimental
 observables. We believe having a kinetic model is useful to tie all experimental observables
 together and to guide future experiments for the following reasons:

 Establishing the basic topology of the CFTR gating mechanism is a fundamental goal as it provides
 an essential framework for kinetic modeling, the context for understanding the impact of mutations
 and drug action and a physical guide for establishing and testing hypotheses about function.

 The prevailing kinetic frameworks available to the field only included states that could be directly
 discerned from electrophysiological data. Parallel assessment of smFRET dynamics and
 electrophysiology demanded additional considerations and constraints to the available kinetic
 information and thus a new kinetic model had to be established that parsimoniously recapitulates
 the major kinetic and structure features of CFTR function revealed by both methodologies as well
 as the estimated ATP hydrolysis rate based on ensemble measurements. Its inclusion ties all
 available information together into a single kinetic framework with a defined topology and is thus
 an essential message of our investigation.

 Kinetic models of function are also critical to scientific advancement because they provide a
 critical framework for others to challenge and test through new experiments. For instance, our
 findings suggest that the kinetic model presented explains the majority of the data obtained, but
 that additional complexities associated with ATP binding and hydrolysis in the consensus site alone
 should likely be considered in future work. The kinetic framework presented also points to the
 need for mechanistic clarifications regarding “flicker” open/closed states that have been evidenced
 in the field for decades. Prior investigations have suggested that such processes reflect “flicker
 closing” events from otherwise open channel states that reflect clogging of the pore by solutes. In
 other words, these events are kinetically “off-pathway”. While we cannot yet be conclusive in this
 regard, the kinetic framework proposed leaves open the possibility that “flicker” events may

instead reflect “flicker open” events that are “on-pathway” barrier crossing attempts prior to ATP
hydrolysis.

Both sets of considerations, together with the identification of rate-limiting intramolecular
conformational changes within CFTR that govern gating kinetics that is defined by the kinetic
framework, will be vital for the field to consider in future investigations.

13) Figure 1i shows that a very large percentage of channel openings have a dwell time of at least
10 ms, which could be detected using smFRET. Though it may not be feasible for this study, it
would be highly convincing if it were possible to design a FRET sensor of channel pore activity
that approximates the electrophysiological readout? This would allow direct comparison within a
single controlled system.

We share the Reviewer’s ambition to directly correlate the conformational transition of CFTR with
a functional readout of pore opening. Making such experiments feasible is certainly an exciting
direction to pursue in the future.

**Reviewer #2 (Remarks to the Author):**

The current manuscript by Levring et al. presents incredibly comprehensive studies of CFTR
function as an ATP-gated ion channel; these studies include electrophysiological recordings of
CFTR channel activity, real-time monitoring of the dimerization of CFTR's two nucleotide
binding domains (NBDs) with smFRET imaging, biochemical measurements of ATP hydrolysis
rate, and computer simulations based on a proposed gating model. Here are the main findings and
their mechanistic implications:

1. The NBDs are readily dimerized by ATP binding to the two ATP-binding sites in fully
phosphorylated CFTR, but the dimerized state (high FRET) is much more stable than the duration
of either channel open state or the closed state at a saturating concentration of ATP. This
uncoupling between NBD dimerization and channel gating suggests that multiple cycles of gate
opening and closing can occur with NBDs remained dimerized (inferred from high FRET),
contradicting previous idea of a strict coupling of NBD dimerization to gate opening and
hydrolysis-triggered NBDs separation (low FRET) to gate closing (Csanady et al., 2010).

2. By manipulating ATP binding to each site (Y1219A for the consensus site and W401A for the
degenerate site), they provided evidence for NBD dimerization by ATP occupancy at either site.
While both NBD dimerization and gating were mostly preserved in W401A, Y1219A channels
showed robust NBD dimerization but little activity, supporting previous reports that gating of
CFTR is mainly through ATP binding to the consensus site (Vergani et al., 2003; Zhou et al.,
2006).

3. By comparing the rate of NBD dimerization and the rate of channel activation upon sudden
addition of ATP, they found that NBD dimerization takes place much faster than gate opening,
contradicting previous proposition that NBD dimerization constitutes the rate-limiting step in
CFTR gating (Vergani et al., 2005). Although NBD dimerization precedes gate opening, NBD
dimerization itself is not sufficient to open the gate because dimerization (high FRET) can be seen
in conditions when there is little channel activity (e.g., upon ATP washout, or in the presence of
inhibitory concentration of ADP).

4. Supporting the idea that ATP at the consensus site plays a critical role in opening the gate, the
authors demonstrated that the disease-causing mutant G551D at the consensus site exhibits
defective gating and NBD dimerization. Interestingly, the dimerized NBDs in G551D show FRET
between high and low FRETs, suggesting the presence of an intermediate NBD dimer proposed
previously (Tsai et al., 2010). Similar but not identical intermediate FRET was seen in another
pathogenic mutation L927P (in TM8), which is 50 Å away from the consensus site. This latter
observation suggests a long-distance effect of the L927P mutation on NBD dimerization, but the
nature of this allosteric effect is unclear.

5. By testing the effects of CFTR potentiators ivacaftor and GLPG1837, they found a dramatic
gating effect with minimal and moderate changes of NBD dimerization in G551D and L927P
variants respectively. These are interesting observations, but it seems that not much additional
insight can be gained (see below).

6. These results were put together to form the basis of a kinetic model, which was further examined
through computer simulations. Although the proposed kinetic mechanism recapitulates many of
the experimental data, some simulated results do deviate from actual data.

No doubt, these are important studies that reveal many unexpected phenomena, which lay a solid
foundation for settling the debates over the coupling mechanism between NBD dimerization and
gate opening, as well as for advancing our overall understanding of the structural-functional
relationships of CFTR gating. This reviewer applauds the authors for such a remarkable
attainment.

We thank this Reviewer for their concise, thorough, and positive summary of our investigations.

Major comments:

1. The two conformations of NBDs—separated versus dimerized—can be resolved by the shift of
FRET from 0.25 to 0.49 (Fig. 1). In Fig. 1e, the FRET signal remains stable at 0.49 at 10 mM ATP
while the electrophysiological recording shows repeated opening and closing of the WT-CFTR
channel. The authors thus concluded that the NBDs remain dimerized throughout multiple rounds
of ATP binding/hydrolysis (Fig. 5). Conceptually, the consensus site in the “dimerized NBDs”
must open up a space wide enough to allow the exchange of hydrolytic products for the next ATP.
Indeed, in ATP washout (and Mg addition) experiments (Fig. 3b, c), transition from high FRET to
low FRET was observed immediately upon ATP removal (or Mg addition). As the waiting time is
too short for ATP dissociation from the degenerate site, this result suggests that upon ATP
hydrolysis at the consensus site, disengagement of the two NBDs does occur. The reason this high
to low FRET transition is not seen in the continuous presence of millimolar ATP may be due to
rapid rebinding and re-dimerization of the disengaged NBDs at millimolar ATP. This scenario
supports the notion that the state “degenerate site ATP, pore closed” should assume a different
dimeric state with a cracked consensus site to allow step 6 (Fig. 5) to take place. The canonical
NBD dimer in the cryo-EM structure leaves no room for ATP to diffuse into (or out of) its binding
pocket.

The CFTR gating cycle involves hydrolytic turnover and nucleotide exchange in the consensus
ATP binding site. We agree with the Reviewer that nucleotide exchange cannot occur from the
conformation adopted by phosphorylated and ATP-bound E1371Q CFTR (PDB 6MSM). In this
state the ATP is occluded from solvent. The absence of observable FRET dynamics in the ATP
saturated state and the monotonic increase of the dwell time of the high FRET state with ATP
concentration supports that ATP may rebind without complete separation of the NBDs. We posit
that small conformational changes occur after ATP hydrolysis that are sufficient to allow inorganic
phosphate and ADP release while the NBD’s remain ostensibly dimerized. As noted in our
discussion (page 12): “*The structure of the NBD-dimerized CFTR¹⁴ suggests that only small*
*changes at the consensus site, such as disrupting the hydrogen bond between R555 and T1246¹⁵,*
*would be sufficient for inorganic phosphate, ADP release and ATP rebinding.” Given the absence*
*of additional structural information on such processes, we may only conclude that the magnitude*
*and rate of such changes are not detectable with the specific structural perspective that CFTR*
*folding and biochemistry enabled us to achieve.*

To clarify this point we revised the text on page 6: “*These findings suggest that FRET-silent*
*processes occur within the NBD-dimerized conformation that trigger channel opening and closure*
*and that only subtle rearrangements at the dimer interface are required for nucleotide exchange.*”

The legend for **Figure 5** was also revised to clarify the need for conformational change to allow
for nucleotide exchange. See also our response for point 4a.

2. I wish the authors would have elaborated the rationale behind choosing positions 388 and 1435
for their FRET pair. T388 is at a position without a concrete secondary structure. Wouldn't this
positioning result in some uncertainty in FRET measurements as the FRET efficiency is
determined by not only the distance but also the relative orientation between the donor and the
acceptor? It would be nice if another pair is chosen to replicate some of the key observations
described above.

As CFTR has intrinsic folding/stability issues, the identification of structurally informative and
functionally tolerable sites of labelling was a substantial undertaking. These efforts were not
elaborated in the present manuscript due to space constraints and because it was noncritical for the
conclusions drawn in the manuscript. We briefly summarize the selection of labelling sites below:

Based on extant structures of dephosphorylated and ATP-free wild-type CFTR (PDB 5UAK) and
phosphorylated and ATP-bound E1371Q CFTR (PDB 6MSM) we measured the separation of
alpha-carbons for all structurally resolved residue pairs in CFTR. Using the Förster radius of the
LD555/LD655 FRET pair, we predicted FRET efficiencies in both conformational states, as well
as Δ FRET upon conformational isomerization (**Figure R3**). From residue pairs with favourable
predicted FRET efficiencies, we selected a subset of residues with solvent-exposed sidechains to
ensure efficient labelling with maleimide-conjugated fluorophores. Residues mutated in cystic
fibrosis patients or reported to affect folding or gating in the literature were excluded.

A panel (40) of cysteine-substituted CFTR variants were tested for whether they expressed, folded,
site-specifically labelled with maleimide-conjugated fluorophores, and retained key functional
properties of the wild-type channel. Several (5) pairs were experimentally tested for interpretable
FRET responses. This led to the final choice of labelling positions 388 and 1435. While T388 is
not involved in secondary structure, the density corresponding to the backbone of this residue is
well-defined in maps of both the ATP-free and dephosphorylated CFTR, as well as the ATP-bound
and phosphorylated E1371Q CFTR. Consistent with this, the 388/1435 pair exhibited tight
distributions of FRET efficiencies in conditions corresponding to those used for structure
determination.

 **Figure R3 | Residue selection.** Heat map of predicted changes in FRET efficiencies upon conformational isomerization for all
 possible pairs of labelling in CFTR.

 3. The W401A and Y1219A mutations were designed to weaken ATP binding. The authors
 observed that the NBDs of W401A and Y1219A mutants can still dimerize, and thus concluded
 that ATP binding at either site is sufficient for NBD dimerization (P7, line 215). Such conclusion
 is based on the assumption that the degenerate site in W401A (or the consensus site in Y1219A)
 must be vacant at 3 mM ATP. However, in electrophysiological study, the Y1219G mutation
 reduces the apparent affinity for ATP to a $K_{1/2}$ of ~5 mM (Zhou et al., 2006). If we assume that
 Y1219A affects the affinity similarly, at least a portion of the Y1219A channel population in Fig.
 2 would still have ATP bound to the consensus site at 3 mM ATP. Given that the degenerate site
 has much higher affinity for ATP, it is unlikely that the degenerate site in W401A would be vacant
 at 3 mM ATP. Thus, the data in Fig. 2 only support that NBDs of W401A and Y1219A can
 dimerize. The occupancy of ATP at each site cannot be determined, and thus whether a single ATP
 is enough to trigger dimerization cannot be inferred conclusively. However, the conclusion may
 not be incorrect once data in Fig. 1j are taken into consideration. The ATP concentration
 dependence of high to low FRET transition rate suggests the existence of different states of NBD
 dimer with one or both sites occupied.

 We agree with the Reviewer that the mutations may not completely abolish ATP binding and we
 revised the text as that the mutations were designed “to *reduce the affinity for ATP*” and “*the*
 *Y1219A variant, which binds ATP principally at the degenerate site*” (page 7). However, the
 fraction of such a dual ATP-occupied mutants is small and its presence will not alter the
 conclusions we draw, for the following reasons:

1. The bulk ATPase activity of the Y1219A variant at 3 mM ATP is reduced to only 15 % of the wild-type CFTR (**Figure 2b**). This indicates that the Y1219A substitution nearly abolished ATP binding at the consensus site.
 2. The open probability of the Y1219A variant is reduced to nearly zero, also suggesting that the substitution nearly abolished ATP binding (**Figure 2f**).
 3. The NBD-dimerization probability of the Y1219A variant is approximately 50 % of the wild-type level, consistent with ATP binding being perturbed. Yet the relative reduction in dimerization probability is much less than the effects on ATP hydrolysis and pore opening. These observations are consistent with the degenerate site ATP supporting dimerization

events, that in the absence ATP occupancy in the consensus site do not lead to pore opening
or ATP hydrolysis (**Figure 2e**).

4. Most importantly, the FRET and gating dynamics of the mutants are completely different
from that of wild-type CFTR, indicating the mutated site is mostly vacant under the
experimental conditions (**Figure 2c**).

5. As the Reviewer points out, data in **Figure 1i** lends further support that NBD may dimerize
with one or both sites occupied.

4. Some of the kinetic steps in Fig. 5 need justification/explanation:

a. Dissociation/binding of ATP in the degenerate site happens when the NBDs are separated (step
7, 8), but dissociation of hydrolytic products and binding of ATP to the consensus site happen
when the NBDs are dimerized (step 4, 5, 6). Some intermediate conformations of NBDs need to
exist.

We agree with the Reviewer that intermediate conformations of NBD separation must exist.
Because we cannot explicitly measure transitions of CFTR to intermediate states or estimate the
lifetimes of inferred intermediates we have not included them in our simplified kinetic scheme in
**Figure 5**.

To clarify this point we revised the legend: “ATP rebinding may occur with subtle rearrangement
at the dimer interface (step 6) or with complete NBD-separation (step 8) to initiate a new gating
cycle.”

b. Why step 3 is drawn differently in Fig. 5 and Fig. E11? Also, why did the authors propose only
the open state hydrolyzes ATP, but not 2 ATP pore flicker-closed state or 2 ATP pore closed state?

In **Figure 5** we deliberately draw step 3 from bracketed channel conformations as we do not have
evidence allowing us to determine which of ‘2 ATP pore flicker-closed’ and ‘2 ATP pore open’
states are catalytically competent. We are also unaware of literature that might distinguish between
models allowing hydrolysis from both or either states. One might speculate that flicker closures
are local to the pore and therefore do not affect hydrolysis. As the rates of exchange between these
states is much greater than the rates into or out of these states, this ambiguity has negligible impact
on our kinetic simulations as direct connectivity to either state yielded indistinguishable results.

For the simulations presented, we explicitly specified the connectivity of the underlying model by
allowing only hydrolysis from the on-pathway, ‘2 ATP pore open’ state. We altered the depiction
of the kinetic scheme in **Extended Data Figure 11** accordingly. [REDACTED]

To our best knowledge, there is no evidence to support that ATP hydrolysis occurs without pore
opening. In our kinetic scheme, we aim only to outline the dominant gating path, deliberately
ignoring rare events, even those that we know to occur.

[REDACTED]

c. Is the flicker-closed state the same as the cryo-EM structure of E1371Q-hCFTR? It's been
known that many of the flickery events seen at negative membrane potentials are not likely real
functional "state". Because of their high voltage dependence (see Fig. E1i.), many of the events
are considered blocking of the pore by cytosolic large anions. To get the right kinetic parameter
for the true flicker-closed state, one has to analyze data from recordings at positive membrane
potentials.

The molecular nature of flicker-closure events, the origin of their voltage-dependence, and whether
the structure of human E1371Q CFTR (PDB 6MSM) reflects a flicker-closed state has yet to be
elucidated to our knowledge. As our kinetic scheme allows ambiguity in this regard, we believe
that detailed discussions on this specific point lie beyond the scope of the current study. For the

purpose of our gating model and simulations, we aim only to recapitulate the empirical observation
of flicker-closure events during the pre-hydrolytic period of the open burst.

504 d. There is a mistake both in the text and in the figure legend. Steps 6 – 8 represent transitions
among closed states; therefore, it is not a new gating cycle, which by definition should include a
closed event and an open event.

We thank the Reviewer for pointing this out. We mean to state that ADP dissociation (step 5) leads
to a dynamic intermediate that may isomerize (step 7) and to which ATP may bind to NBD-
separated (step 8) or NBD-dimerized states (step 6). A new gating cycle is initiated after ATP
binding (steps 1 and 2). We do not consider steps 6-8 a separate gating cycle. We have revised the
text as (page 13): *“Dissociation of ADP (Step 5) results in a dynamic intermediate to which ATP*
*can rebind (steps 6-8) thereby initiating another gating cycle.”* and the legend of **Figure 5** as *“ADP*
*dissociation (step 5) leads to a dynamically isomerizing intermediate (step 7). ATP rebinding may*
*occur with subtle rearrangement at the dimer interface (step 6) or with complete NBD-separation*
*(step 8) to initiate a new gating cycle.”*

Minor comments:

1. I am not sure I truly understand the meaning of the term “hierarchical” gating mechanism.

We use this word to emphasize our finding that NBD dimerization does not necessarily lead to
pore opening. Pore opening requires conformational changes within the NBD-dimerized channel,
governed by ATP hydrolysis.

2. P2, line 3: Do you mean electrophysiological properties?

Corrected.

3. P5, line 140: It should be “...to a stably NBD-dimerized state.”

Corrected.

4. P6, line 157: Previous electrophysiological studies of R117H-CFTR and zebrafish CFTR, as
well as the cryo-EM structure of E1371Q all indicate the presence of conductive and non-
conductive states with dimerized NBDs. Please more explicitly state the unique insight from these
data.

As the Reviewer has pointed out earlier, whether NBD dimerization is strictly coupled to gate
opening has been a long-debated topic. There are reports in the literature supporting both
arguments. Here we address this question by directly comparing the kinetics of NBD dimerization
with channel gating. Furthermore, our finding that CFTR spends most of the time in the NBD-
dimerized state is original.

5. P6, line 161: ...with ATP concentration (Figure 1j), not 1k.

Corrected.

6. P6, line 172: The ADP experiments are interesting in that the low FRET value seems different from that without ATP. Is it reproducible? If it is, doesn't that suggest different degrees of disengagement of NBDs?

The data is reproducible from three independent experiments. To investigate if this reflects global or local conformational changes of the NBDs would require cryo-EM analyses that are beyond the scope of this study.

7. P7, line 191: The time constant for ligand exchange at the degenerate site is around 30 s (Tsai et al., 2010), which interestingly is very similar to the duration of high FRET upon ATP washout (Fig. 3b).

Please see our response to point 10 below.

8. P7, line 226: It should be open probability, not opening probability. (In other parts of the text too.)

Corrected here and throughout the text.

9. P8, line 237: The changes of FRET upon sudden addition of ATP show heterogeneity. This observation may be related to a recent paper by Yeh et al. (2021). The stated similarity to W401A and double exponential current rising upon addition of ATP (Fig. 1b and E1b) all remind this reviewer of the results published in this paper, which shows different closed channel conformations (perhaps different NBDs conformations) once ATP is removed.

We thank the Reviewer for pointing this out. The different configurations of closed states described by Yeh et al. (2021) are clearly evident in our ATP washout experiments (**Figure 3b**) where molecules first release nucleotide from the consensus site, but retain ATP in the degenerate site which supports dimerization (**Extended Data Figure 8g**). Reversible rundown may correspond to release of degenerate site ATP leading to stably separated NBDs. We have now added the following sentence on page 9: "*Dissociation of ATP from both sites likely leads to the reversible rundown of CFTR currents that occurs after prolonged exposure to nucleotide-free solutions*³⁰"

The observed heterogeneity in the ATP injection experiment (**Figure 3a**) may have a different origin. At time of ATP injection CFTR molecules had been in ATP-free buffer for a prolonged period and both ATP binding sites should be vacant. To us, the simplest origin of heterogeneity arises from stochastic and sequential ATP binding to the individual binding sites (**Figure R5**). That is, the population is homogenous at time of injection. CFTR molecules that bind ATP in the consensus site first, may dimerize, hydrolyze ATP, release products, and relax back to the starting point. Repeated cycles of this may occur. Upon ATP binding in the degenerate site the NBDs dimerize stably. Eventually, every molecule reaches steady state with both binding sites simultaneously occupied.

Figure R5 | Sequential ATP binding to consensus and degenerate binding sites in the pre-steady state. Schematic of the events occurring after rapid ATP application to PKA-phosphorylated CFTR. If ATP is bound at the consensus site first, CFTR enters Gating cycle A. The NBDs of CFTR dimerize, the pore opens, ATP is hydrolyzed, and catalytic products are released. Repeated cycles may occur. Every passage through Gating cycle A involves complete NBD dimerization and separation. Eventually, CFTR molecules will bind ATP in both sites simultaneously, and gate at steady state (Gating cycle B). Gating cycle B involves only subtle rearrangements at the NBD interface.

10. P8, line 255: Of note, rapid removal of ATP caused a double exponential decay of WT-CFTR currents (Fig. 2B in Lin et al., 2014) with time constants ($\tau_1 < 1$ s and $\tau_2 = 29$ s) similar to these values reported for NBD dimerization relaxation. These shouldn't be just coincident.

We agree. The apparent correlation between the biexponential time constants for relaxation of pore opening and dimerization suggests a common underlying molecular mechanism determining both transitions.

To us, the simplest explanation is that CFTR relaxes in two steps associated with release of ATP from either site. Hydrolysis-catalyzed release from the consensus site is fast. Dissociation from the degenerate site is slow. The intermediate with ATP in the degenerate site alone behaves like the Y1219A CFTR variant, exhibiting high dimerization probability but low channel open probability. This interpretation is in line with conclusions made by Lin et al (2014).

The biphasic current relaxation upon nucleotide exchange reported by Tsai et al., 2010 (Reviewer comment 7) is also consistent with this model. Further, we note that the reduced rate of the slow exchange component reported by Tsai et al ($\tau_2 = 51$ s) compared to the slow decay component measured in washout experiments ($\tau_2 \approx 20$ -30 s) is consistent with our observations. During the second phase of washout, CFTR dynamically isomerizes between dimerized and separated conformations (**Figure 3b**). When NBDs separate, both the consensus and degenerate site are able to readily release nucleotide. In the ligand exchange experiment continuous presence of nucleotide stabilizes the dimerized conformation thereby reducing the rate of exchange at the degenerate site.

We revised the text to reference the correlation with the cited manuscripts (page 9): “*Parallel FRET experiments showed that the time course of NBD-separation is biphasic, with time constants of 1.6 seconds and 20 seconds, respectively (Figure 3b). These rates correlate with the double*

*exponential time constants reported for CFTR current decay and ligand exchange*^{18,29}. *This*
*apparent correlation suggests a common underlying molecular mechanism determining both*
*transitions.”*

11. P9, line 277: Not a correct reference and interpretation. The cited ligand exchange experiments
lead to two different turn-over rates of ATP and the idea of a “partial dimer.” On the other hand,
by carrying out different ligand exchange experiments, Jih et al. (2012) propose a re-entry pathway
featuring repetitive cycles of ATP turnover in a single opening burst. Both studies done by Hwang
and colleagues invoke a partial dimer state where the consensus site is open up to allow replenish
of ATP. They do not suggest that ATP turnover can occur in a single canonical NBD-dimerized
conformation, where the ligand is occluded.

We thank the Reviewer for pointing out the differences in the models. The shared concept of the
present inferences and the cited manuscript by Tsai et al., is that during wild-type CFTR gating at
steady state, the NBDs do not fully separate for each hydrolysis event and open burst. Rather they
remain associated through continuous ATP binding in the degenerate site. Nonetheless, nucleotide
exchange at the consensus site necessarily requires some degree of separation (see also discussion
above). Tsai et al. propose the formation of a partial dimer. By contrast, we do not detect changes
in FRET efficiency between repeated gating cycles and therefore infer that the changes required
for nucleotide exchange must be subtle in magnitude and/or transient in time. We revise the text
to reflect this (pages 9-10): *“These findings suggest that the probability of ATP rebinding exceeds*
*that of complete NBD separation when ATP concentration is greater than 100 μM. This concept,*
*consistent with ligand exchange experiments*¹⁸, *suggests that repetitive cycles of ATP turnover can*
*occur in an ostensibly NBD-dimerized conformation with only subtle changes at the consensus site*
*required for nucleotide exchange. In cellular settings, repetitive gating cycles are therefore*
*expected to persist until the finite rate of NBD separation at cellular ATP concentrations allows*
*the dephosphorylated R-domain to reinsert, terminating CFTR gating.”*

12. P10, line 312-314: I don’t understand what the authors mean by the similar nature in “the
similar nature of both mutations’ impacts on NBD dimerization and channel opening”. G551D and
L927P actually exhibit different behaviors in their experiments. In Fig. 4b, the transitions between
high and low FRET in G551D is different from those of L927P. Also, GLPG1837 drastically
changes L927P’s FRET pattern, but barely affects G551D. Moreover, electrophysiological
properties of these two mutants and pharmacological effects of CFTR potentiators should be
discussed in a more quantitative manner (see comment 18 below) before one can make more
definitive conclusions.

This section has been extensively revised based on this comment and Reviewer 1’s comments.
Thank you.

13. P11, line 347: It should be noted that both VX-770 and GLPG1837 work on a CFTR variant
with its NBD2 completely removed (Yeh et al., 2017; Yeh et al., 2015). These drugs also exert
dramatical gating effects on G551D whose NBDs may not assume a canonical NBD dimer state
(Lin et al., 2014; Yeh et al., 2017) (Fig. 4 in current study).

Thank you for pointing this out. We have revised the text to include this information. On page 12:
*“These data lead to the conclusion that the major effect of Ivacaftor or GLPG1837 is not to support*
*transition from NBD-separated to dimerized conformations. Rather, these potentiators principally*
*operate by promoting pore opening in CFTR when NBDs are already dimerized. In other words,*
*potentiators affect the coupling efficiency between NBD dimerization and channel opening,*
*possibly by stabilizing the TMDs in the pore open configuration⁴³. This effect also manifests in*
*variants unable to form a canonical NBD dimer, such as G551D and a variant devoid of the entire*
*NBD⁴⁰.”*

14. P11, line 375-377: Previous studies suggest that reagents such as NPPB and high-affinity ATP
analogs, act synergistically with Ivacaftor and GLPG1837 on G551D channels by promoting NBD
dimerization (Lin et al., 2016; Yeh et al., 2017). The authors may consider incorporating these
studies in the Discussion.

We included a discussion of the investigational compound NPPB in the revision (page 12): *“The*
*investigational compound 5-nitro-2-(3-phenylpropylamino) benzoate (NPPB) was proposed to*
*stimulate pore opening by such a mechanism⁴⁴.”*

15. Figure 1: I suggest the authors unify the spelling of “dimerization.” It is dimerization in the
text but dimerisation in Fig 1 and Fig 4.

Spelling has been unified as “dimerization”.

16. Figure 1e: If I understand correctly, the electrophysiological recordings are from WT-CFTR,
not from the CFTRFRET protein. Wouldn't it be a better comparison to use the
electrophysiological recording of CFTRFRET protein here? Of note, the CFTRFRET construct
altered the majority of the endogenous cysteine residues; this alteration could affect CFTR gating
as seen in the literature. In Figure E1, the authors demonstrated their ability to use CFTRFRET
proteins in electrophysiological experiments and concluded that its gating kinetics are similar to
WT-CFTR (but see comment 19 below).

Yes, **Figure 1e** displays electrophysiological recordings of wild-type CFTR. The comparison is
justified by measurements made in **Extended Data Figure 1**, which demonstrate that the
CFTR_{FRET} and wild-type variants have similar time-courses of current relaxation in response to
rapid ATP application or withdrawal. Dependences on ATP and PKA, and sensitivity to
GLPG1837 are also similar. Finally, ATP hydrolysis, which reflects flux through the gating cycle
is also comparable between the two variants.

Consistent with literature reports (e.g. Cui et al. 2006 and Mense et al. 2006), we found that
substituting all cysteines in CFTR to serine or alanine caused expression defects that precluded
purification. We therefore conducted an extensive screen of cysteine variants to isolate residues
causing this unwanted effect. Residues C524 and C590 are buried within NBD1 and could be
retained without a great extent of non-specific fluorophore labelling. C76L and C592M
substitutions were found to not affect expression. Collectively, these changes to the cysteine-less
CFTR variant allowed purification.

While cysteine substitutions may cause subtle changes to gating, our observations seem
incompatible with large gating defects that would alter the major conclusions drawn in this
manuscript.

To further substantiate that key gating properties are retained by the CFTR_{FRET} variant we add a
comparison of open probabilities and open dwell times for the wild-type CFTR and the
fluorophore-conjugated CFTR_{FRET} to the revision (**Extended Data Figure 11,m**).

Cui, L., et al. The role of cystic fibrosis transmembrane conductance regulator phenylalanine 508
side chain in ion channel gating. *J Physiol* **572**, 347-358 (2006).

Mense, M., et al. In vivo phosphorylation of CFTR promotes formation of a nucleotide-binding
domain heterodimer. *EMBO J* **25**, 4728-4739 (2006).

17. Figure 1j: I don't understand the data points at 0 ATP. Do the authors still observe FRET
transitions even without ATP, meaning that the NBDs dimerize without ATP? In 1e, FRET always
stays at low FRET.

NBD-dimerization in the absence of ATP does occur, albeit infrequently (see e.g. **Figure 1e** - end
of the no ATP trace and **Extended Data Figure 10**). We also observe infrequent pore opening in
the absence of ATP (**Extended Data Figure 10**) consistent with literature reports (e.g. Wang et
al. 2010 and Mihályi et al. 2016). Both dimerization and pore opening also occur with Apyrase,
which was added to remove potential ATP contamination.

Our working model is that CFTR gating, like other stochastic processes in biology, is facilitated
by ATP binding and NBD dimerization, but can occur independently with very low efficiency and
duration with finite probability.

Wang, W., et al. ATP-independent CFTR channel gating and allosteric modulation by
phosphorylation. *Proc Natl Acad Sci USA* **107**, 3888–3893 (2010).

Mihályi, C., Töröcsik, B., & Csanády, L. Obligate coupling of CFTR pore opening to tight
nucleotide-binding domain dimerization. *eLife* **5**, e18164 (2016).

18. Figure 4b: In response to GLPG1837, the current trace of G551D barely changes as if
GLPG1837 has little effect. This potentiation effect is far from the reported ~30-fold stimulation
shown in Fig. 4g. Please comment on the discrepancy and consider replacing the recording with a
more representative one. Similar concerns are raised for the effect of GLPG1837 on L927P. The
single-channel activity in 4b does not change 20-fold as indicated in Fig. 4g.

The open probability of G551D is very low and gating is slow. Therefore, in any given 60 second
window we are likely to see zero or one open events. Neither is truly representative. We have
updated the current trace in **Figure 4b** to better represent the relative current stimulation with
GLPG1837.

The window of time presented for the L927P trace is also updated in **Figure 4b** of the revision.
The traces reflect the relative potentiation presented in **Figure 4g**. Because of the very short open
dwell time and only modestly affected closed dwell time of unpotentiated L927P CFTR, the open
probability may be perceived greater in sample traces than it is.

19. Figure 4f: It is shown that the Po of E1371Q is ~0.7 and increases to ~0.8 with GLPG1837. It
is puzzling how E1371Q has a Po of only 0.7. Numerous studies showed that with hydrolysis
eliminated, the closing rate of E1371Q (or E1371S) is >100-fold slower than WT, resulting in a
Po close to unity (Bompadre et al., 2005; Csanady et al., 2013; Vergani et al., 2003; Yu et al.,
2016). Indeed, the E1371Q single channel in Figure 1f seems to be almost always open, though it
is hard to determine whether those brief downward deflections are flicker closures or interburst
closures with this time scale. It would be helpful to expand an E1371Q single channel trace to
allow a better assessment of channel behavior in a quantitative manner.

The deflections are flicker-closures – however, the duration of flickers make them hard to
distinguish from inter-burst closures. Consistent with literature, we determine very slow current
relaxation for E1371Q upon ATP withdrawal in excised inside-out patches: $\tau_{closure} = 427 \pm 91$ s
(mean and standard error, n = 6 patches) (**Extended Data Figure 8a**). Thus, the vast majority of
the observed closures do not involve release of ATP from the consensus site.

We add an expanded view of the E1371Q single-channel trace to **Figure 1e** of the revision.

20. Figure E1i: From the raw trace, the Po of the FRET construct is apparently higher than the
WT. The open burst time of FRET variant is visibly longer than WT. Is this a consistent finding?
Please comment on the discrepancy.

We have updated the single-channel traces of the current-voltage relationship in **Extended Data**
**Figure 1i** to better reflect the open probability of the CFTR_{FRET} variant. We also add a comparison
of open probabilities and open dwell times for the phosphorylated wild-type and fluorophore-
conjugated CFTR_{FRET} variant channels (**Extended Data Figure 11-m**). We do not find a
statistically significant increase in open probability for the CFTR_{FRET} variant.

21. Figure E8g: What is the second arrow below the ADP+Pi for?

The multiple arrows aimed to illustrate that several molecular events occur within this step. The
second arrow is removed in the revision to avoid confusion.

22. Figure E11b, d, and e: In b, the simulated single-channel trace shows a much longer open time
than the experiment, while the closed time seems not affected much (or slightly shorter). However,
in d and e, the experimental and simulated dwell times match perfectly but the two closed times
deviate significantly. Please comment on this discrepancy.

The open burst dwell times of the experimental and simulated traces in **Extended Data Figure**
**11b** match, as is also reflected in the quantification in **Extended Data Figure 11d**. However, as
we do not have an accurate estimate of flicker-closure rates, this may give rise to the different
appearances of the open burst in experimental and simulated traces. Due to the limited bandwidth
of our experiments, we ignore flicker-closure events in idealization of experimental single-channel
recordings. The dwell time histogram therefore reflect the life-time of the open burst. To best
mimic the experimental analysis, the dwell times of the open burst were also extracted from
simulated traces, again ignoring flicker-closure events. In our revised manuscript we clarify these
points in the legend of **Extended Data Figure 11**.

23. Figure E11h: The experimental ATP turnover rate is 15/min (or 0.25/s), corresponding to one
ATP hydrolyzed every four seconds, a much longer time than a gating cycle time of wildtype
CFTR (~1 s). If, as proposed, multiple ATP hydrolysis cycles occur, the theoretical ATP hydrolysis
rate must be even higher than 1/s. As stated, there is a likelihood that this biochemical measurement
is not very accurate because of imperfect protein preparation. Alternatively, the proposed gating
model is overly simplified. Either way, I wonder whether the simulation really provides more
insights into the gating mechanism of CFTR. Sometimes it may be better to be vaguely right than
precisely wrong. I leave this to the authors to decide.

As we have elaborated in the response to Reviewer 1 point 12, we believe having a kinetic model
is useful to tie all experimental observables together and to guide future experiments.

References:

Bompadre, S. G., Cho, J. H., Wang, X., Zou, X., Sohma, Y., Li, M., & Hwang, T. C. (2005).
CFTR gating II: Effects of nucleotide binding on the stability of open states. *J Gen Physiol*,
125(4), 377-394. <https://doi.org/10.1085/jgp.200409228>

Csanady, L., Mihalyi, C., Szollosi, A., Torocsik, B., & Vergani, P. (2013). Conformational
changes in the catalytically inactive nucleotide-binding site of CFTR. *J Gen Physiol*, 142(1), 61-
73. <https://doi.org/10.1085/jgp.201210954>

Csanady, L., Vergani, P., & Gadsby, D. C. (2010). Strict coupling between CFTR's catalytic
cycle and gating of its Cl⁻ ion pore revealed by distributions of open channel burst durations.
*Proc Natl Acad Sci U S A*, 107(3), 1241-1246. <https://doi.org/10.1073/pnas.0911061107>

Jih, K. Y., Sohma, Y., Li, M., & Hwang, T. C. (2012). Identification of a novel post-hydrolytic
state in CFTR gating. *J Gen Physiol*, 139(5), 359-370. <https://doi.org/10.1085/jgp.201210789>

Lin, W.-Y., Jih, K.-Y., & Hwang, T.-C. (2014). A single amino acid substitution in CFTR
converts ATP to an inhibitory ligand. *The Journal of General Physiology*, 144(4), 311-320.
<https://doi.org/10.1085/jgp.201411247>

Lin, W. Y., Sohma, Y., & Hwang, T. C. (2016). Synergistic Potentiation of Cystic Fibrosis
Transmembrane Conductance Regulator Gating by Two Chemically Distinct Potentiators,
Ivacaftor (VX-770) and 5-Nitro-2-(3-Phenylpropylamino) Benzoate. *Mol Pharmacol*, 90(3), 275-
285. <https://doi.org/10.1124/mol.116.104570>

Tsai, M. F., Li, M., & Hwang, T. C. (2010). Stable ATP binding mediated by a partial NBD
dimer of the CFTR chloride channel. *J Gen Physiol*, 135(5), 399-414.
<https://doi.org/10.1085/jgp.201010399>

Vergani, P., Lockless, S. W., Nairn, A. C., & Gadsby, D. C. (2005). CFTR channel opening by
ATP-driven tight dimerization of its nucleotide-binding domains. *Nature*, 433(7028), 876-880.
<https://doi.org/10.1038/nature03313>

Vergani, P., Nairn, A. C., & Gadsby, D. C. (2003). On the mechanism of MgATP-dependent
gating of CFTR Cl⁻ channels. *J Gen Physiol*, 121(1), 17-36.

[http://www.ncbi.nlm.nih.gov/entrez/query.fcgi?cmd=Retrieve&db=PubMed&dopt=Citation&list](http://www.ncbi.nlm.nih.gov/entrez/query.fcgi?cmd=Retrieve&db=PubMed&dopt=Citation&list_uids=12508051)
uids=12508051

Yeh, H. I., Sohma, Y., Conrath, K., & Hwang, T. C. (2017). A common mechanism for CFTR
potentiators. *J Gen Physiol*, 149(12), 1105-1118. <https://doi.org/10.1085/jgp.201711886>

Yeh, H. I., Yeh, J. T., & Hwang, T. C. (2015). Modulation of CFTR gating by permeant ions. *J*
*Gen Physiol*, 145(1), 47-60. <https://doi.org/10.1085/jgp.201411272>
Yeh, H. I., Yu, Y. C., Kuo, P. L., Tsai, C. K., Huang, H. T., & Hwang, T. C. (2021). Functional
stability of CFTR depends on tight binding of ATP at its degenerate ATP-binding site. *J Physiol*,
599(20), 4625-4642. <https://doi.org/10.1113/JP281933>
Yu, Y. C., Sohma, Y., & Hwang, T. C. (2016). On the mechanism of gating defects caused by
the R117H mutation in cystic fibrosis transmembrane conductance regulator. *J Physiol*, 594(12),
3227-3244. <https://doi.org/10.1113/jp271723>
Zhou, Z., Wang, X., Liu, H. Y., Zou, X., Li, M., & Hwang, T. C. (2006). The two ATP binding
sites of cystic fibrosis transmembrane conductance regulator (CFTR) play distinct roles in gating
kinetics and energetics. *J Gen Physiol*, 128(4), 413-422. <https://doi.org/10.1085/jgp.200609622>

**Reviewer #3 (Remarks to the Author):**

The manuscript by Levring et al describes characterization of molecular cycles involved in
activation of the CFTR channels. Specifically, single molecule FRET studies done in parallel with
single channel electrophysiology enable to characterize the relationship between ATP binding,
hydrolysis and channel opening. Furthermore, how disease-causing mutations and therapeutic
correctors modulates these events are studied and discussed.

The study is very nicely crafted, the experiments are very well performed, and the conclusions are
convincing.

This work brings a clear view on a long-standing issue in the CF field and deserves publication.

We thank this Reviewer for these supportive summary comments.

Nevertheless, a number of points require clarification.

1/ FRET values

CryoEM structures of human CFTR have -so far- captured 2 conformations with the NBDs either
separated or dimerized. Structural data on ABC structures suggest more extended conformational
diversity with several different substates. The differences in FRET values observed in the present
manuscript may be indicative of such diversity. Therefore, the authors need to clearly assign -when
possible- the link between the FRET valued and the identified conformer.

The ATP-free CFTR shows a low FRET state at 0.25. Does this match the cryo-EM structures of
the NBD separated state ? Similarly does the 0.49 FRET value match the NBD-dimerized state for
the used probes? The authors need to run careful simulations (for examples with FPS, see Kalinin
et al) with the appropriate probe models to evaluate the calculated FRET value for each
conformation.

As requested, we performed all-atom molecular dynamics simulations of CFTR (PDB codes
5UAK and 6MSM) with the LD555 and LD655 dyes attached at their sites of labeling for
comparison with the FRET experiments. As previously described, we utilized a simplified
potential that maintains native contacts in the original structure while allowing the fluorophores to
explore all possible positions and conformations (Noel et al., 2016 and Girodat et al., 2020). Using
this approach, we can confirm that our FRET efficiencies are consistent with the existing CryoEM
structures of the two conformations (**Table R1**). For reference, we also included the distances
obtained from FPS as suggested by the Reviewer (Kalinin et al., 2012 and Sindbert et al., 2011),
which show larger differences, which we attribute to the simplified nature of the ball-and-stick
model used by the FPS program.

ATP-bound dephosphorylated CFTR shows a 0.28 FRET value. How does that translate in terms
of distance change given the R0 for these probes ?

Regarding the ATP-bound dephosphorylated state (0.28 FRET efficiency), this corresponds to an
inter-dye distance of 72.6 Å, which is a ~2 Å decrease from the apo state.

State	PDB	FRET Efficiency	FRET Distance	SMOG	FPS (R_{mp})	C_{α} - C_{α}
Deph. Apo	5UAK	0.25	74.5	72.1	65.9	69.9
Deph. ATP		0.28	72.6			
Phos. ATP	6MSM	0.49	62.4	66.0	55.4	46.4

Table R1. Comparison of inter-dye distances estimated from FRET and CryoEM structures.

Mean FRET efficiency values were translated into inter-dye distances (“FRET Distance”) using an R_0 value of 62 Å, which was previously established for the LD555-LD655 fluorophore pair (Girodat et al. 2020). Mean inter-dye distances (calculated between the central carbon of each dye’s polymethine chain) from all-atom, implicit-solvent simulations (SMOG; Noel et al. 2016) of CFTR with LD dyes attached over 50 million timesteps. FPS (R_{mp}) is the distance between the mean sampling positions from calculations using the ball-and-stick model in the FPS software (Kalinin et al., 2012 and Sindbert et al., 2011) using the parameters for the Cy3 and Cy5 dyes, which have a similar structure to the LD dyes. C_{α} - C_{α} is the distance in the PDB structures between the alpha carbons of the residues to which the dyes were attached. All distances are in Å.

Noel, J. K., Levi, M., Raghunathan, M., Lammert, H., Hayes, R. L., Onuchic, J. N., & Whitford,
P. C. (2016). SMOG 2: a versatile software package for generating structure-based models. PLoS
computational biology, 12(3), e1004794.

Girodat, D., Pati, A. K., Terry, D. S., Blanchard, S. C., & Sanbonmatsu, K. Y. (2020). Quantitative
comparison between sub-millisecond time resolution single-molecule FRET measurements and
10-second molecular simulations of a biosensor protein. PLoS computational biology, 16(11),
e1008293.

Kalinin, S., Peulen, T., Sindbert, S., Rothwell, P. J., Berger, S., Restle, T., ... & Seidel, C. A.
(2012). A toolkit and benchmark study for FRET-restrained high-precision structural modeling.
Nature methods, 9(12), 1218-1225.

Sindbert, S., Kalinin, S., Nguyen, H., Kienzler, A., Clima, L., Bannwarth, W., ... & Seidel, C. A.
(2011). Accurate distance determination of nucleic acids via Forster resonance energy transfer:
implications of dye linker length and rigidity. Journal of the American Chemical Society, 133(8),
2463-2480.

The corresponding cryo-EM structure showed in Figure E4 does not address these issues, in great
part due to the limited resolution. In fact, the authors only vaguely describe “proximal changes in
local structure”. What does that mean? In my opinion in its current state this structure does not
bring relevant information and should be either improved or removed from the manuscript.

We have performed additional cryo-EM analysis and improved the structure to 4.3 Å, which
enables us to position the TMDs and NBDs with confidence. Furthermore, the presence of ATP is
unambiguous (**Figure E4**). Comparing this structure with the two published conformations show
that the dephosphorylated CFTR retains the NBD-separated conformation even with ATP bound
at both NBDs. In addition to correlating the FRET value with structure, this structure also provides
an important missing piece in understanding CFTR regulation. It shows that ATP-binding to
dephosphorylated CFTR does not induce large-scale conformational changes, thus explaining why
R-domain phosphorylation is required for channel opening even at physiological ATP
concentrations (1-10 mM).

2/ The E1371Q mutant appear to be locked in the high FRET state during the time course of the
experiments, at least in the trace shown (100s, Fig1f). However, the distribution on Fig1c seems
to show a significant low FRET population (~5-10%), pretty much identical to that seen for the
“wt” CFTR (which can hydrolyze ATP and thus should spend more time in the NBD separated).

The fraction of E1371Q molecules that does not occupy the high FRET state reflects two
populations: 1) Infrequent NBD-separation events from the high FRET state (**Figure R6a**); 2) A
small fraction of nonresponsive molecules. The latter population does not respond to stimuli like
ATP addition and might reflect molecules that have denatured during handling, surface tethering,
and/or the small fraction of non-specifically fluorophore-labelled molecules (**Extended Data**
**Figure 2a**).

In that line, it is unclear to me why the authors see no NBD separation in this mutant after
phosphatase treatment (Fig E3). One may expect to see some separation over the course of the
measurement.

The NBD separation of E1371Q in response to phosphatase does occur, but exceedingly slow. The
zoomed-in view of **Extended Data Figure 3r** reveals a subtle inflection in the dimerized state
occupancy of E1371Q CFTR upon phosphatase injection (**Figure R6b**). The response is too slow
to accurately estimate a time-constant in our smFRET setup.

Our experiments suggest that autoinhibitory re-engagement of the R domain requires prior NBD-
separation. NBD-separation is limited by the rate of hydrolytic or non-hydrolytic ATP dissociation
from wild-type and E1371Q CFTR variants, respectively. Our best estimate of non-hydrolytic ATP
dissociation is from E1371Q current relaxation upon ATP withdrawal measured in inside-out
excised patches (**Extended Data Figure 8a**). We have measured the time constant of this reaction
to be $\tau_{closure} = 427 \pm 91$ s (mean and standard error, $n = 6$). A similar value was reported by David
Gadsby and colleagues (Vergani et al. 2005): $\tau_{closure} = 476$ s.

In the continued presence of ATP, non-hydrolytic ATP dissociation from dephosphorylated
E1371Q CFTR may lead to either R domain autoinhibition or ATP rebinding. Thus, 450 seconds
should be a lower bound on the predicted time-constant of relaxation. Altogether, this is consistent
with the slow rate of NBD separation measured in **Extended Data Figure 3r**.

**Figure R6 | E1371Q CFTR NBD-separation.**
**a.** Example single-molecule trace showing a rare NBD separation event (indicated by an asterisk) for E1371Q CFTR. CFTR was
phosphorylated and in the presence of 3 mM ATP. Top panel shows donor (green) and acceptor (red) fluorescence intensities.
Bottom panel shows FRET. **b.** Depopulation from the High FRET state after λ phosphatase injection (at the dashed line) for E1371Q
CFTR. The plot is a zoomed-in view of **Extended Data Figure 3r** in the manuscript.

Vergani, P., Lockless, S., Nairn, A. et al. CFTR channel opening by ATP-driven tight dimerization
of its nucleotide-binding domains. *Nature* **433**, 876–880 (2005).

In the same Figure, can the author comment on the difference in ATP concentration dependence
of channel gating vs NBD dimerization? At 3mM ADP 80% of the channels are still in dimers but
gating is almost gone.

Due to space-constraints we did not describe the origin of the asymmetry between ADP-mediated
inhibition of pore opening and NBD-dimerization. Our working model (**Figure R7**) posits that
ADP competition for the consensus site is sufficient to prevent opening of the pore, but is
insufficient to cause separation of NBDs. Thus, at intermediate ADP concentrations CFTR
channels are trapped in a configuration equivalent to the post-hydrolytic state which is also
transited during normal gating (**Figure 5**). At higher concentration, ADP competes for both
nucleotide-binding sites to inhibit both NBD-dimerization and pore opening.

**Figure R7 | Competitive inhibition by ADP.** At lower concentrations, ADP competes for ATP binding in the consensus site to
inhibit pore opening but not NBD dimerization. Higher ADP concentrations compete for ATP binding in the degenerate site to
inhibit NBD dimerization.

3/ Very little is shown regarding the membrane-embedded measurements.

What does the sentence “While the fraction of the molecules responsive to ATP was significantly
reduced” refer to ? Only single traces are shown. No statistics (“significantly” ?).

We refer to our response to Reviewer 1 point 11.

The text was revised to remove the word “*significantly*”.

4/ ATP binding sites

The authors use the W401A and Y1219A mutants originally described by Hwang and colleagues
(although mutated into Gly in that study). What do the authors mean by “destabilize ATP
binding” ?

It should be noted that the mutations decrease affinity but may not prevent it. For example, W401A
is still able to bind ATP (albeit with lower affinity) and 3mM is quite a large concentration.
Perturbing the ATP binding at the degenerate site may also alter the conformational equilibrium
of NBD1 that we recently described (Scholl et al, 2021). On P8 the authors state that for mutant
Y1219A “ATP binds only at the degenerate site”. This is not proven.

We agree with the Reviewer that the mutations may not completely abolish ATP binding and we
revised the text as the mutations used were designed “*to reduce the affinity for ATP*” and “*the*”

*Y1219A variant, which binds ATP principally at the degenerate site*” (page 7). However, the
fraction of such a dual ATP-occupied mutants is small and its presence will not alter the
conclusions we draw. Please see our response to Reviewer 2, point 3.

On P7, the authors describe that “the Y1219A variant, which binds ATP mainly at the degenerate
site, transitioned between NBD-dimerized and separated states slowly, and very rarely exhibited
channel opening” but they only show single traces. They should show distributions.

As suggested, the distributions of NBD dimerization and channel opening are now included as
**Figure 2e** and **2f**.

5/ Figure E3K is unclear to me. Is it expected to see such a significant effect at 0.1 and 0.3 mM
ADP while 3mM ATP is maintained (and while Fig E3I show predominantly NBD in dimer form)?

Based on the literature, we do expect to observe ADP inhibition under those experimental
conditions. The dose-dependence of competitive ADP inhibition described in **Figure E3K** is
consistent with measurements by Welsh and colleagues (Randak and Welsh 2005).

Randak, C. O. & Welsh, M. J. ADP inhibits function of the ABC transporter cystic fibrosis
transmembrane conductance regulator via its adenylate kinase activity. *Proc Natl Acad Sci USA*
**102**, 2216–2220 (2005).

I assume that the last part of the graph has 0mM ADP (correct?).

Yes, at the end of the current trace in **Extended Data Figure 3k** there is no ADP.

Why is the rebound much faster than what is observed during the first 2 mins?

We think that this is related to the ‘reversible rundown’ effect described by Hwang and colleagues
(Yeh et al. 2021). Hwang and colleagues describe a slow component for the current activation after
ATP application, that appears after extended intervals without ATP. The effect is ascribed to
release of ATP from the degenerate ATP binding site. Our model for competitive inhibition
described in **Figure R7** predicts that the ADP-dependent current inhibition observed in **Extended**
**Data Figure 3k** would not involve ATP dissociation from the degenerate site, and thus that no
reversible rundown should occur. Therefore, a slow component of activation is evident upon initial
ATP application, but not upon ADP withdrawal.

Yeh, H.-I., Yu, Y.-C., Kuo, P.-L., Tsai, C.-K., Huang, H.-T. and Hwang, T.-C. Functional stability
of CFTR depends on tight binding of ATP at its degenerate ATP-binding site. *J Physiol*, **599**:
4625-4642 (2021).

6/ The coupling ratio seems unclear to me. What if the two would anticorrelate ? The coupling
ratio would still be 1 ?

The definition of coupling ratio is clarified in the revised legend of **Figure 2** as: “*open probability*
*divided by dimerization probability.*” In interpreting the coupling ratio it is important to note that
NBD dimerization and pore opening correlate – at least for wild-type CFTR. In **Figures 1d,g** we
demonstrate correlation between current amplitude and NBD-dimerization probability in the time-
courses of activation and in the ATP dose-dependences for wild-type CFTR. The coupling ratio is
then the scalar that relates open probability and dimerization probability.

Whilst not essential for the interpretations made in the manuscript, the ratio may be most easily
understood by making a simplifying assumption: NBD-dimerization is required for pore opening.
In support of this assumption, we note that across all variants and conditions that we have tested,
the probability of dimerization exceeds the probability of opening. Under this assumption, the
coupling ratio reflects: What is the probability of pore opening, given that the NBDs have
dimerized? We find that ATP occupancy in the consensus site is crucial for the coupling, disease
mutations weaken coupling, and potentiators partially rescue coupling.

7/ P7 the authors state “We infer from these observations that channel opening probability is
enhanced by interactions of CFTR with the terminal phosphate moiety of the ATP molecule bound
within the consensus site.”. This should be readily testable as the terminal phosphate at the
consensus site is coordinated by K1250 and T1246. Point mutations should support this statement.

Effects of substituting the Walker A lysine of the consensus site (K1250) have been reported in
the literature. Consistent with our inferences and analogous to the E1371Q variant studied in the
present manuscript, substitution of K1250 results in CFTR variants that bind ATP but lose
hydrolysis activity, thereby trapping CFTR in dramatically prolonged open bursts (Gunderson and
Kopito, 1995 and Zeltwanger et al. 1999). Like E1371Q, these variants likely mimic the pre-
hydrolytic state of wild-type CFTR bound to two ATP molecules.

While T1246 coordinates the terminal phosphate moiety of the consensus site ATP, the side-chain
is also engaged in an important hydrogen bond with R555 across the NBD interface. Disruption
of the NBD dimer interface was demonstrated in the literature by T1246N substitution (Vergani et
al. 2005). Deconvolving the dual effects of substitution at this site is likely to be difficult.

Gunderson KL, Kopito RR. Conformational states of CFTR associated with channel gating: the
role ATP binding and hydrolysis. *Cell* **82**: 231-9 (1995).

Zeltwanger S, Wang F, Wang GT, Gillis KD, Hwang TC. Gating of cystic fibrosis transmembrane
conductance regulator chloride channels by adenosine triphosphate hydrolysis. Quantitative
analysis of a cyclic gating scheme. *J Gen Physiol* **113**:541-54 (1999).

Vergani P, Lockless SW, Nairn AC, Gadsby DC. CFTR channel opening by ATP-driven tight
dimerization of its nucleotide-binding domains. *Nature* **433**: 876–880 (2005).

8/ Mutation G551D

The authors show an intermediate FRET value of 0.37. The authors suggest that the mutation
introduce a “steric and electrostatic repulsion at the NBD interface” which is very vague. An Asp
at position 551 will likely prevent binding of ATP at the consensus site (at least in a NBD dimerize
state) because the acidic side chain would directly and closely face the terminal phosphate. This
suggests that the NBD1-NBD2 dimer is not formed at the consensus site and that the protein is
adopting an intermediate/different conformation. This also suggest that the ATP “stimulation” is
due to dimerization at the degenerate site. Therefore the “dimerized dwell time” on Fig E9E may
actually not correspond to dimerized state as “expected”, ie. where both sites are closed, as
observed by Cryo EM.

We agree and revised the manuscript. Page 10: “The high FRET, NBD-dimerized conformation is

likely to be different from that of E1371Q CFTR previously observed by cryo-EM, evident by a
lower coupling ratio (**Figure 4e**) and a shorter life-time (**Extended Data Figure 9e,f**). In
agreement with these data, a recent cryo-EM study showed that the G551D variant adopts
conformations in between the extremes observed for fully NBD-separated (PDB 5UAK) and -
dimerized (PDB 6MSM) states³⁵.”

Wang C., et al. Mechanism of dual pharmacological correction and potentiation of human CFTR.
*bioRxiv* 2022.10.10.510913; doi: <https://doi.org/10.1101/2022.10.10.510913> (2022).

In addition, the statement on P10 that “Importantly, both the G551D and L927P variants exhibited
ATP-dependent FRET response indicative of wild-type ATP binding affinities” is hard to
reconcile with the position of D551 in the mutant

The FRET response of G551D refers to formation of the intermediate conformation, in which ATP
binds in the NBD partially-separated conformation, where residue 551 likely does not contribute
to binding. We revised the manuscript (pages 10-11): “For both G551D and L927P variants, FRET
transitions exhibited ATP-dependence indicative of wild-type ATP binding affinities (**Figure E9i-**
**k**). Their functional defects are caused by deficits in ATP effecting formation of a tight NBD dimer
and in the coupling of the allosteric processes within NBD-dimerized CFTR that give rise to
channel opening (**Figure 4e**).”

Furthermore, the effect of potentiators on G551D are not clear, and not discussed.

Ivacaftor was approved to treat G551D patients, we would expect to see an effect. The authors
show some individual traces with GLPG1837, but not the approved drug Ivacaftor (why?) from
which there seem to be no significant effect. The authors should explore and or discuss this. In
contrast, stimulation of the L927P mutation does lead to fully dimerized state which does seem to
correlate with increase channel opening (although Fig 4f shows quite modest improvement).

GLPG1837 and Ivacaftor both have large effects on G551D pore opening and share a common
binding site (Liu et al. 2019), and mechanism of action (Yeh, et al. 2017). While Ivacaftor is
clinically approved, electrophysiologists (including us) favor GLPG1837 because Ivacaftor is very
hydrophobic, making it difficult to determine its effective concentration, which has led to some
inconsistent reporting for this compound (Csanády and Töröcsik, 2019). As Ivacaftor and
GLPG1837 share the same mechanism of action, lessons learned from one compound can be
applied to the other.

Our smFRET measurements show that GLPG1837 does not rescue the defect of G551D
dimerization. Consistently, Hunt and colleagues (Wang et al. 2022) described a lack of an effect
with Ivacaftor in their structural studies of a CFTR variant harboring the G551D substitution.

While GLPG1837 does promote formation of the dimer for the L927P CFTR variant, the key
observation is that the relative stimulation of open probability greatly exceeds the relative
stimulation of dimerization probability (**Figure 4f,g**). We conclude that the major effect of
potentiators is not to support transition from the separated to dimerized conformation. Rather,
potentiator binding acts allosterically on molecules that have already formed an NBD dimer by
favoring conformational changes that couple dimerization to pore opening.

As suggested, we have revised the text to incorporate these discussions (page 11): “*Consistent*
*with previous reports*⁴⁰⁻⁴³, we observed that both potentiators induced marked increases of channel
*open probabilities (Figures 4b,f, E10a). By comparison their effects on NBD dimerization were*
*much smaller for all CFTR variants tested (Figures 4b,f, E10a). For example, GLPG1837*
*increased the open probability of the G551D variant by more than 30-fold, while the change in*
*NBD dimerization was marginal (Figure 4b). This observation, together with the recent cryo-EM*
*study of the G551D CFTR variant in the presence of Ivacaftor*³⁵, *demonstrates that neither*
*Ivacaftor nor GLPG1837 promotes NBD dimerization. Similarly, for the L927P variant, the*
*relative stimulation of open probability greatly exceeded the relative stimulation of dimerization*
*probability (Figure 4f,g).”*

Liu, F., et al. Structural identification of a hotspot on CFTR for potentiation. *Science*, **364**, 1184–
1188 (2019).

Yeh, H. I., Sohma, Y., Conrath, K., & Hwang, T. C. A common mechanism for CFTR potentiators.
*J Gen Physiol* **149**, 1105–1118 (2017).

Csanády L, Töröcsik B. Cystic fibrosis drug ivacaftor stimulates CFTR channels at picomolar
concentrations. *Elife*. Jun 17;8:e46450 (2019).

Wang C., et al. Mechanism of dual pharmacological correction and potentiation of human CFTR.
*bioRxiv* 2022.10.10.510913; doi: <https://doi.org/10.1101/2022.10.10.510913> (2022).

Reviewer Reports on the First Revision:

Referees' comments:

Referee #1 (Remarks to the Author):

The authors have made a strong effort and well answered the questions raised in the review. The paper is worthy of publication in its current form.

Referee #2 (Remarks to the Author):

This revised manuscript by Levring et al. has been improved significantly. I only have the following comments for the authors to consider before they finalize their paper:

1. It remains vague to me what the word “hierarchical” means when is used to describe the gating mechanism of CFTR (in Abstract and discussion). The key issue under debate is whether NBD dimerization leads to obligatory gate opening (strict coupling model), or NBD dimerization does not guarantee gate opening (allosteric mechanism with a probabilistic relationship between NBD dimerization and gate opening). The data presented in the current paper apparently support the latter idea; thus, a simple allosteric gating mechanism should be sufficient.
2. On page 8, line 164: It is not appropriate to conclude that “NBD-dimerization is insufficient for channel opening” at this point (I don’t mean this conclusion is wrong), because the steady state data (open probability and probability of dimerized state) are used to draw conclusion for a kinetic step, channel opening. Here one can certainly conclude that the NBD-dimerized state is not equal to the open channel conformation. In fact, if one puts Figures 1i (red) and 3e together, NBD dimerization and gate opening seem very much coupled. The most important data for the conclusion that “NBD-dimerization is insufficient for channel opening” are those in Figure 3b and EFigure 3i: robust NBD-dimerization in conditions where negligible channel opening is expected.
3. Figure 3: The x-axis scale of upper panel in a is different from that in the lower panel. Is this correct? In b, the x-axis is not labeled in the upper panel. Is it the same as that in the lower panel?
4. Page 12, line 387: “nearly quantitatively”?

Referee #3 (Remarks to the Author):

In the revised version of the manuscript and in the rebuttal, Levring et al at thoroughly answering all comments and questions from the 3 reviewers (while not performing new experiments...).

Regarding my specific questions, I am overall satisfied with the answers, I would still make note of the following :

1/ Line 896. What is the author’s interpretation of a 2A change ?

2/ Line 944: the ~0.2 FRET population is present in phosphorylated WT and 1371Q, but not (to my

eye at least) in phosphorylated APO wt. How is that compatible with denatured subpopulation ?

3/ Line 1022. I was expecting a distance distribution.

4/Line 1069: After reading the authors reply and going back to the two references describing mutation of K1250 (which effectively prevents hydrolysis and not binding, like E1371Q), I'm still puzzled by the sentence "We infer from these observations that channel opening probability is enhanced by interactions of CFTR with the terminal phosphate moiety of the ATP molecule bound within the consensus site." These interactions are established from the structure: K1250, T1246, Q1291 and the backbone nitrogen of G551. They almost perfectly conserved in the degenerate site (K464, T460, Q493 and G1349), suggesting that their assumption may not be correct and that something else must be at play.

**Author Rebuttals to First Revision:**

**Point-by-point response**

We sincerely thank the Reviewers for their added time and comments.

**Referee #1 (Remarks to the Author):**

The authors have made a strong effort and well answered the questions raised in the review. The
paper is worthy of publication in its current form.

**Referee #2 (Remarks to the Author):**

This revised manuscript by Levring et al. has been improved significantly. I only have the
following comments for the authors to consider before they finalize their paper:

1. It remains vague to me what the word “hierarchical” means when is used to describe the gating
mechanism of CFTR (in Abstract and discussion). The key issue under debate is whether NBD
dimerization leads to obligatory gate opening (strict coupling model), or NBD dimerization does
not guarantee gate opening (allosteric mechanism with a probabilistic relationship between NBD
dimerization and gate opening). The data presented in the current paper apparently support the
latter idea; thus, a simple allosteric gating mechanism should be sufficient.

We have changed the title to “*CFTR function, pathology and pharmacology at single-molecule*
*resolution*”.

The wordings in the abstract, introduction, and discussion have been adjusted. Page 2: “*CFTR*
*exhibits an **allosteric** gating mechanism in which conformational changes within the NBD-*
*dimerized channel, governed by ATP hydrolysis, regulate chloride conductance*”. Page 3: “*The*
*information obtained reveals an **allosteric** gating mechanism...*”. Page 11: “*Strikingly, the*
***allosteric** relationship evidenced between NBD dimerization and pore opening held true across*
*diverse conditions and CFTR variants...*”.

2. On page 8, line 164: It is not appropriate to conclude that “NBD-dimerization is insufficient for
channel opening” at this point (I don’t mean this conclusion is wrong), because the steady state
data (open probability and probability of dimerized state) are used to draw conclusion for a kinetic
step, channel opening. Here one can certainly conclude that the NBD-dimerized state is not equal
to the open channel conformation. In fact, if one puts Figures 1i (red) and 3e together, NBD
dimerization and gate opening seem very much coupled. The most important data for the
conclusion that “NBD-dimerization is insufficient for channel opening” are those in Figure 3b and
EFigure 3i: robust NBD-dimerization in conditions where negligible channel opening is expected.

As suggested, we simplified the statement on page 6 to “*We thus conclude that both conductive*
*and non-conductive NBD-dimerized states must exist.*”.

3. Figure 3: The x-axis scale of upper panel in a is different from that in the lower panel. Is this
correct? In b, the x-axis is not labeled in the upper panel. Is it the same as that in the lower panel?

The interpretation of the Reviewer is correct. We have now labelled the x-axes of all the panels to
avoid confusion.

4. Page 12, line 387: “nearly quantitatively”?

We deleted “nearly quantitatively”, it now reads “*At physiological ATP concentrations, fully*
*phosphorylated CFTR remains NBD dimerized for many cycles of ATP turnover and pore*
*opening.*”

**Referee #3 (Remarks to the Author):**

In the revised version of the manuscript and in the rebuttal, Levring et al at thoroughly answering
all comments and questions from the 3 reviewers (while not performing new experiments...).
Regarding my specific questions, I am overall satisfied with the answers, I would still make note
of the following:

1/ Line 896. What is the author’s interpretation of a 2Å change?

Our structure clearly demonstrated that ATP binding to the dephosphorylated CFTR does not lead
to large-scale conformational change. Therefore the 2 Å change likely reflects local
conformational changes of the sites of labelling as we stated on page 5: “*The small shift in FRET*
*efficiency is likely due to local changes that affect either the position and/or dynamics of the sites*
*of labelling*”.

2/ Line 944: the ~0.2 FRET population is present in phosphorylated WT and 1371Q, but not (to
my eye at least) in phosphorylated APO wt. How is that compatible with denatured subpopulation?

Given that the 0.2 FRET population does not resolve from the dominant 0.25 population for the
phosphorylated APO wt condition, it may simply be hidden in the left shoulder of the 0.25 peak.
The same is true for dephosphorylated APO wt and dephosphorylated ATP wt conditions.

3/ Line 1022. I was expecting a distance distribution.

We believe it is more appropriate to present the distributions of NBD dimerization and channel
opening, thus we can directly compare the Y1219A variant with the WT CFTR and other variants.

4/Line 1069: After reading the authors reply and going back to the two references describing
mutation of K1250 (which effectively prevents hydrolysis and not binding, like E1371Q), I’m still
puzzled by the sentence “We infer from these observations that channel opening probability is
enhanced by interactions of CFTR with the terminal phosphate moiety of the ATP molecule bound
within the consensus site.” These interactions are established from the structure: K1250, T1246,
Q1291 and the backbone nitrogen of G551. They almost perfectly conserved in the degenerate site
(K464, T460, Q493 and G1349), suggesting that their assumption may not be correct and that
something else must be at play.

For clarity we removed this sentence from the manuscript.